# SING: SDE Inference via Natural Gradients

**Amber Hu**[*]
Stanford University
amberhu@stanford.edu

**Henry D. Smith**[*]
Stanford University
smithhd@stanford.edu

**Scott W. Linderman**
Stanford University
swl1@stanford.edu

## Abstract

Latent stochastic differential equation (SDE) models are important tools for the unsupervised discovery of dynamical systems from data, with applications ranging from engineering to neuroscience. In these complex domains, exact posterior inference of the latent state path is typically intractable, motivating the use of approximate methods such as variational inference (VI). However, existing VI methods for inference in latent SDEs often suffer from slow convergence and numerical instability. Here, we propose *SDE Inference via Natural Gradients* (SING), a method that leverages natural gradient VI to efficiently exploit the underlying geometry of the model and variational posterior. SING enables fast and reliable inference in latent SDE models by approximating intractable integrals and parallelizing computations in time. We provide theoretical guarantees that SING approximately optimizes the intractable, continuous-time objective of interest. Moreover, we demonstrate that better state inference enables more accurate estimation of nonlinear drift functions using, for example, Gaussian process SDE models. SING outperforms prior methods in state inference and drift estimation on a variety of datasets, including a challenging application to modeling neural dynamics in freely behaving animals. Altogether, our results illustrate the potential of SING as a tool for accurate inference in complex dynamical systems, especially those characterized by limited prior knowledge and non-conjugate structure.[1]

## 1 Introduction

Stochastic differential equations (SDEs) are powerful tools for modeling complex, time-varying systems in scientific domains such as physics, engineering, and neuroscience [1, 2]. In many real-world settings, we do not observe the state of the system directly, but instead must infer it from noisy measurements. For example, in neuroscience, we often wish to uncover continuously evolving internal brain states underlying high-dimensional neural recordings [3]. *Latent SDE models* address this problem by positing a continuous-time stochastic latent process that generates observations through a measurement model.

A key challenge in latent SDE modeling is posterior inference over the latent state trajectory. Accurate inference enables principled understanding and prediction of the underlying state, but it is often computationally intractable to perform analytically due to non-conjugate model structure. Methods for approximating the intractable posterior include Markov chain Monte Carlo [4, 5], approximate Bayesian smoothing [2, 6], and variational inference (VI) [7–11]. In particular, VI offers a flexible and scalable framework for approximate inference in latent SDE models. In their seminal work, Archambeau et al. [7, 8] formulate the VI problem for latent SDEs as constrained optimization over the family of linear, time-varying SDEs. While variants of this approach have been widely employed [9, 12–15], they often suffer from slow convergence and numerical instability. These shortcomings motivate the need for more efficient and robust VI methods in this setting.

---

[*]Equal contribution
[1]Code: https://github.com/lindermanlab/sing

39th Conference on Neural Information Processing Systems (NeurIPS 2025).

In this work, we introduce SING: SDE Inference via Natural Gradients, a method that leverages natural gradient variational inference (NGVI) for latent SDE models. SING exploits the underlying geometry of the prior process and variational posterior, which leads to fast and stable latent state inference, even in challenging models with highly nonlinear dynamics and/or non-conjugate observation models. Our contributions are as follows:

(i) We derive natural gradient updates for latent SDE models and introduce methods for computing them efficiently;

(ii) We show how SING can be parallelized over time, enabling scalability to long sequences;

(iii) We provide theoretical guarantees that SING approximately optimizes the continuous-time objective of interest; and

(iv) We demonstrate that improved inference with SING enables more accurate drift estimation, including in models with Gaussian process drifts.

## 2 Background and problem statement

### 2.1 Latent stochastic differential equation models

We consider the class of models in which the latent states evolve according a stochastic differential equation (SDE) and give rise to conditionally independent observations. The prior on the latent states $\boldsymbol{x} \in \mathbb{R}^D$ takes the form

$$p(\boldsymbol{x}) : d\boldsymbol{x}(t) = \boldsymbol{f}(\boldsymbol{x}(t))dt + \boldsymbol{\Sigma}^{\frac{1}{2}}d\boldsymbol{w}(t), \quad \boldsymbol{x}(0) \sim \mathcal{N}(\boldsymbol{\nu}, \boldsymbol{V}), \quad 0 \le t \le T_{\max} \tag{1}$$

where $\boldsymbol{f} : \mathbb{R}^D \to \mathbb{R}^D$ is the drift function describing the system dynamics, $\boldsymbol{\Sigma}$ is a time-homogeneous noise covariance matrix, and $(\boldsymbol{w}(t))_{0 \le t \le T_{\max}}$ is a standard $D$-dimensional Brownian motion. $\boldsymbol{\nu}$ and $\boldsymbol{V}$ are the mean and covariance of the initial state. For our exposition, we will model $\boldsymbol{f}$ as a deterministic drift function with parameters $\boldsymbol{\theta}$. We will subsequently expand our scope in Section 3.5 to the case of a Gaussian process prior on $\boldsymbol{f}$. For simplicity of presentation, we assume $\boldsymbol{f}$ is time-homogeneous; however, our framework can be readily extended to the time-inhomogeneous case as well.

While the latent trajectory $(\boldsymbol{x}(t))_{0 \le t \le T_{\max}}$ is unobserved, we observe data $\mathcal{D} = \{t_i, \boldsymbol{y}(t_i)\}_{i=1}^n$, where $t_i \in [0, T_{\max}]$ are observation times and $\boldsymbol{y}(t_i) \in \mathbb{R}^N$ are corresponding observations modeled as

$$\mathbb{E}[\boldsymbol{y}(t_i) \mid \boldsymbol{x}] = g\left(\boldsymbol{C}\boldsymbol{x}(t_i) + \boldsymbol{d}\right). \tag{2}$$

Here, $g(\cdot)$ is a pre-determined inverse link function and $\{\boldsymbol{C}, \boldsymbol{d}\}$ are learnable hyperparameters defining an affine mapping from latent to observed space. Of particular interest is the setting $D \ll N$, where the latent space is much lower-dimensional than the observations. We provide a schematic of the generative model in Figure 1A.

For the latent SDE model, we aim to solve two problems: (i) inferring a posterior over the latent trajectories $\boldsymbol{x}$, conditional on the data $\mathcal{D}$ and (ii) learning the prior and output hyperparameters, collectively denoted by $\boldsymbol{\Theta} = \{\boldsymbol{\theta}, \boldsymbol{C}, \boldsymbol{d}\}$.

### 2.2 Natural gradient variational inference for exponential family models

SING leverages natural gradient variational inference (NGVI) to perform approximate posterior inference in latent SDE models. Before presenting our method, we first provide relevant background on NGVI for exponential families. In this section, we will consider the general VI problem in which we have a variational posterior $\bar{q}(\boldsymbol{z}|\boldsymbol{\eta})$ with parameters $\boldsymbol{\eta} \in \mathbb{R}^p$. The goal of VI is to maximize a lower bound $\mathcal{L}(\boldsymbol{\eta})$ to the marginal log likelihood [16, 17].

One way to maximize $\mathcal{L}(\boldsymbol{\eta})$ is via natural gradient ascent [18] with respect to $\boldsymbol{\eta}$, which adjusts the standard gradient update to account for the geometry of the distribution space. In practice, this corresponds to the update rule

$$\boldsymbol{\eta}^{(j+1)} = \arg\min_{\boldsymbol{\eta}} -\boldsymbol{\eta}^\top \nabla_{\boldsymbol{\eta}}\mathcal{L}(\boldsymbol{\eta}^{(j)}) + \frac{1}{\rho} \cdot \underbrace{\frac{1}{2}(\boldsymbol{\eta} - \boldsymbol{\eta}^{(j)})^\top \mathcal{F}(\boldsymbol{\eta}^{(j)})(\boldsymbol{\eta} - \boldsymbol{\eta}^{(j)})}_{\approx \mathrm{D_{KL}}\left(\bar{q}(\boldsymbol{z}|\boldsymbol{\eta}^{(j)}) \,\|\, \bar{q}(\boldsymbol{z}|\boldsymbol{\eta})\right)} \tag{3}$$

$$\implies \boldsymbol{\eta}^{(j+1)} = \boldsymbol{\eta}^{(j)} + \rho[\mathcal{F}(\boldsymbol{\eta}^{(j)})]^{-1}\nabla_{\boldsymbol{\eta}}\mathcal{L}(\boldsymbol{\eta}^{(j)}), \tag{4}$$

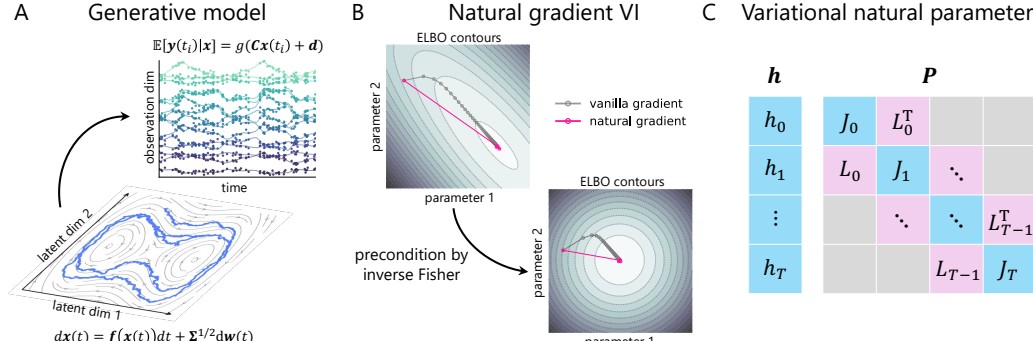

Figure 1: A schematic figure describing SING. **A**: In the generative model, a low-dimensional latent SDE gives rise to noisy, conditionally independent observations at timestamps $\{t_i\}_{i=1}^{n}$. **B**: SING leverages NGVI to perform fast and reliable approximate posterior inference in latent SDE models. NGVI exploits the geometry of the model by effectively preconditioning updates by an inverse Fisher information matrix, often leading to faster convergence than vanilla gradient ascent. **C**: On a discretized time grid $\boldsymbol{\tau}$, the variational posterior is a multivariate Gaussian distribution with a block tridiagonal precision matrix.

where $\boldsymbol{\eta}^{(j)}$ denotes the variational parameter at iteration $j$, $\rho$ is the step size, and $\mathcal{F}(\boldsymbol{\eta}) = \mathbb{E}_{\bar{q}(\boldsymbol{z}|\boldsymbol{\eta})}\left[\nabla_{\boldsymbol{\eta}} \log \bar{q}(\boldsymbol{z}|\boldsymbol{\eta})\nabla_{\boldsymbol{\eta}} \log \bar{q}(\boldsymbol{z}|\boldsymbol{\eta})^{\top}\right]$ is the Fisher information matrix. The key idea of natural gradient methods [17–21] is to perform an update that both increases the ELBO and remains close to the current variational distribution (in KL divergence). This interpretation relies upon a second-order Taylor expansion of the KL divergence, as denoted in eq. (3); we provide a derivation in Appendix B.1. See Figure 1B for a geometric interpretation of natural gradient ascent.

However, the inverse Fisher information in the update eq. (4) is intractable to compute when $\boldsymbol{\eta}$ is high-dimensional. This is common in models with temporal structure, including the latent SDE model presented in Section 2.1. The key insight behind NGVI [22–27] is that eq. (4) can be computed efficiently when $\bar{q}(\boldsymbol{z}|\boldsymbol{\eta})$ constitutes a minimal exponential family:

$$\bar{q}(\boldsymbol{z}|\boldsymbol{\eta}) = h(\boldsymbol{z}) \exp\left(\langle \boldsymbol{\eta}, \boldsymbol{T}(\boldsymbol{z})\rangle - A(\boldsymbol{\eta})\right), \quad A(\boldsymbol{\eta}) = \log\left(\int h(\boldsymbol{z}) \exp\left(\langle \boldsymbol{\eta}, \boldsymbol{T}(\boldsymbol{z})\rangle\right) d\boldsymbol{z}\right). \quad (5)$$

Here, $\boldsymbol{\eta}$ are called the natural parameters of the exponential family, $\boldsymbol{T}(\boldsymbol{z})$ are the sufficient statistics, $\boldsymbol{\mu} = \mathbb{E}_{\bar{q}(\boldsymbol{z}|\boldsymbol{\eta})}[\boldsymbol{T}(\boldsymbol{z})]$ are the mean parameters, and $A(\boldsymbol{\eta})$ is the log normalizer. The exponential family density eq. (5) is defined for all $\boldsymbol{\eta}$ for which $A(\boldsymbol{\eta}) < \infty$. We refer the reader to Appendix B for an overview of exponential families. Let $\boldsymbol{\mu}(\boldsymbol{\eta}) := \mathbb{E}_{\bar{q}(\boldsymbol{z}|\boldsymbol{\eta})}[\boldsymbol{T}(\boldsymbol{z})]$ be the bijective mapping from natural to mean parameters. In Appendix B, we show that the Fisher information is equal to the Jacobian of the mapping $\boldsymbol{\mu}(\boldsymbol{\eta})$ i.e., $\mathcal{F}(\boldsymbol{\eta}) = \nabla_{\boldsymbol{\eta}}\boldsymbol{\mu}(\boldsymbol{\eta})$. As a result, eq. (4) can be rewritten as

$$\boldsymbol{\eta}^{(j+1)} = \boldsymbol{\eta}^{(j)} + \rho \nabla_{\boldsymbol{\mu}}\mathcal{L}(\boldsymbol{\mu}^{(j)}). \quad (6)$$

This reparametrization avoids the need to compute or invert a large Fisher information matrix, allowing for tractable updates even for high-dimensional parameter spaces.

## 2.3 Variational inference in latent SDE models

In the latent SDE model in Section 2.1, one can show that the true posterior $p(\boldsymbol{x}|\mathcal{D})$ is a time-varying SDE with the same diffusion coefficient as the prior:

$$p(\boldsymbol{x} \mid \mathcal{D}) : d\boldsymbol{x}(t) = \tilde{\boldsymbol{f}}(\boldsymbol{x}(t), t|\mathcal{D})dt + \boldsymbol{\Sigma}^{\frac{1}{2}} d\boldsymbol{w}(t), \quad \boldsymbol{x}(0) \sim p_{\mathcal{D}}, \quad 0 \le t \le T_{\max}.$$

The drift $\tilde{\boldsymbol{f}}$ of the posterior SDE is the solution to a linear partial differential equation (PDE) [28–31]. In the case where the prior eq. (1) is a linear SDE and the observation model eq. (2) is linear and Gaussian, both $\tilde{\boldsymbol{f}}$ and the marginal density of the posterior $p(\boldsymbol{x}(t)|\mathcal{D})$ can be computed via a Kalman-Bucy smoother [32]. However, in non-conjugate settings, neither quantity admits a closed-form, and solving for them numerically using PDE solvers can be computationally expensive.

We propose a VI method for approximate inference on the latent trajectories $(\boldsymbol{x}(t))_{0 \le t \le T_{\max}}$. As in prior works [7, 8], we posit a variational family $q(\boldsymbol{x})$ of linear time-varying SDEs,

$$q(\boldsymbol{x}) : d\boldsymbol{x}(t) = (\boldsymbol{A}(t)\boldsymbol{x}(t) + \boldsymbol{b}(t))dt + \boldsymbol{\Sigma}^{\frac{1}{2}} d\boldsymbol{w}(t), \quad \boldsymbol{x}(0) \sim \mathcal{N}(\boldsymbol{m}_0, \boldsymbol{S}_0), \quad 0 \le t \le T_{\max}, \quad (7)$$

where the variational parameters are $(\boldsymbol{A}(t))_{0 \leq t \leq T_{\max}}$ and $(\boldsymbol{b}(t))_{0 \leq t \leq T_{\max}}$. In other words, we approximate the posterior drift $\tilde{\boldsymbol{f}}$ with a linear function $\boldsymbol{f}_q(\boldsymbol{x}(t), t) := \boldsymbol{A}(t)\boldsymbol{x}(t) + \boldsymbol{b}(t)$ at each time.

In the setting where we have multiple independent trials $\ell$ sampled from the generative model $(\boldsymbol{x}^{(\ell)}(t))_{0 \leq t \leq T_{\max}}, \{\boldsymbol{y}^{(\ell)}(t_i)\}_{i=1}^n$, the true posterior factorizes across trials. Hence, we choose our variational approximation to also factorize across trials. In other words, we approximate the posterior for each trial $p(\boldsymbol{x}^{(\ell)}|\mathcal{D})$ with a linear, time-varying SDE as in eq. (7), where the variational parameters are allowed to differ across trials. In practice, we find that this variational family is highly flexible (see Section 5) and is capable of accurately inferring latents sampled according to SDEs with highly nonlinear (e.g., bifurcating) dynamics.

Our choice of variational family (7) yields the following evidence lower bound (ELBO) to the marginal log likelihood:

$$\mathcal{L}_{\text{cont}}(q, \boldsymbol{\Theta}) := \mathbb{E}_q[\log p(\boldsymbol{y}|\boldsymbol{x}, \boldsymbol{\Theta})] - \mathrm{D}_{\text{KL}}\left(q(\boldsymbol{x}) \,\|\, p(\boldsymbol{x}|\boldsymbol{\Theta})\right). \tag{8}$$

Our goal is to maximize $\mathcal{L}_{\text{cont}}(q, \boldsymbol{\Theta})$ with respect to $q$, given $\boldsymbol{\Theta}$ fixed. However, this objective is typically intractable to optimize directly. In practice, a common approach is to instead perform inference over a finite-dimensional distribution $q(\boldsymbol{x}_{0:T})$, where $\boldsymbol{x}_i := \boldsymbol{x}(\tau_i)$ and $\boldsymbol{\tau} = \{\tau_i\}_{i=0}^T \subseteq [0, T_{\max}]$, $\tau_0 = 0$, $\tau_T = T_{\max}$ is a set of time points that includes observation times $\{t_i\}_{i=1}^n$.

When the prior SDE $p(\boldsymbol{x})$ is nonlinear, the transition distribution $p(\boldsymbol{x}_{0:T})$ does not have a closed-form expression. Therefore, we approximate $p$ with its Euler–Maruyama discretization, which we call $\tilde{p}$. Since both $\tilde{p}$ and $q$ have Gaussian transition densities, this leads to a tractable approximation to the continuous-time ELBO $\mathcal{L}_{\text{cont}}(q, \boldsymbol{\Theta})$,

$$\mathcal{L}_{\text{approx}}(q, \boldsymbol{\Theta}) := \mathbb{E}_q[\log p(\boldsymbol{y}|\boldsymbol{x}, \boldsymbol{\Theta})] - \mathrm{D}_{\text{KL}}\left(q(\boldsymbol{x}_{0:T}) \,\|\, p(\boldsymbol{x}_{0:T}|\boldsymbol{\Theta})\right). \tag{9}$$

For derivations of the ELBOs $\mathcal{L}_{\text{cont}}$ and $\mathcal{L}_{\text{approx}}$, see Appendix C.

## 3 SING: SDE Inference via Natural Gradients

### 3.1 NGVI for latent SDE models

We propose SING, a method that uses NGVI to optimize the ELBO $\mathcal{L}_{\text{approx}}$ in the latent SDE model. SING can be applied to any combination of prior drift and likelihood model and enjoys fast convergence by exploiting the geometry of the natural parameter space. SING builds upon recent work from Verma et al. [33]; however, it has several notable differences that we detail in Appendix I.

We begin by recognizing that since eq. (7) is Gaussian and Markovian, its marginalized distribution $q(\boldsymbol{x}_{0:T})$ can be written as a minimal exponential family,

$$q(\boldsymbol{x}_{0:T}|\boldsymbol{\eta}) = \exp\left(-\frac{1}{2}\sum_{i=0}^T \boldsymbol{x}_i^\top \boldsymbol{J}_i \boldsymbol{x}_i - \sum_{i=0}^{T-1} \boldsymbol{x}_{i+1}^\top \boldsymbol{L}_i \boldsymbol{x}_i + \sum_{i=0}^T \boldsymbol{h}_i^\top \boldsymbol{x}_i - A(\boldsymbol{\eta})\right). \tag{10}$$

In eq. (10), the natural parameters are $\boldsymbol{\eta} = \left[\{\boldsymbol{h}_i, -\frac{1}{2}\boldsymbol{J}_i\}_{i=0}^T, \{-\boldsymbol{L}_i^\top\}_{i=0}^{T-1}\right]$ and the corresponding sufficient statistics are $\boldsymbol{T}(\boldsymbol{x}_{0:T}) = \left[\{\boldsymbol{x}_i, \boldsymbol{x}_i\boldsymbol{x}_i^\top\}_{i=0}^T, \{\boldsymbol{x}_{i+1}\boldsymbol{x}_i^\top\}_{i=0}^{T-1}\right]$. Since $q(\boldsymbol{x}_{0:T})$ is Markovian, its full precision matrix is block tridiagonal, as depicted in Figure 1C.

To apply the NGVI update eq. (6) to $\mathcal{L}_{\text{approx}}$, we must first consider $q(\boldsymbol{x}_{0:T})$ in terms of its mean parameters, which are simply the expectations of $\boldsymbol{T}(\boldsymbol{x}_{0:T})$ under $q$. We denote these together as $\boldsymbol{\mu} := \left[\{\boldsymbol{\mu}_{i,1}, \boldsymbol{\mu}_{i,2}\}_{i=0}^T, \{\boldsymbol{\mu}_{i,3}\}_{i=0}^{T-1}\right]$. From here, eq. (6) simplifies to the following updates on the natural parameters, which decompose over time steps $i = 0, \ldots, T$:

$$(\boldsymbol{h}_i^{(j+1)}, \boldsymbol{J}_i^{(j+1)}) = (1-\rho)(\boldsymbol{h}_i^{(j)}, \boldsymbol{J}_i^{(j)}) + \rho\nabla_{(\boldsymbol{\mu}_{i,1}, \boldsymbol{\mu}_{i,2})}\mathbb{E}_{q^{(j)}}\left[\log p(\boldsymbol{y}_i|\boldsymbol{x}_i)\delta_i + \log \tilde{p}(\boldsymbol{x}_{i+1}, \boldsymbol{x}_i|\boldsymbol{x}_{i-1})\right]$$

$$\boldsymbol{L}_i^{(j+1)} = (1-\rho)\boldsymbol{L}_i^{(j)} + \rho\nabla_{\boldsymbol{\mu}_{i,3}}\mathbb{E}_{q^{(j)}}\left[\log \tilde{p}(\boldsymbol{x}_{i+1}|\boldsymbol{x}_i)\right]. \tag{11}$$

In eq. (11), $q^{(j)} := q(\boldsymbol{x}_{0:T}|\boldsymbol{\eta}^{(j)})$ is the variational posterior at iteration $j$ and $\delta_i$ is an indicator denoting whether there is an observation at time $\tau_i$. We provide a derivation in Appendix D.

As written, the updates in eq. (11) are challenging to compute because they involve (i) approximating intractable expectations and (ii) converting from natural to mean parameters between iterations. In the subsequent sections, we propose computational methods to address each of these problems.

## 3.2 ELBO approximation guarantee

Prior to detailing how we compute the SING updates eq. (11), we provide theoretical justification that maximizing the ELBO $\mathcal{L}_{\text{approx}}$ will approximately maximize the true, continuous-time ELBO of interest $\mathcal{L}_{\text{cont}}$. Moreover, on a subset of the variational parameter space, the approximation error tends to zero uniformly as the size of the grid $\boldsymbol{\tau}$ tends to zero.

**Theorem 1:** Under mild regularity conditions (see Appendix E), there exists a constant $C$ such that for any mesh size $\Delta t := \max_{i=0}^{T-1} \Delta_i$, $\Delta_i = \tau_{i+1} - \tau_i$ of the time grid, $|\mathcal{L}_{\text{cont}} - \mathcal{L}_{\text{approx}}| \leq C(\Delta t)^{1/2}$.

*Proof sketch.* One compares the continuous- and discrete-time log-densities and applies error bounds on the Euler–Maruyama approximation. The full statement and proof can be found in Appendix E.

To the best of our knowledge, our work is the first in the topic of variational inference for latent SDE models to provide an explicit rate of convergence for the discrete time ELBO to its continuous counterpart.

## 3.3 Computing expectations of prior transition densities

The updates in eq. (11) rely on computing expectations of the form $\mathbb{E}_q[\log \tilde{p}(\boldsymbol{x}_{i+1}|\boldsymbol{x}_i)]$. This is difficult in general because the prior drift $\boldsymbol{f}(\boldsymbol{x}(t))$ can be nonlinear and the expectation is taken over the pairwise distribution $q(\boldsymbol{x}_i, \boldsymbol{x}_{i+1})$, which has total dimension $\mathbb{R}^{2D}$. In the following proposition, we show using Stein's lemma that we can rewrite the expectation over just $q(\boldsymbol{x}_i)$, which reduces the integration dimensionality to $\mathbb{R}^D$. The proof can be found in Appendix F.

**Proposition 1:** The term $\mathbb{E}_q[\log \tilde{p}(\boldsymbol{x}_{i+1}|\boldsymbol{x}_i)]$ can equivalently be written in terms of the mean parameters $\boldsymbol{\mu}_i$, $\boldsymbol{\mu}_{i+1}$ and the expectations $\mathbb{E}_q[\boldsymbol{f}(\boldsymbol{x}_i)]$, $\mathbb{E}_q[\boldsymbol{f}(\boldsymbol{x}_i)^\top \boldsymbol{\Sigma}^{-1} \boldsymbol{f}(\boldsymbol{x}_i)]$, and $\mathbb{E}_q[\boldsymbol{J_f}(\boldsymbol{x}_i)]$, where $\boldsymbol{J_f}(\boldsymbol{x}_i)$ denotes the Jacobian of $\boldsymbol{f}(\cdot)$ at $\boldsymbol{x}_i$.

The key implication of Proposition 1 is that the difficult expectations over the nonlinear prior transition terms can be reduced in dimensionality from $\mathbb{R}^{2D}$ to $\mathbb{R}^D$. This makes these terms in SING's updates significantly more tractable. In practice, there are many methods to compute these expectations, including Gauss-Hermite quadrature in low-dimensional systems or Monte Carlo in higher-dimensional systems. Empirically, we observe that even with a single Monte Carlo sample, SING performs accurate approximate inference over the latent trajectories in up to 50-dimensional latent spaces (Section 5.1).

## 3.4 Parallelizing SING

In many real-world settings, sequential inference on long time series can be computationally challenging. In this section, we show how SING can be fully parallelized, yielding efficient inference that scales logarithmically with the number of time steps.

The computational bottleneck of Section 3.1 is converting from natural parameters $\boldsymbol{\eta}$ to mean parameters $\boldsymbol{\mu}$ for the updates in eq. (11). By the identity $\nabla_{\boldsymbol{\eta}} A(\boldsymbol{\eta}) = \boldsymbol{\mu}$, we can perform this conversion by first computing $A(\boldsymbol{\eta})$ and then applying autodifferentiation with respect to $\boldsymbol{\eta}$. However, existing algorithms for this problem take $\mathcal{O}(D^3 T)$ time (see Appendix G.1 for further details). While $D$ is typically small in our setting, the linear scaling in $T$ may be prohibitive for long sequences.

Our key insight is that the log normalizer $A(\boldsymbol{\eta})$ of eq. (10) can be computed in parallel time by using associative scans [34]. Given a binary associative operator $\bullet$ and a sequence $(a_1, a_2, \ldots, a_n)$, the associative scan computes the prefix sum $(a_1, (a_1 \bullet a_2), \ldots, (a_1 \bullet a_2 \bullet \cdots \bullet a_n))$ in logarithmic time using a divide-and-conquer approach. By recognizing the log normalizer as

$$A(\boldsymbol{\eta}) = \log \int \underbrace{\exp\left(-\frac{1}{2}\boldsymbol{x}_0^\top \boldsymbol{J}_0 \boldsymbol{x}_0 + \boldsymbol{h}_0^\top \boldsymbol{x}_0\right)}_{:= a_{-1,0}} \prod_{i=1}^{T} \underbrace{\exp\left(-\frac{1}{2}\boldsymbol{x}_i^\top \boldsymbol{J}_i \boldsymbol{x}_i - \boldsymbol{x}_i^\top \boldsymbol{L}_{i-1} \boldsymbol{x}_{i-1} + \boldsymbol{h}_i^\top \boldsymbol{x}_i\right)}_{:= a_{i-1,i}} d\boldsymbol{x}_{0:T} \quad (12)$$

we can apply the associative scan to compute $A(\boldsymbol{\eta})$. We define the unnormalized Gaussian potentials in eq. (12) as the sequence elements $a_{i-1,i}$ and marginalize out one variable at a time via the binary associative operator $a_{i,j} \bullet a_{j,k} = \int a_{i,j} a_{j,k} d\boldsymbol{x}_j := a_{i,k}$. To marginalize out the variable $\boldsymbol{x}_T$, we also define the dummy element $a_{T,T+1} = 1$. As a result, the log of the full scan $(a_{-1,0} \bullet a_{0,1} \bullet \cdots \bullet a_{T,T+1})$ evaluates exactly to $A(\boldsymbol{\eta})$. This algorithm has time complexity $\mathcal{O}(D^3 \log T)$, resulting in significant

speedups as sequence length increases (Figure 5). In Appendix G.2, we derive the analytical expression for the Gaussian marginalization operator and analyze our algorithm's time complexity.

## 3.5 SING-GP: Drift estimation with Gaussian process priors

So far, we have assumed that the drift function $\boldsymbol{f}(\boldsymbol{x}(t)|\boldsymbol{\Theta})$ is deterministic with learnable hyperparameters $\boldsymbol{\theta}$. However, in practice, we often have limited prior knowledge about the functional form of the drift but would still like to learn or infer its structure. A natural approach for this problem is to place a Gaussian process (GP) [35] prior over the drift,

$$f_d(\cdot) \stackrel{\text{iid}}{\sim} \mathcal{GP}(0, \kappa_{\boldsymbol{\theta}}(\cdot, \cdot)), \quad \text{for } d = 1, \dots, D,$$

where $\boldsymbol{f}(\cdot) = [f_1(\cdot), \dots, f_D(\cdot)]^\top$, $\kappa_{\boldsymbol{\theta}}(\cdot, \cdot)$ is a kernel function, and $\boldsymbol{\theta}$ are kernel hyperparameters. This results in a well-studied model called the GP-SDE [13, 14], which is appealing for its ability to encode rich prior structure in dynamics and provide posterior uncertainty estimates. However, the practical utility of GP-SDEs has been limited by the inference methods used to fit them [7, 8], which we show in Section 5.2 often exhibit slow convergence and numerical instability.

Here, we extend SING to enable accurate and reliable inference in GP-SDE models, resulting in a new method called SING-GP. While we defer technical details to Appendix K, we outline the key ideas here. Following Duncker et al. [13], we introduce an augmented variational posterior of the form $q(\boldsymbol{x}_{0:T}, \boldsymbol{f})$ and use sparse GP techniques with inducing points [36] to perform approximate inference over $q(\boldsymbol{f})$. This leads to convenient closed-form updates on $q(\boldsymbol{f})$. In Section 5.2, we illustrate that SING-GP performs competitively to flexible parametric drift functions in low-data regimes, while also directly providing posterior uncertainty estimates on the drift. In Section 5.4, we use SING-GP on a challenging application of modeling latent neural dynamics in freely moving animals.

# 4 Related work

**VI for latent SDE models** A number of VI methods for latent SDEs posit larger variational families than eq. (7), consisting of non-Gaussian diffusion processes [10, 11, 37–41]. Ryder et al. [10] parameterize the variational family as the collection of diffusion processes with neural network drift and diffusion coefficients. Hereafter, we refer to the probability distributions belonging to this variational family as neural stochastic differential equations, or 'neural-SDEs'. Li et al. [11] also consider a neural-SDE variational family and introduce a stochastic adjoint method to optimize the continuous-time ELBO. While memory efficient, the stochastic adjoint method requires sampling from the variational posterior at each gradient step. Most recently, Bartosh et al. [41] introduce a simulation-free variational inference algorithm, 'SDE Matching', for neural-SDE variational families. In Appendix L.2, we compare SING to SDE Matching on a Lorenz attractor benchmark from [11]. We find that, although SING learns a Gaussian process approximation to the posterior, it outperforms SDE Matching on recovery of the ground truth latents and achieves competitive, albeit slightly worse, performance on drift recovery.

In settings where statistical and computational efficiency are critical, many works have studied Gaussian diffusion process variational families of the form eq. (7). In their seminal work, Archambeau et al. [7, 8] propose an algorithm 'VI for Diffusion Processes' (VDP), which optimizes the continuous-time ELBO in eq. (8) via Lagrange multipliers by incorporating the evolution of the marginal mean and covariance of $q(\boldsymbol{x}(t))$ as constraints. A detailed overview of VDP is provided in Appendix H. Duncker et al. [13] use sparse GP approximations to extend VDP to GP-SDE models, a method we refer to as 'VDP-GP'. Related to VDP, Course and Nair [9, 12] express the continuous-time ELBO entirely in terms of the marginal mean and covariance and then approximate these functions using a neural network encoder architecture.

Most relevant to our work, Verma et al. [33] propose applying NGVI to the discrete-time ELBO from eq. (9) in the case of a deterministic drift. Unlike SING, Verma et al. [33] do not address how to tractably compute $\mathcal{L}_{\text{approx}}$, and they do not suggest how to efficiently convert from natural to mean parameters. Moreover, no theoretical guarantees are given as to how well $\mathcal{L}_{\text{approx}}$ approximates $\mathcal{L}_{\text{true}}$. A complete comparison is given in Appendix I.

**Connection to GP regression problems** Several works have applied NGVI for efficient inference in GP regression problems by exploiting the fact that many GPs admit equivalent representations as

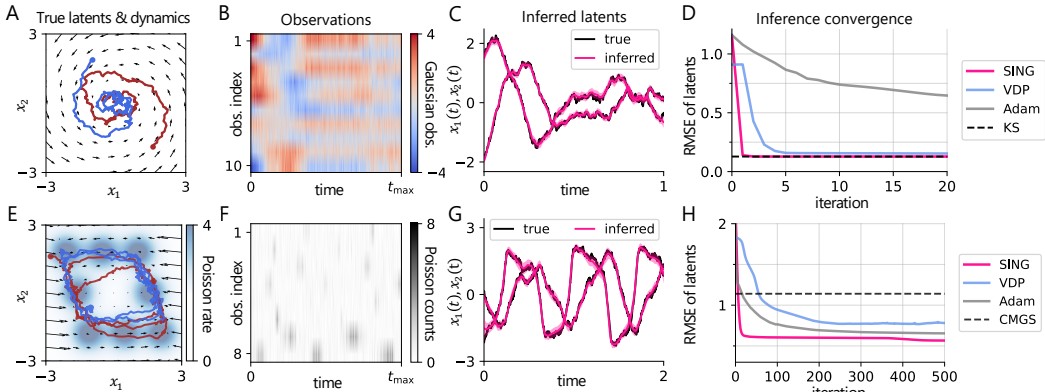

Figure 2: **(Top row)** We apply SING to a LDS with Gaussian observations. **A**: True latents are sampled from a LDS characterized by a stable spiral. **B**: Observations are 10-dimensional Gaussian variables. **C**: True vs. inferred latents on an example trial, with 95% posterior credible intervals. **D**: Comparison between SING and several baselines over iterations. **(Bottom row)** We apply SING to simulated place cell activity, where both the prior and observation models are nonlinear. **E**: True latents are sampled from a Van der Pol oscillator and represent trajectories through 2D space. Tuning curves modeled by radial basis functions represent expected firing rates at each location in latent space. **F**: Observations are Poisson spike counts for 8 neurons. **G, H**: See C, D.

linear time-invariant SDEs [2, 42]. For instance, Chang et al. [43] and Dowling et al. [44] consider GP regression problems with non-Gaussian likelihoods. To avoid cubic scaling in the number of data points, they reformulate the GP as a linear SDE and then apply NGVI to perform approximate inference in this non-conjugate model. Hamelijnck et al. [45] combine this approach with sparse GP approximations to perform inference in a spatio-temporal setting. While these works focus on linear SDE representations of GPs in regression problems, SING targets estimation of both the latent process and drift and applies to the broader class of *nonlinear* SDE priors.

**Parallelizing sequential models** Associative scans have been successfully applied to parallelize inherently sequential computations in areas such as classical Bayesian inference and deep sequence modeling [45–50]. Särkkä and García-Fernández [46] and Hassan et al. [47] derived associative scans for parallelizing Bayesian filtering and smoothing algorithms, and Hamelijnck et al. [45] extended these algorithms to inference in non-conjugate GP regression. We note that our associative scan in Section 3.4 is reminiscent of the one in Hassan et al. [47], which operates on potentials for inference in the hidden Markov model. However, SING differs in that it leverages these techniques to compute the log normalizer in a Gaussian Markov chain with no observations, which enables efficient parameter conversion in its natural gradient-based updates.

# 5 Experiments

## 5.1 Inference on synthetic data

We begin by applying SING to perform latent state inference in three synthetic datasets. Throughout this section, we assume that all hyperparameters $\Theta$ are fixed to their true values during model fitting, allowing us to directly assess the quality of latent state inference of SING compared to several baseline methods. We provide full experimental details for this section in Appendix L.1.

**Linear dynamics, Gaussian observations** First, we simulate 30 trials of a 2D latent SDE with linear drift exhibiting a counter-clockwise stable spiral using Euler–Maruyama discretization (Figure 2A). Then, we generate 10D Gaussian observations conditional on the latents (Figure 2B). We fit SING for 20 iterations and find that it is able to accurately recover the true latent states (Figure 2C).

We evaluate the inference quality of SING using root mean-squared error (RMSE) between the true and inferred latent states over iterations (Figure 2D). We compare SING to three baselines: (1) Kalman smoothing (KS) [51], (2) VDP [7], and (3) direct optimization of $\mathcal{L}_{approx}(\boldsymbol{\eta})$ using Adam [52]. Since this is a linear Gaussian model, exact posterior inference of $q(\boldsymbol{x}_{0:T})$ is tractable and can be obtained via KS. In Figure 2D, we find that SING recovers the true posterior in a single iteration,

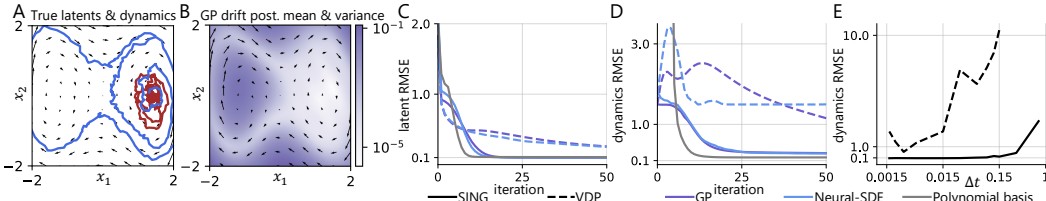

Figure 3: A comparison of SING and VDP for drift estimation. **A**: True latent trajectories evolve according to the 2D Duffing equation. **B**: Posterior mean (arrows) and variance (shading) corresponding to a GP prior drift with RBF kernel. SING-GP places high posterior uncertainty in regions unseen by the (true) latent trajectories. **C, D**: Latent RMSE and dynamics RMSE for SING and VDP across three classes of prior drift (GP, neural-SDE, polynomial basis). SING consistently outperforms VDP for hyperparameter learning, across all three drift classes. **E**: Dynamics RMSE for SING and VDP as a function of the grid size $\Delta t$ for the neural network drift.

consistent with the known result that NGVI performs one-step exact inference in conjugate models (see derivation in Appendix J). In contrast, VDP takes several iterations to converge in this setting; we provide theoretical justification for this finding in Appendix J. Finally, while Adam uses momentum and adaptive step sizes, it is outperformed by both SING and VDP in this experiment.

**Place cell model** Next, we simulate a dataset inspired by *place cells* in the hippocampus [53]. Place cells fire selectively when an animal occupies specific locations in space, and are often modeled using location-dependent tuning curves. We simulate 30 trials of latent states from a Van der Pol oscillator to represent animal trajectories through 2D space. We then construct tuning curves for $N = 8$ neurons using radial basis functions with centers placed along the latent trajectories (Figure 2E). Finally, we simulate spike counts for neuron $n$ in time bin $\tau_i$ as $y_n(\tau_i) \sim \mathrm{Pois}(f_n(\boldsymbol{x}(\tau_i)))$ (Figure 2F).

This setup is particularly challenging due to the non-conjugacy between the nonlinear dynamics and Poisson observations. Nevertheless, SING accurately infers the underlying latent states (Figure 2G). We compare SING to several baselines, including VDP and Adam on the natural parameters as in the previous experiment (Figure 2H). Since KS does not apply in this non-conjugate setting, we instead use the discrete-time conditional moments Gaussian smoother (CMGS) [54] as a third baseline, which assumes linear Gaussian approximations of the dynamics and observation models. Altogether, we find that SING outperforms all baselines in both convergence speed and final accuracy. In contrast, VDP requires careful tuning of its learning rate for stability and CMGS relies on linear approximations that incur error in highly nonlinear models like this one. These results highlight SING's robustness in settings with nonlinear dynamics and observation models, while existing methods struggle.

**Embedded Lorenz attractor** We next demonstrate that SING can perform accurate inference in high-dimensional latent dynamical systems. To do so, we embed the Lorenz attractor into latent spaces of dimension $D = 3, 5, 10, 20, 50$ by sampling the first three coordinates according an SDE with Lorenz drift and the

$D = 50$

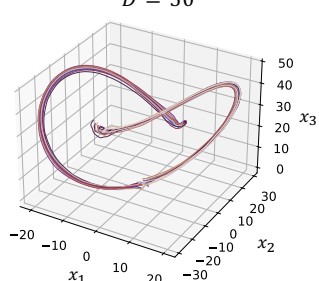

| D | latents RMSE | |
| | Monte Carlo | quadrature |
|---|---|---|
| 3 | 0.1419 | 0.1417 |
| 5 | 0.1870 | 0.1869 |
| 10 | 0.2726 | ✗ |
| 20 | 0.3978 | ✗ |
| 50 | 0.6778 | ✗ |

Figure 4: **(Top)** Recovered latent trajectories in the first 3 dimensions of a 50-dimensional embedded Lorenz attractor. **(Bottom)** Comparison of latents RMSE between Monte Carlo- and quadrature-based approximations of expectations. Monte Carlo results are averaged over 5 random seeds, with negligible standard errors (omitted).

remaining dimensions according to a random walk. For each $D$, we simulate 30 trials and generate 100-dimensional Gaussian observations. We fit SING for 1000 iterations using a single Monte Carlo sample per expectation, while the quadrature baseline uses 6 nodes per latent dimension (Section 3.3). Full experimental details can be found in Appendix L.1.3.

Figure 4 (top) shows that SING accurately recovers the underlying attractor even in a 50-dimensional latent space. The table in Figure 4 (bottom) further shows that Monte Carlo-based expectations match the accuracy of quadrature for low latent dimensions ($D \leq 5$), while remaining tractable as $D$

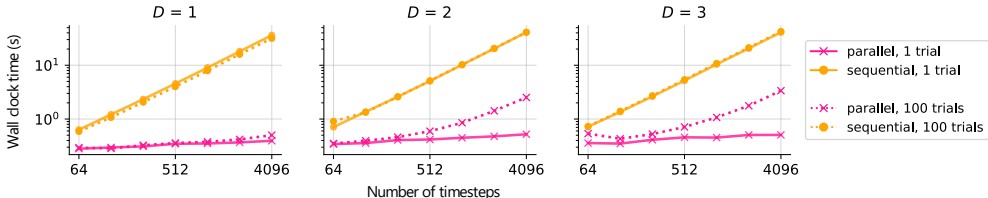

Figure 5: Runtime comparisons between parallelized SING and its sequential counterpart.

increases. This demonstrates that Monte Carlo enables scalable and memory-efficient inference and does not sacrifice accuracy in the low-dimensional regime where both methods are feasible.

## 5.2 Drift estimation on synthetic data

Next, we demonstrate that the fast, stable latent state inference enabled by SING results in improvements to drift estimation and parameter learning. We will seek to learn the drift parameters $\boldsymbol{\theta}$ together with the output parameters $\{\boldsymbol{C}, \boldsymbol{d}\}$. All experimental details can be found in Appendix L.3.

Our synthetic data example consists of a 2D latent nonlinear dynamical system evolving according to the Duffing equation, which has two stable fixed points at $(\pm\sqrt{2}, 0)$ in addition to an unstable fixed point at the origin (Figure 3A). We sample four trials starting near the rightmost fixed point with 10D Gaussian observations at 30% of the grid points. This example is challenging insofar as observations are sparse and the true latent trajectories do not cover the entire latent space.

We consider three classes of prior drift functions for our experiments: (i) a Gaussian process drift with radial basis function (RBF) kernel (Section 3.5); (ii) a neural network drift; and (iii) a linear combination of polynomial basis functions up to order three. We perform 50 variational EM iterations, alternating between performing inference and learning the hyperparameters $\boldsymbol{\Theta}$. During the learning step, we optimize the drift parameters $\boldsymbol{\theta}$ using Adam on the ELBO. The updates for the output parameters $\{\boldsymbol{C}, \boldsymbol{d}\}$ can be performed in closed form (see Appendix L.1).

Our results demonstrate that SING facilitates accurate and efficient parameter learning for diverse classes of prior drift. Although the GP prior constitutes a nonparametric family of drift functions, it performs comparably to flexible parametric drift classes, both in terms of latents RMSE (Figure 3C) and RMSE between the learned and true dynamics (Figure 3D). Unlike the parametric drift classes, though, SING-GP quantifies the uncertainty in the inferred dynamics about the leftmost fixed point (Figure 3B). Whereas SING recovers the true latents and dynamics in fewer than 25 iterations, VDP results in worse performance on both metrics in 50 vEM iterations. In Figure 3E, we observe that for both SING and VDP, decreasing $\Delta t$ results in more faithful approximation to the continuous-time ELBO $\mathcal{L}_{\text{cont}}$, and hence more accurate recovery of the dynamics.

## 5.3 Runtime comparisons

Here, we demonstrate the computational speedups enabled by using associative scans to parallelize the natural to mean parameter conversion in SING (Section 3.4). Recall that while the standard sequential method of performing this conversion takes $\mathcal{O}(D^3 T)$, our parallelized version takes $\mathcal{O}(D^3 \log T)$. In Figure 5, we compare the wall clock time of these two approaches for inference in a LDS with Gaussian observations. For different values of $D$ and $T$, we fit SING for 20 iterations over 5 randomly sampled datasets from this model and report the average runtime on an NVIDIA A100 GPU.

On single trials, parallel SING achieves nearly constant runtime scaling with sequence length, and is roughly $100\times$ faster than sequential SING for $T = 4096$. This improvement holds across all tested values of $D$. Even for batches of 100 trials, where parallel resource constraints become more significant, parallel SING maintains favorable scaling compared to the sequential baseline. We provide experiment details and additional results over different batch sizes in Appendix L.4.

## 5.4 Application to modeling neural dynamics during aggression

Finally, we apply SING-GP to the challenging task of modeling latent neural dynamics of aggressive behavior in freely moving mice. Prior work identified approximate line attractors governing neural dynamics in this setting [55, 56]. More recently, Hu et al. [14] fit GP-SDE models (see Section 3.5) with a smooth, piecewise linear kernel to uncover such dynamics. However, they relied on VDP-GP for inference, which we demonstrated can be slow to converge and may do so to suboptimal solutions.

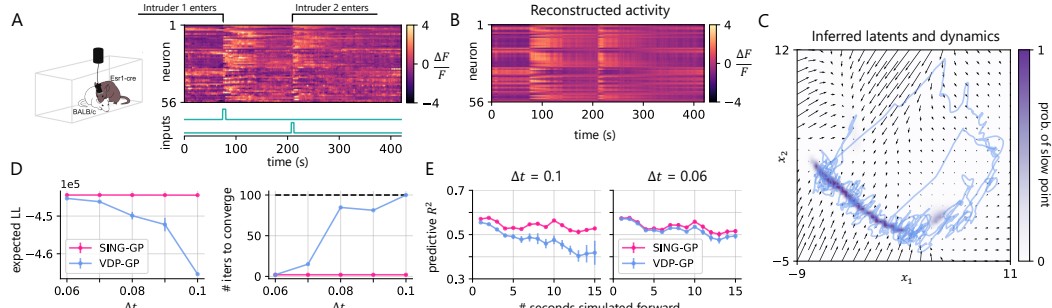

Figure 6: Results on modeling neural dynamics during aggression. **A**: Experimental setup, adapted from Vinograd et al. [55]. The data consists of calcium imaging from a mouse interacting with two intruders. We model intruder effects using step function inputs. **B**: Reconstructed neural activity from a SING-GP model fit. **C**: Inferred neural latents and dynamics from SING-GP, with high-probability slow point regions shown in purple. **D**: Comparison between SING-GP and VDP-GP on final expected log-likelihood and convergence speed for different discretizations. Error bars are $\pm$ 2SE over 5 initializations of $\Theta$. **E**: Comparison of a forward simulation $R^2$ metric (see Appendix L.5 for details) to assess the goodness of fit for latents and dynamics.

We revisit these analyses using a dataset from Vinograd et al. [55], which consists of calcium imaging of ventromedial hypothalamus neurons from a mouse interacting with two intruders (Figure 6A). We model this data using GP-SDEs that incorporate external inputs to capture intruder effects (see Appendix K.5 for more details), and we choose the same kernel from Hu et al. [14] to encode interpretable structure into the drift. We fit these models both using SING-GP and VDP-GP for 100 variational EM iterations. Full experimental details can be found in Appendix L.5.

Our SING-GP model captures variance in observed neural activity (Figure 6B) and uncovers low-dimensional representations consistent with prior findings (Figure 6C). Moreover, the GP-SDE enables posterior inference over slow points; we use this to verify that SING-GP indeed finds an approximate line attractor (Figure 6C, purple). In Figure 6D, we assess the effect of discretization in each method on final expected log-likelihood and convergence speed. SING-GP is robust to $\Delta t$ in both metrics, while VDP-GP's performance suffers for large $\Delta t$ values. This highlights a key advantage of the SING framework: robustness to $\Delta t$ enables accurate, memory-efficient inference at coarse discretizations. Finally, we evaluate the quality of model fit via a forward simulation metric (Figure 6E). While SING-GP maintains high predictive $R^2$ for trajectories simulated up to 15 seconds forward, VDP-GP's predictive performance degrades more rapidly and is again sensitive to the choice of $\Delta t$.

## 6   Discussion

We introduced SING, a method for approximate posterior inference in latent SDE models that harnesses the exponential family structure of the variational posterior to achieve rapid, stable convergence. By deriving tractable NGVI updates and parallelizing the natural-to-mean parameter conversion via associative scans, SING scales to long time series and handles nonlinear dynamics along with complex observation models with ease. Although we optimize a discrete-time ELBO, our theoretical bound (Theorem 1) ensures that SING closely approximates the continuous-time objective. We show via extensive experiments, including an application to real neural data, that SING outperforms prior methods in inference accuracy, drift estimation, and robustness to discretization.

However, SING makes simplifying assumptions that suggest several directions for future work. First, rather than fixing a discretization $\tau$ of $[0, T_{\max}]$ a priori, one could consider adaptively learning locations at which to discretize inference in order to better approximate rapidly changing dynamics. Second, future work should explore incorporating learnable locations of inducing points in SING-GP to allow for more flexible and scale-aware GP priors on the drift. Nevertheless, our results already demonstrate SING's effectiveness in a range of challenging practical settings. Altogether, SING is positioned to be a reliable and broadly applicable tool for facilitating scientific discovery in complex dynamical systems.

## Acknowledgments and Disclosure of Funding

This work was supported by grants from the NIH BRAIN Initiative (U19NS113201, R01NS131987, R01NS113119, & RF1MH133778), the NSF/NIH CRCNS Program (R01NS130789), and the Sloan, Simons, and McKnight Foundations. Henry Smith is supported by the NSF Graduate Research Fellowship (DGE-2146755) and the Knight-Hennessy graduate fellowship. We thank Aditi Jha, Etaash Katiyar, Hannah Lee, and other members of the Linderman Lab for helpful feedback throughout this project. We also thank Adi Nair, Amit Vinograd, and David Anderson for their assistance with our analysis of the hypothalamic data [55]. The authors have no competing interests to declare.

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

# Appendix

## A  Technical assumptions

Throughout our paper, we consider probability measures defined on the measurable space $(C[0, T_{\max}]^D, \mathcal{B}_{C[0,T_{\max}]^D})$, where $C[0, T_{\max}]^D$ is the Banach space of continuous functions endowed with the supremum norm.

In order to ensure the SDEs eq. (1) and eq. (7) admit unique, strong solutions, we will assume $\boldsymbol{f}$ and $\boldsymbol{f}_q$ satisfy the standard Lipschitz and linear growth conditions:

**Assumption (Drift Regularity):** There exist constants $C_1, C_2 \geq 0$ such that for all $\boldsymbol{x}, \boldsymbol{y} \in \mathbb{R}^D$ and $t, s \in [0, T_{\max}]$:

$$\|\boldsymbol{f}(\boldsymbol{x}, t) - \boldsymbol{f}(\boldsymbol{y}, s)\| \ \leq \ C_1\big(\|\boldsymbol{x} - \boldsymbol{y}\| + |t - s|\big), \quad \|\boldsymbol{f}(\boldsymbol{x}, t)\| \ \leq \ C_2\big(1 + \|\boldsymbol{x}\|\big),$$

and the same bounds hold for $\boldsymbol{f}_q$.

In addition, we will assume all SDEs are sufficiently regular such that Girsanov's Theorem (Oksendal [1], Theorem 8.6.4) can be applied. See, for example, Novikov's criterion. For the sake of completeness, we restate this result below:

**Theorem 2 (Girsanov's Theorem for SDEs):**  Suppose the SDEs

$$\nu_1 : d\boldsymbol{x}(t) = b_1(\boldsymbol{x}(t), t)dt + \boldsymbol{\Sigma}^{1/2}d\boldsymbol{w}(t), \quad 0 \leq t \leq T_{\max} \tag{13}$$

$$\nu_2 : d\boldsymbol{x}(t) = (b_1(\boldsymbol{x}(t), t) + b_2(\boldsymbol{x}(t), t))dt + \boldsymbol{\Sigma}^{1/2}d\boldsymbol{w}(t), \quad 0 \leq t \leq T_{\max} \tag{14}$$

have the same initial law $\nu_1(\boldsymbol{x}(0)) = \nu_2(\boldsymbol{x}(0))$ and admit unique, strong solutions. Then $\nu_2$ is absolutely continuous with respect to $\nu_1$ and the Radon-Nikodym density is

$$\mathcal{Z}_{T_{\max}}(\nu_1 || \nu_2) := \exp\left\{ \int_0^{T_{\max}} \langle \boldsymbol{\Sigma}^{-1/2}b_2(\boldsymbol{x}(t), t), d\boldsymbol{w}^{\nu_1}(t)\rangle - \frac{1}{2}\int_0^{T_{\max}} \|\boldsymbol{\Sigma}^{-1/2}b_2(\boldsymbol{x}(t), t)\|^2 dt \right\}.$$

where $(\boldsymbol{w}^{\nu_1}(t))_{0 \leq t \leq T_{\max}}$ is a $\nu_1$-Brownian motion. In particular, for any bounded functional $\Phi$ defined on $C[0, T_{\max}]^D$,

$$\mathbb{E}_{\nu_2}[\Phi(\boldsymbol{x})] = \mathbb{E}_{\nu_1}\left[\Phi(\boldsymbol{x})\mathcal{Z}_{T_{\max}}(\nu_1 || \nu_2)\right].$$

In particular, Theorem 2 implies that the KL divergence between eq. (1) and eq. (7) is given by

$$\mathrm{D}_{\mathrm{KL}}\left(q \| p\right) = \mathbb{E}_q\left[\log\left(\frac{q(\boldsymbol{x}(0))}{p(\boldsymbol{x}(0))}\mathcal{Z}_{T_{\max}}(p || q)\right)\right]$$

$$= \mathrm{D}_{\mathrm{KL}}\left(q(\boldsymbol{x}(0)) \| p(\boldsymbol{x}(0))\right) + \frac{1}{2}\int_0^{T_{\max}} \mathbb{E}_q\|\boldsymbol{f}(\boldsymbol{x}(t), t) - \boldsymbol{f}_q(\boldsymbol{x}(t), t)\|_{\boldsymbol{\Sigma}^{-1}} dt.$$

Here, and throughout the appendix, we adopt the notation $\|\boldsymbol{z}\|_{\boldsymbol{M}} := \boldsymbol{z}^\top \boldsymbol{M}\boldsymbol{z}$ for $\boldsymbol{M} \in \mathbb{R}^{p \times p}$ symmetric, positive semi-definite.

## B  Exponential families and natural gradients

### B.1  Overview of exponential families

Consider the collection of probability densities indexed by $\boldsymbol{\eta} \in \mathbb{R}^p$

$$\bar{q}(\boldsymbol{z}|\boldsymbol{\eta}) = h(\boldsymbol{z})\exp(\langle \boldsymbol{\eta}, \boldsymbol{T}(\boldsymbol{z})\rangle - A(\boldsymbol{\eta})), \quad A(\boldsymbol{\eta}) = \log\left(\int h(\boldsymbol{z})\exp(\langle \boldsymbol{\eta}, \boldsymbol{T}(\boldsymbol{z})\rangle)\nu(d\boldsymbol{z})\right)$$

with respect to $\nu$, which we consider to be either (i) $p$-dimensional volume (i.e., Lebesgue measure) or (ii) counting tuples of $p$ integers (i.e., counting measure).[2] $\bar{q}(\boldsymbol{z}|\boldsymbol{\eta})$ is a well-defined probability density whenever $A(\boldsymbol{\eta}) < \infty$. The set $\Xi = \{\boldsymbol{\eta} \in \mathbb{R}^p : A(\boldsymbol{\eta}) < \infty\}$ is the natural parameter space of the exponential family, and $\boldsymbol{\eta}$ are its natural parameters. We assume $\Xi$ is an open set, in which case the exponential family is said to be "regular". We call $h(\boldsymbol{z})$ the base measure of the

---

[2]Exponential families can be defined for an arbitrary measure $\nu$, but these are the two most common instances.

exponential family; $\boldsymbol{T}(\boldsymbol{z}) \in \mathbb{R}^p$ are its sufficient statistics; and $A(\boldsymbol{\eta})$ is its log normalizer or log partition function. Corresponding to the sufficient statistics are the mean parameters $\boldsymbol{\mu} \in \mathbb{R}^p$ of the exponential family, which are simply the expectations of the sufficient statistics under the exponential family, $\boldsymbol{\mu} := \mathbb{E}_q[\boldsymbol{T}(\boldsymbol{z})]$.

As an example, consider the $p$-dimensional multivariate normal distribution $\mathcal{N}(\boldsymbol{z}|\boldsymbol{\xi}, \boldsymbol{\Lambda})$ with unknown mean $\boldsymbol{\xi}$ and unknown, positive-definite covariance $\boldsymbol{\Lambda}$. Then we can write this family in exponential family form as follows:

$$\bar{q}(\boldsymbol{z}|\boldsymbol{\xi}, \boldsymbol{\Lambda}) = (2\pi)^{-p/2}|\boldsymbol{\Lambda}|^{-1/2} \exp\left(-\frac{1}{2}\|\boldsymbol{z} - \boldsymbol{\xi}\|^2_{\boldsymbol{\Lambda}^{-1}}\right)$$

$$= (2\pi)^{-p/2} \exp\left(-\frac{1}{2}\mathrm{tr}\left(\boldsymbol{\Lambda}^{-1}\boldsymbol{z}\boldsymbol{z}^\top\right) + \boldsymbol{z}^\top\boldsymbol{\Lambda}^{-1}\boldsymbol{\xi} - \frac{1}{2}\{\log|\boldsymbol{\Lambda}| + \boldsymbol{\xi}^\top\boldsymbol{\Lambda}^{-1}\boldsymbol{\xi}\}\right).$$

Let us identify $\boldsymbol{\eta} = (\boldsymbol{\eta}_1, \boldsymbol{\eta}_2) = (\mathrm{vec}(\boldsymbol{\Lambda}^{-1}), \boldsymbol{\Lambda}^{-1}\boldsymbol{\xi})$ as well as $h(\boldsymbol{z}) = (2\pi)^{-p/2}$, where $\mathrm{vec}(\boldsymbol{M}), \boldsymbol{M} \in \mathbb{R}^{p\times p}$ represents the $p^2$-dimensional vector representation of the matrix $\boldsymbol{M}$. The natural parameter space is $\boldsymbol{\Xi} = \{\mathrm{vec}(\boldsymbol{M}) : \boldsymbol{M} \in \mathbb{R}^{p\times p} \text{ symmetric, positive-definite}\} \times \mathbb{R}^p$. The sufficient statistics of the exponential family are $\boldsymbol{T}(\boldsymbol{z}) = (-\frac{1}{2}\mathrm{vec}(\boldsymbol{z}\boldsymbol{z}^\top), \boldsymbol{z})$ and the mean parameters are $\boldsymbol{\mu} = (-\frac{1}{2}\mathrm{vec}(\boldsymbol{\Lambda} + \boldsymbol{\xi}\boldsymbol{\xi}^\top), \boldsymbol{\xi})$. It remains to compute the log normalizer $A(\boldsymbol{\eta})$. In particular,

$$\log|\boldsymbol{\Lambda}| = -\log|\boldsymbol{\Lambda}^{-1}| = -\log|\boldsymbol{\eta}_1|$$

$$\boldsymbol{\xi}^\top\boldsymbol{\Lambda}^{-1}\boldsymbol{\xi} = \boldsymbol{\eta}_2^\top(\boldsymbol{\eta}_1)^{-1}\boldsymbol{\eta}_2$$

which implies $A(\boldsymbol{\eta}) = \frac{1}{2}(\boldsymbol{\eta}_2^\top(\boldsymbol{\eta}_1)^{-1}\boldsymbol{\eta}_2 - \log|\boldsymbol{\eta}_1|)$. Here we slightly abuse notation by treating $\boldsymbol{\eta}_1$ interchangeably as a matrix and a vector.

Our definition of the exponential family $\bar{q}(\boldsymbol{z}|\boldsymbol{\eta})$ leads us to two important results regarding the log normalizer.

**Lemma 1:** The log normalizer $A(\boldsymbol{\eta})$ is infinitely differentiable on the natural parameter space $\boldsymbol{\Xi}$. Moreover, the log normalizer satisfies

$$\nabla_{\boldsymbol{\eta}}A(\boldsymbol{\eta}) = \mathbb{E}_{\bar{q}}[\boldsymbol{T}(\boldsymbol{z})] = \boldsymbol{\mu}, \quad \nabla_{\boldsymbol{\eta}}^2 A(\boldsymbol{\eta}) = \mathrm{Cov}_{\bar{q}}[\boldsymbol{T}(\boldsymbol{z})]. \tag{15}$$

*Proof.* A proof of the first statement can be found in Lehmann and Romano [57, Theorem 2.7.1]. For the second, differentiate the log normalizer twice

$$\nabla_{\boldsymbol{\eta}}A(\boldsymbol{\eta}) = \nabla_{\boldsymbol{\eta}}\log\left(\int h(\boldsymbol{z})\exp(\langle\boldsymbol{\eta}, \boldsymbol{T}(\boldsymbol{z})\rangle)\nu(d\boldsymbol{z})\right)$$

$$= \frac{\int \boldsymbol{T}(\boldsymbol{z})h(\boldsymbol{z})\exp(\langle\boldsymbol{\eta}, \boldsymbol{T}(\boldsymbol{z})\rangle)\nu(d\boldsymbol{z})}{\int h(\boldsymbol{z})\exp(\langle\boldsymbol{\eta}, \boldsymbol{T}(\boldsymbol{z})\rangle)\nu(d\boldsymbol{z})}$$

$$= \int \boldsymbol{T}(\boldsymbol{z})h(\boldsymbol{z})\exp(\langle\boldsymbol{\eta}, \boldsymbol{T}(\boldsymbol{z})\rangle - A(\boldsymbol{\eta}))\nu(d\boldsymbol{z})$$

$$= \mathbb{E}_{\bar{q}}[\boldsymbol{T}(\boldsymbol{z})]$$

$$\nabla_{\boldsymbol{\eta}}^2 A(\boldsymbol{\eta}) = \int \nabla_{\boldsymbol{\eta}}\left\{\boldsymbol{T}(\boldsymbol{z})h(\boldsymbol{z})\exp(\langle\boldsymbol{\eta}, \boldsymbol{T}(\boldsymbol{z})\rangle - A(\boldsymbol{\eta}))\right\}\nu(d\boldsymbol{z})$$

$$= \mathbb{E}_{\bar{q}}[\boldsymbol{T}(\boldsymbol{z})\boldsymbol{T}(\boldsymbol{z})^\top] - (\mathbb{E}_{\bar{q}}[\boldsymbol{T}(\boldsymbol{z})])(\mathbb{E}_{\bar{q}}[\boldsymbol{T}(\boldsymbol{z})])^\top$$

$$= \mathrm{Cov}_{\bar{q}}[\boldsymbol{T}(\boldsymbol{z})].$$

$\square$

As an immediate consequence of the identities in eq. (15), we can give an equivalent expression for the Fisher information matrix of the exponential family $\bar{q}(\boldsymbol{z}|\boldsymbol{\eta})$.

**Lemma 2:** Consider $\bar{q}(\boldsymbol{z}|\boldsymbol{\eta})$ constituting an exponential family. And let $\frac{d\boldsymbol{\mu}(\boldsymbol{\eta})}{d\boldsymbol{\eta}} = \nabla_{\boldsymbol{\eta}}^2 A(\boldsymbol{\eta})$ be the Jacobian of the mapping from natural to mean parameters. Then the following are equivalent:

$$\nabla_{\boldsymbol{\eta}}^2 A(\boldsymbol{\eta}) = \mathrm{Cov}_{\bar{q}(\boldsymbol{z}|\boldsymbol{\eta})}[\boldsymbol{T}(\boldsymbol{z})] = \mathcal{F}(\boldsymbol{\eta}),$$

where $\mathcal{F}(\boldsymbol{\eta})$ denotes the Fisher information of the exponential family.

*Proof.* The first identity is exactly the second statement in Lemma 1. For the second, recall the identity

$$\mathcal{F}(\boldsymbol{\eta}) = \mathrm{Cov}_{\bar{q}(\boldsymbol{z}|\boldsymbol{\eta})}[\nabla_{\boldsymbol{\eta}} \log \bar{q}(\boldsymbol{z}|\boldsymbol{\eta})]$$

for the Fisher information. Plugging in the density of the exponential family, we get

$$\mathcal{F}(\boldsymbol{\eta}) = \mathrm{Cov}_{\bar{q}(\boldsymbol{z}|\boldsymbol{\eta})}[\nabla_{\boldsymbol{\eta}} \langle \boldsymbol{\eta}, \boldsymbol{T}(\boldsymbol{z}) \rangle] = \mathrm{Cov}_{\bar{q}(\boldsymbol{z}|\boldsymbol{\eta})}[\boldsymbol{T}(\boldsymbol{z})].$$

$\square$

Define the set of mean parameters attainable by some probability density with respect to $\nu$

$$\mathcal{M} := \left\{ \boldsymbol{\mu} \in \mathbb{R}^p \mid \boldsymbol{\mu} = \mathbb{E}_{\tilde{q}(\boldsymbol{z})}[\boldsymbol{T}(\boldsymbol{z})], \ \tilde{q} \text{ a probability density wrt } \nu \right\}.$$

Note that, in this definition, $\tilde{q}(\boldsymbol{z})$ does not necessarily belong to the exponential family $\bar{q}(\boldsymbol{z}|\boldsymbol{\eta})$. Proposition 1 demonstrates that $\nabla_{\boldsymbol{\eta}} A(\boldsymbol{\eta})$ maps from the interior of the natural parameter space $\Xi$ to the mean parameter space $\mathcal{M}$. In fact, this mapping is surjective onto $\mathrm{relint}(\mathcal{M})$, the relative interior of $\mathcal{M}$ (see Wainright et al. [58, Theorem 3.3]). Recall that for a convex set $C$,

$$\mathit{relint}(C) = \{c \in C \mid \text{for every } c' \neq c, \text{ there exists } d \in C, \lambda \in (0,1) \text{ such that } c = \lambda c' + (1-\lambda)d\}.$$

A question of considerable interest is when, in addition, $\nabla_{\boldsymbol{\eta}} A(\boldsymbol{\eta})$ will be one-to-one. In other words, when can each mean parameter $\boldsymbol{\mu}$ be identified with a unique member of the exponential family.

An exponential family is said to be minimal if its sufficient statistics do not satisfy any linear constraints. If a $p$-dimensional exponential family is not minimal, it can be reduced to a $(p-1)$-dimensional exponential family by rewriting one sufficient statistic as a linear combination of the others. In the multivariate normal family example, the exponential family as written is not minimal, since $\boldsymbol{z}\boldsymbol{z}^\top$ is symmetric, and hence there are duplicate statistics along the off-diagonal. To reparameterize this exponential family in minimal form, one can define the symmetric vectorization operator $\mathrm{svec}(\boldsymbol{M}) := (\boldsymbol{M}_{11}, \sqrt{2}\boldsymbol{M}_{21}, \boldsymbol{M}_{22}, \sqrt{2}\boldsymbol{M}_{31}, \sqrt{2}\boldsymbol{M}_{32}, \boldsymbol{M}_{33}, \ldots, \boldsymbol{M}_{pp}) \in \mathbb{R}^{p(p+1)/2}$ for $\boldsymbol{M} \in \mathbb{R}^{p \times p}$ symmetric and then rewrite the multivariate normal density as

$$\bar{q}(\boldsymbol{z}|\boldsymbol{\xi}, \boldsymbol{\Lambda}) = (2\pi)^{-p/2} \exp\left( -\frac{1}{2}\mathrm{svec}(\boldsymbol{\Lambda}^{-1})^\top \mathrm{svec}(\boldsymbol{z}\boldsymbol{z}^\top) + \boldsymbol{z}^\top \boldsymbol{\Lambda}^{-1}\boldsymbol{\xi} - \frac{1}{2}\{\log|\boldsymbol{\Lambda}| + \boldsymbol{\xi}^\top \boldsymbol{\Lambda}^{-1}\boldsymbol{\xi}\} \right).$$

The natural parameters of the minimal exponential family are $\boldsymbol{\eta} = (\boldsymbol{\eta}_1, \boldsymbol{\eta}_2) = (\mathrm{svec}(\boldsymbol{\Lambda}^{-1}), \boldsymbol{\Lambda}^{-1}\boldsymbol{\xi})$. Note that in minimal exponential family form, the natural parameter space has dimension $p(p+1)/2 + p$, whereas in the non-minimal form it has dimension $p^2 + p$.

It is a classical result (see Wainright et al. [58, Proposition 3.2]) that when $\bar{q}(\boldsymbol{z}|\boldsymbol{\eta})$ constitutes a minimal exponential family, then $\nabla_{\boldsymbol{\eta}} A(\boldsymbol{\eta})$ is also one-to-one. Moreover, the mean parameter space $\mathcal{M}$ is full-dimensional, meaning $\mathrm{int}(\mathcal{M}) = \mathrm{relint}(\mathcal{M})$. Observe that since $\nabla_{\boldsymbol{\eta}} A(\boldsymbol{\eta})$ is then both surjective and one-to-one, the inverse mapping $[\nabla A]^{-1} : \mathrm{int}(\mathcal{M}) \to \Xi$ is well-defined. As a consequence of the Inverse Function Theorem, we have that for minimal exponential families, the inverse mapping $[\nabla A]^{-1}(\boldsymbol{\mu})$ has Jacobian $[\mathrm{Cov}_{\bar{q}(\boldsymbol{z}|\boldsymbol{\eta}(\boldsymbol{\mu}))}[\boldsymbol{T}(\boldsymbol{z})]]^{-1} = F(\boldsymbol{\eta}(\boldsymbol{\mu}))^{-1}$.

Lastly, we give an expression for the KL divergence in an exponential family.

**Lemma 3:** Suppose $\bar{q}(\boldsymbol{z}|\boldsymbol{\eta}_1)$ and $\bar{q}(\boldsymbol{z}|\boldsymbol{\eta}_2)$ belong to the same $p$-dimensional exponential family with natural parameters $\boldsymbol{\eta}_1$ and $\boldsymbol{\eta}_2$, respectively. Then

$$\mathrm{D}_{\mathrm{KL}}\left(\bar{q}(\boldsymbol{z}|\boldsymbol{\eta}_1) \parallel \bar{q}(\boldsymbol{z}|\boldsymbol{\eta}_2)\right) = \langle \boldsymbol{\eta}_1 - \boldsymbol{\eta}_2, \boldsymbol{\mu}_1 \rangle - A(\boldsymbol{\eta}_1) + A(\boldsymbol{\eta}_2).$$

*Proof.* Write out the difference between the log densities of $\bar{q}(\boldsymbol{z}|\boldsymbol{\eta}_1)$ and $\bar{q}(\boldsymbol{z}|\boldsymbol{\eta}_2)$. Then recognize that the contribution of the base measure $h(\boldsymbol{z})$ cancels and that $\mathbb{E}_{q(\boldsymbol{z}|\boldsymbol{\eta}_1)} \boldsymbol{T}(\boldsymbol{z}) = \boldsymbol{\mu}_1$. $\square$

Recall that in Section 2.2, we motivated natural gradient VI by claiming

$$\mathrm{D}_{\mathrm{KL}}\left(\bar{q}(\boldsymbol{z}|\boldsymbol{\eta}_1) \parallel \bar{q}(\boldsymbol{z}|\boldsymbol{\eta}_2)\right) \approx \frac{1}{2}(\boldsymbol{\eta}_2 - \boldsymbol{\eta}_1)^\top \mathcal{F}(\boldsymbol{\eta}_1)(\boldsymbol{\eta}_2 - \boldsymbol{\eta}_1).$$

We are now equipped to justify this claim by Taylor expanding $A(\boldsymbol{\eta}_2)$ about $\boldsymbol{\eta}_1$ and invoking Lemma 3:

$$A(\boldsymbol{\eta}_2) = A(\boldsymbol{\eta}_1) + \langle \boldsymbol{\eta}_2 - \boldsymbol{\eta}_1, \boldsymbol{\mu}_1 \rangle + \frac{1}{2}(\boldsymbol{\eta}_2 - \boldsymbol{\eta}_1)^\top \mathcal{F}(\boldsymbol{\eta}_1)(\boldsymbol{\eta}_2 - \boldsymbol{\eta}_1) + \mathcal{O}(\|\boldsymbol{\eta}_2 - \boldsymbol{\eta}_1\|^3)$$

$$\implies \underbrace{\langle \boldsymbol{\eta}_1 - \boldsymbol{\eta}_2, \boldsymbol{\mu}_1 \rangle - A(\boldsymbol{\eta}_1) + A(\boldsymbol{\eta}_2)}_{=\mathrm{D}_{\mathrm{KL}}(\bar{q}(\boldsymbol{z}|\boldsymbol{\eta}_1) \,\|\, \bar{q}(\boldsymbol{z}|\boldsymbol{\eta}_2))} = \frac{1}{2}(\boldsymbol{\eta}_2 - \boldsymbol{\eta}_1)^\top \mathcal{F}(\boldsymbol{\eta}_1)(\boldsymbol{\eta}_2 - \boldsymbol{\eta}_1) + \mathcal{O}(\|\boldsymbol{\eta}_2 - \boldsymbol{\eta}_1\|^3).$$

In the first line, we used the result $\nabla_{\boldsymbol{\eta}} A(\boldsymbol{\eta}_1) = \boldsymbol{\mu}_1$ from Lemma 1 as well as the result $\nabla_{\boldsymbol{\eta}}^2 A(\boldsymbol{\eta}_1) = \mathcal{F}(\boldsymbol{\eta}_1)$ from Lemma 2.

## B.2  Natural gradients and exponential families

Next, we discuss why the natural gradient VI update eq. (4) simplifies to a gradient update of the ELBO $\mathcal{L}(\boldsymbol{\eta})$ with respect to the mean parameters $\boldsymbol{\mu}$, rather than the natural parameters $\boldsymbol{\eta}$, when the variational posterior constitutes a minimal exponential family.

Recall eq. (4) that the natural gradient VI update is

$$\boldsymbol{\eta}^{(j+1)} = \boldsymbol{\eta}^{(j)} + \rho[\mathcal{F}(\boldsymbol{\eta}^{(j)})]^{-1}\nabla_{\boldsymbol{\eta}}\mathcal{L}(\boldsymbol{\eta}^{(j)}).$$

From Lemma 2, we have that in a minimal exponential family the Jacobian of the change-of-variable from natural to mean parameters is the Fisher information matrix $\mathcal{F}(\boldsymbol{\eta})$. This implies, by the Inverse Function Theorem, that the inverse Fisher information matrix is the Jacobian of the change-of-variable from mean to natural parameters. This allows us to rewrite the NGVI update as

$$\boldsymbol{\eta}^{(j+1)} = \boldsymbol{\eta}^{(j)} + \rho[\mathcal{F}(\boldsymbol{\eta}^{(j)})]^{-1}\nabla_{\boldsymbol{\eta}}\mathcal{L}(\boldsymbol{\eta}^{(j)})$$

$$= \boldsymbol{\eta}^{(j)} + \rho\left(\frac{d\boldsymbol{\eta}(\boldsymbol{\mu})}{d\boldsymbol{\mu}}(\boldsymbol{\mu}^{(j)})\right)\nabla_{\boldsymbol{\eta}}\mathcal{L}(\boldsymbol{\eta}^{(j)})$$

$$= \boldsymbol{\eta}^{(j)} + \rho\nabla_{\boldsymbol{\mu}}\mathcal{L}(\boldsymbol{\eta}^{(j)}).$$

In the final line, we used the chain rule. The equivalence between these two updates boils down to the fact that, in minimal exponential families, one can change between natural and mean parameters by multiplying by the Fisher information matrix.

Prior to concluding this section, we provide some helpful natural gradient identities that will enable us to compute our SING natural gradient VI updates.

**Lemma 4:**  Suppose $\bar{q}(\boldsymbol{z}|\boldsymbol{\eta}_1)$ and $\bar{q}(\boldsymbol{z}|\boldsymbol{\eta}_2)$ belong to the same $p$-dimensional exponential family with natural parameters $\boldsymbol{\eta}_1$ and $\boldsymbol{\eta}_2$, respectively. Then

$$\nabla_{\boldsymbol{\mu}_1}\mathrm{D}_{\mathrm{KL}}\left(\bar{q}(\boldsymbol{z}|\boldsymbol{\eta}_1) \,\|\, \bar{q}(\boldsymbol{z}|\boldsymbol{\eta}_2)\right) = \boldsymbol{\eta}_1 - \boldsymbol{\eta}_2$$
$$\nabla_{\boldsymbol{\mu}_1}\mathbb{E}_{\bar{q}(\boldsymbol{z}|\boldsymbol{\eta}_1)}\log\bar{q}(\boldsymbol{z}|\boldsymbol{\eta}_1) = \boldsymbol{\eta}_1.$$

*Proof.*  Differentiating the expression for the KL divergence Lemma 3, we obtain

$$\nabla_{\boldsymbol{\mu}_1}\mathrm{D}_{\mathrm{KL}}\left(\bar{q}(\boldsymbol{z}|\boldsymbol{\eta}_1) \,\|\, \bar{q}(\boldsymbol{z}|\boldsymbol{\eta}_2)\right) = \nabla_{\boldsymbol{\mu}_1}\left(\langle\boldsymbol{\eta}_1 - \boldsymbol{\eta}_2, \boldsymbol{\mu}_1\rangle - A(\boldsymbol{\eta}_1) + A(\boldsymbol{\eta}_2)\right)$$

$$= \left(\frac{d\boldsymbol{\eta}(\boldsymbol{\mu})}{d\boldsymbol{\mu}}(\boldsymbol{\mu}_1)\right)\boldsymbol{\mu}_1 + (\boldsymbol{\eta}_1 - \boldsymbol{\eta}_2) - \left(\frac{d\boldsymbol{\eta}(\boldsymbol{\mu})}{d\boldsymbol{\mu}}(\boldsymbol{\mu}_1)\right)\underbrace{\frac{d}{d\boldsymbol{\eta}}A(\boldsymbol{\eta}_1)}_{=\boldsymbol{\mu}_1}$$

$$= \boldsymbol{\eta}_1 - \boldsymbol{\eta}_2,$$

as claimed. As for the gradient of the negative entropy of an exponential family, we have by a nearly identical argument

$$\nabla_{\boldsymbol{\mu}_1}\mathbb{E}_{\bar{q}(\boldsymbol{z}|\boldsymbol{\eta}_1)}\log\bar{q}(\boldsymbol{z}|\boldsymbol{\eta}_1) = \nabla_{\boldsymbol{\mu}_1}\left(\langle\boldsymbol{\eta}_1, \boldsymbol{\mu}_1\rangle - A(\boldsymbol{\eta}_1)\right)$$

$$= \left(\frac{d\boldsymbol{\eta}(\boldsymbol{\mu})}{d\boldsymbol{\mu}}(\boldsymbol{\mu}_1)\right)\boldsymbol{\mu}_1 + \boldsymbol{\eta}_1 - \left(\frac{d\boldsymbol{\eta}(\boldsymbol{\mu})}{d\boldsymbol{\mu}}(\boldsymbol{\mu}_1)\right)\underbrace{\frac{d}{d\boldsymbol{\eta}}A(\boldsymbol{\eta}_1)}_{=\boldsymbol{\mu}_1}$$

$$= \boldsymbol{\eta}_1.$$

$\square$

An important point is that while the natural gradient of the KL divergence and entropy admit simple, closed forms in an exponential family, the same is not true for the gradient with respect to the natural parameters. Differentiating the expression for the KL divergence from Lemma 3, we obtain

$$
\nabla_{\boldsymbol{\eta}_1} D_{KL} \left( \bar{q}(\boldsymbol{z}|\boldsymbol{\eta}_1) \parallel \bar{q}(\boldsymbol{z}|\boldsymbol{\eta}_2) \right) = \nabla_{\boldsymbol{\eta}_1} \left( \langle \boldsymbol{\eta}_1 - \boldsymbol{\eta}_2, \boldsymbol{\mu}_1 \rangle - A(\boldsymbol{\eta}_1) + A(\boldsymbol{\eta}_2) \right)
$$

$$
= \boldsymbol{\mu}_1 + \left( \frac{d\boldsymbol{\mu}(\boldsymbol{\eta})}{d\boldsymbol{\eta}}(\boldsymbol{\eta}_1) \right) (\boldsymbol{\eta}_1 - \boldsymbol{\eta}_2) - \boldsymbol{\mu}_1
$$

$$
= F(\boldsymbol{\eta}_1)(\boldsymbol{\eta}_1 - \boldsymbol{\eta}_2).
$$

Notably, this expression requires multiplication by the Fisher information matrix, which, as we pointed out in Section 3.1, is intractable in high-dimensional exponential families.

## C  ELBO derivation

We first provide a derivation of the continuous-time ELBO, $\mathcal{L}_{\text{cont}}$. Using the notation from Theorem 2, we have

$$
\log p(\boldsymbol{y}|\boldsymbol{\Theta}) = \log \int p(\boldsymbol{y}|\boldsymbol{x}, \boldsymbol{\Theta}) p(\boldsymbol{x}|\boldsymbol{\Theta}) d\boldsymbol{x}
$$

$$
\overset{(i)}{=} \log \int p(\boldsymbol{y}|\boldsymbol{x}, \boldsymbol{\Theta}) \mathcal{Z}_{T_{\max}}(q\|p) \frac{p(\boldsymbol{x}(0))}{q(\boldsymbol{x}(0))} q(\boldsymbol{x}) d\boldsymbol{x}
$$

$$
\overset{(ii)}{\geq} \int \log \left( p(\boldsymbol{y}|\boldsymbol{x}, \boldsymbol{\Theta}) \mathcal{Z}_{T_{\max}}(q\|p) \frac{p(\boldsymbol{x}(0))}{q(\boldsymbol{x}(0))} \right) q(\boldsymbol{x}) d\boldsymbol{x}
$$

$$
= \mathbb{E}_q[\log p(\boldsymbol{y}|\boldsymbol{x}, \boldsymbol{\Theta})] - D_{KL} \left( q(\boldsymbol{x}) \parallel p(\boldsymbol{x}|\boldsymbol{\Theta}) \right)
$$

$$
= \mathbb{E}_q[\log p(\boldsymbol{y}|\boldsymbol{x}, \boldsymbol{\Theta})] - D_{KL} \left( q(\boldsymbol{x}(0)) \parallel p(\boldsymbol{x}(0)) \right) - \underbrace{\frac{1}{2} \int_0^{T_{\max}} \mathbb{E}_q \| \boldsymbol{f}(\boldsymbol{x}(t)|\boldsymbol{\Theta}) - \boldsymbol{f}_q(\boldsymbol{x}(t), t) \|_{\boldsymbol{\Sigma}^{-1}}^2 dt}_{= D_{KL}(q(\boldsymbol{x}|\boldsymbol{x}(0)) \parallel p(\boldsymbol{x}|\boldsymbol{x}(0), \boldsymbol{\Theta}))}.
$$

Equality (i) follows from Girsanov's Theorem and inequality (ii) follows from Jensen's inequality. Here, the notation $p(\boldsymbol{x}|\boldsymbol{\Theta})d\boldsymbol{x}$ is adopted for readability. Formally, $p(\boldsymbol{x}|\boldsymbol{\Theta})d\boldsymbol{x}$ denotes integration with respect to the probability measure $p(\boldsymbol{x}|\boldsymbol{\Theta})$ over paths $C[0, T_{\max}]^D$.

Note that if the parameters of $p(\boldsymbol{x}(0)) = \mathcal{N}(\boldsymbol{\nu}, \boldsymbol{V})$ are learned, then during the learning step one will set $p(\boldsymbol{x}(0)) = q(\boldsymbol{x}(0))$, and the second term of the ELBO disappears.

Next, for the discrete-time ELBO, replacing the path measures $q(\boldsymbol{x})$ and $p(\boldsymbol{x})$ with $q(\boldsymbol{x}_{0:T})$ and $p(\boldsymbol{x}_{0:T})$, respectively, and following an identical argument yields the lower bound on the marginal log likelihood

$$
\log p(\boldsymbol{y}|\boldsymbol{\Theta}) \geq \mathbb{E}_q[\log p(\boldsymbol{y}|\boldsymbol{x}, \boldsymbol{\Theta})] - \underbrace{\mathbb{E}_q \left[ \log \frac{q(\boldsymbol{x}_{0:T})}{p(\boldsymbol{x}_{0:T}|\boldsymbol{\Theta})} \right]}_{D_{KL}(q(\boldsymbol{x}_{0:T}) \parallel p(\boldsymbol{x}_{0:T}|\boldsymbol{\Theta}))}.
$$

Substituting $\tilde{p}(\boldsymbol{x}_{0:T}|\boldsymbol{\Theta})$ for $p(\boldsymbol{x}_{0:T}|\boldsymbol{\Theta})$ in the denominator of the second term yields $\mathcal{L}_{\text{approx}}$.

## D  Derivation of SING updates

In this section we derive the SING updates eq. (11) as the natural gradient ascent update eq. (4) performed the approximate ELBO $\mathcal{L}_{\text{approx}}$.

Prior to computing the gradient, we first define $\boldsymbol{m}_i = \mathbb{E}_q[\boldsymbol{x}_i]$ and $\boldsymbol{S}_i = \text{Cov}_q[\boldsymbol{x}_i], 0 \leq i \leq T$ to be the marginal mean and covariance of $\boldsymbol{x}_i$ under $q$ as well as $\boldsymbol{S}_{i+1,i} = \text{Cov}_q[\boldsymbol{x}_{i+1}, \boldsymbol{x}_i], 0 \leq i \leq T - 1$ to be the pairwise covariance of $\boldsymbol{x}_{i+1}$ and $\boldsymbol{x}_i$ under $q$. We shall adopt this notation throughout the remainder of the appendix. By our definition of the mean parameters from Section 3.1, we have $\boldsymbol{\mu}_{i,1} = \boldsymbol{m}_i, \boldsymbol{\mu}_{i,2} = \boldsymbol{S}_i + \boldsymbol{m}_i \boldsymbol{m}_i^\top, \boldsymbol{\mu}_{i,3} = \boldsymbol{S}_{i+1,i} + \boldsymbol{m}_{i+1} \boldsymbol{m}_i^\top$.

Now, to compute the NGVI update step eq. (4), we first invoke the exponential family identity for the negative entropy from Appendix B.2, Lemma 4: $\nabla_{\boldsymbol{\mu}} \mathbb{E}_q \log q(\boldsymbol{x}_{0:T}) = \boldsymbol{\eta}$. Consequently, eq. (4)

simplifies to

$$
\begin{aligned}
\boldsymbol{\eta}^{(j+1)} &= \boldsymbol{\eta}^{(j)} + \rho \nabla_{\boldsymbol{\mu}} \mathcal{L}(\boldsymbol{\mu}^{(j)}) \\
&= (1 - \rho)\boldsymbol{\eta}^{(j)} + \rho \nabla_{\boldsymbol{\mu}} \mathbb{E}_q[\log \tilde{p}(\boldsymbol{x}_{0:T}|\boldsymbol{\Theta})p(\boldsymbol{y}|\boldsymbol{x}, \boldsymbol{\Theta})] \\
&= (1 - \rho)\boldsymbol{\eta}^{(j)} + \rho \nabla_{\boldsymbol{\mu}} \Bigg\{ \sum_{i=0}^{T-1} \{\mathbb{E}_q[\log \tilde{p}(\boldsymbol{x}_{i+1}|\boldsymbol{x}_i, \boldsymbol{\Theta})] + \mathbb{E}_q[\log p(\boldsymbol{y}_i|\boldsymbol{x}_i, \boldsymbol{\Theta})\delta_i]\} \\
&\quad + \mathbb{E}_q[\log \tilde{p}(\boldsymbol{x}_0|\boldsymbol{\Theta})] + \mathbb{E}_q[\log p(\boldsymbol{y}_T|\boldsymbol{x}_T, \boldsymbol{\Theta})\delta_T] \Bigg\}.
\end{aligned}
$$

From here, we consider the gradient with respect to $\boldsymbol{\mu}_{i,1}, \boldsymbol{\mu}_{i,2}, \boldsymbol{\mu}_{i,3}$ separately. Looking at our expression for $\mathbb{E}_q[\log \tilde{p}(\boldsymbol{x}_{i+1}|\boldsymbol{x}_i, \boldsymbol{\Theta})]$ in Appendix F, Proposition 1 we see that $\mathbb{E}_q[\log \tilde{p}(\boldsymbol{x}_{i+1}|\boldsymbol{x}_i, \boldsymbol{\Theta})]$ depends only on $(\boldsymbol{m}_i, \boldsymbol{m}_{i+1}, \boldsymbol{S}_i, \boldsymbol{S}_{i+1}, \boldsymbol{S}_{i+1,i})$, and hence only on $(\boldsymbol{\mu}_{i,1}, \boldsymbol{\mu}_{i+1,1}, \boldsymbol{\mu}_{i,2}, \boldsymbol{\mu}_{i+1,2}, \boldsymbol{\mu}_{i,3})$. Likewise, $\mathbb{E}_q[\log p(\boldsymbol{y}_i|\boldsymbol{x}_i, \boldsymbol{\Theta})]$ depends only on the marginal parameters of $q$ at time $\tau_i$, $(\boldsymbol{m}_i, \boldsymbol{S}_i)$, and hence only on $(\boldsymbol{\mu}_{i,1}, \boldsymbol{\mu}_{i,2})$. $\mathbb{E}_q[\log \tilde{p}(\boldsymbol{x}_0|\boldsymbol{\Theta})]$, depends only on $(\boldsymbol{m}_0, \boldsymbol{S}_0)$, and hence only on $(\boldsymbol{\mu}_{0,1}, \boldsymbol{\mu}_{0,2})$. Separating out the contributions of $\boldsymbol{\mu}_{i,1}, \boldsymbol{\mu}_{i,2}, \boldsymbol{\mu}_{i,3}$, we recover the local updates eq. (11).

In order to compute the gradients with respect to $\boldsymbol{\mu}_i = (\boldsymbol{\mu}_{i,1}, \boldsymbol{\mu}_{i,2}, \boldsymbol{\mu}_{i,3})$, we perform a local change-of-variable for each $i = 0, \ldots, T-1$. Specifically, we use the bijective map $(\boldsymbol{\mu}_i, \boldsymbol{m}_{i+1}, \boldsymbol{S}_{i+1}) \mapsto (\boldsymbol{m}_i, \boldsymbol{m}_{i+1}, \boldsymbol{S}_i, \boldsymbol{S}_{i+1}, \boldsymbol{S}_{i+1,i})$ in order to first compute gradients with respect to $(\boldsymbol{m}_i, \boldsymbol{m}_{i+1}, \boldsymbol{S}_i, \boldsymbol{S}_{i+1}, \boldsymbol{S}_{i+1,i})$ and subsequently multiply the result by the Jacobian of this map. Note that the Jacobian is of dimension $D(2 + 3D) \times D(2 + 3D)$, which importantly does not scale with the size $T$ of the grid $\boldsymbol{\tau}$.

# E  Proofs for continuous-time approximation

**Theorem 1 (Precise Statement)**: Assume $\boldsymbol{f}$ is globally Lipschitz, jointly in $(\boldsymbol{x}, t)$

$$
\|\boldsymbol{f}(\boldsymbol{x}, t) - \boldsymbol{f}(\boldsymbol{y}, s)\| \leq L_{\boldsymbol{f}}(\|\boldsymbol{x} - \boldsymbol{y}\| + |t - s|)
$$

in addition to

$$
\sup_{0 \leq t \leq T_{\max}} \|\boldsymbol{A}(t)\|, \quad \sup_{0 \leq t \leq T_{\max}} \|\boldsymbol{b}(t)\| \leq M, \quad \|\boldsymbol{A}(t) - \boldsymbol{A}(s)\|, \|\boldsymbol{b}(t) - \boldsymbol{b}(s)\| \leq M|t - s|, \ 0 \leq s \leq t \leq T_{\max}.
$$

Suppose further that $\boldsymbol{A}(t)$ is differentiable on $0 < t < T_{\max}$.

Then the error between the continuous and approximate ELBO satisfies

$$
|\mathcal{L}_{\text{cont}}(q, \boldsymbol{\Theta}) - \mathcal{L}_{\text{approx}}(q, \boldsymbol{\Theta})| \leq (\Delta t)^{1/2} \cdot C(\mathbb{E}_q\|\boldsymbol{x}(0)\|^2, T_{\max}, M, D, L_{\boldsymbol{f}}, \|\boldsymbol{\Sigma}^{-1/2}\|, \|\boldsymbol{\Sigma}^{1/2}\|)
$$

where $\Delta t = \max_{i=0}^{T-1} \Delta_i$ is the mesh of the grid $\boldsymbol{\tau}$. The constant $C(\mathbb{E}_q\|\boldsymbol{x}(0)\|^2, T_{\max}, M, D, L_{\boldsymbol{f}})$ does not depend on $(\boldsymbol{A}(t))_{0 \leq t \leq T_{\max}}$ or $(\boldsymbol{b}(t))_{0 \leq t \leq T_{\max}}$, except through $M$.

*Proof of Theorem 1.* For ease of notation, we will drop the dependence of $\boldsymbol{f}$ on $\boldsymbol{\Theta}$. And without loss of generality, we will assume $\boldsymbol{\Sigma}^{1/2} = \boldsymbol{I}$. This is because the ELBO is invariant under invertible transformations, and so applying the change-of-variable $\boldsymbol{x} \mapsto \boldsymbol{v} := \boldsymbol{\Sigma}^{-1/2}\boldsymbol{x}$ reduces the diffusion coefficient to identity and changes the prior drift to $\boldsymbol{\Sigma}^{-1/2} \circ \boldsymbol{f} \circ \boldsymbol{\Sigma}^{1/2}$. The variational posterior drift also transforms in the same way. The transformed prior drift has Lipschitz constant $\|\boldsymbol{\Sigma}^{-1/2}\|\|\boldsymbol{\Sigma}^{1/2}\|L_{\boldsymbol{f}}$.

Recall that we defined the difference between subsequent grid points to be $\Delta_i = \tau_{i+1} - \tau_i$, and let us define $\gamma(\boldsymbol{x}(u), u) = \boldsymbol{f}_q(\boldsymbol{x}(u), u) - \boldsymbol{f}(\boldsymbol{x}(u), u)$. Then we decompose the error between the continuous and approximate ELBO as

$$
\begin{aligned}
|\mathcal{L}_{\text{cont}}(q, \boldsymbol{\Theta}) - \mathcal{L}_{\text{approx}}(q, \boldsymbol{\Theta})| &\leq \left| \frac{1}{2} \int_0^{T_{\max}} \mathbb{E}_q\|\boldsymbol{f}(\boldsymbol{x}, t) - \boldsymbol{f}_q(\boldsymbol{x}, t)\|^2 dt - \frac{\Delta_i}{2} \sum_{i=0}^{T-1} \mathbb{E}_q \|\gamma(\boldsymbol{x}_i, \tau_i)\|^2 \right| \\
&\quad + \left| \frac{\Delta_i}{2} \sum_{i=0}^{T-1} \mathbb{E}_q \|\gamma(\boldsymbol{x}_i, \tau_i)\|^2 - \mathbb{E}_q \log \prod_{i=0}^{T-1} \frac{q(\boldsymbol{x}_{i+1}|\boldsymbol{x}_i)}{\tilde{p}(\boldsymbol{x}_{i+1}|\boldsymbol{x}_i)} \right|.
\end{aligned} \tag{16}
$$

We will proceed by bounding each of these two terms individually. To do so, we first state and prove the following three :

**Lemma 5:**

$$\sup_{0 \le t \le T_{\max}} \mathbb{E}_q \|\boldsymbol{x}(t)\|^2 \le (\mathbb{E}_q \|\boldsymbol{x}(0)\|^2 + DT_{\max} + M^2) \exp\{(2M+1)T_{\max}\} := C_0(\mathbb{E}_q \|\boldsymbol{x}(0)\|^2, T_{\max}, M, D)$$

*Proof.* By our assumptions on $(\boldsymbol{A}(u))_{0 \le u \le T_{\max}}$ and $(\boldsymbol{b}(u))_{0 \le u \le T_{\max}}$, the SDE $q$ admits a unique strong solution that satisfies $\int_0^1 \mathbb{E}_q \|\boldsymbol{x}(t)\|^2 dt < \infty$. By Ito's Lemma,

$$d\|\boldsymbol{x}(t)\|^2 = 2\langle \boldsymbol{x}(t), \boldsymbol{f}_q(\boldsymbol{x}(t), t)\rangle dt + 2\langle \boldsymbol{x}(t), d\boldsymbol{w}(t)\rangle + Ddt.$$

Taking the expectation of both sides and recognizing that the Ito integral is a bona fide martingale,

$$\mathbb{E}_q \|\boldsymbol{x}(t)\|^2 = \mathbb{E}_q \|\boldsymbol{x}(0)\|^2 + \int_0^t \{2\mathbb{E}_q \langle \boldsymbol{x}(u), \boldsymbol{f}_q(\boldsymbol{x}(u), u)\rangle + D\} du$$

$$\le \mathbb{E}_q \|\boldsymbol{x}(0)\|^2 + \int_0^t \{2\mathbb{E}_q\{\|\boldsymbol{x}(u)\|(M\|\boldsymbol{x}(u)\| + M)\} + D\} du$$

$$\le \mathbb{E}_q \|\boldsymbol{x}(0)\|^2 + D \cdot t + M^2 t + \int_0^t (2M+1)\mathbb{E}_q \|\boldsymbol{x}(u)\|^2 du.$$

Grönwall's inequality yields the desired inequality,

$$\sup_{0 \le t \le T_{\max}} \mathbb{E}_q \|\boldsymbol{x}(t)\|^2 \le (\mathbb{E}_q \|\boldsymbol{x}(0)\|^2 + DT_{\max} + M^2) \exp\{(2M+1)T_{\max}\}.$$

$\square$

**Lemma 6:** For all $0 \le s \le t \le T_{\max}$, we have the following bound

$$\mathbb{E}_q \|\boldsymbol{x}(t) - \boldsymbol{x}(s)\|^2 \le C_1(\mathbb{E}_q \|\boldsymbol{x}(0)\|^2, T_{\max}, M, D) \cdot (t-s)$$

for $C_1(\mathbb{E}_q \|\boldsymbol{x}_0\|^2, T_{\max}, M, D)$ not depending on $s, t$.

*Proof.* We start by rewriting the squared norm difference between $\boldsymbol{x}(t)$ and $\boldsymbol{x}(s)$

$$\|\boldsymbol{x}(t) - \boldsymbol{x}(s)\|^2 = \left\| \int_s^t d\boldsymbol{w}^q(u) + \int_s^t \boldsymbol{f}_q(\boldsymbol{x}(u), u) du \right\|^2$$

$$= \left\| \int_s^t d\boldsymbol{w}^q(u) + \int_s^t \boldsymbol{A}(u)(\boldsymbol{x}(u) - \boldsymbol{x}(s)) du + \int_s^t (\{\boldsymbol{A}(u) - \boldsymbol{A}(s)\}\boldsymbol{x}(s) + \{\boldsymbol{b}(u) - \boldsymbol{b}(s)\}) du \right.$$

$$\left. + (t-s)\boldsymbol{f}_q(\boldsymbol{x}(s), s) \right\|^2$$

$$\le 4\left\{ \left\| \int_s^t d\boldsymbol{w}^q(u) \right\|^2 + \left\| \int_s^t \boldsymbol{A}(u)(\boldsymbol{x}(u) - \boldsymbol{x}(s)) du \right\|^2 \right.$$

$$\left. + \left\| \int_s^t (\{\boldsymbol{A}(u) - \boldsymbol{A}(s)\}\boldsymbol{x}(s) + \{\boldsymbol{b}(u) - \boldsymbol{b}(s)\}) du \right\|^2 + (t-s)^2 \|\boldsymbol{A}(s)\boldsymbol{x}(s) + \boldsymbol{b}(s)\|^2 \right\},$$

where we use the bound $(\sum_{\ell=1}^L a_\ell)^2 \le L(\sum_{\ell=1}^L a_\ell^2)$. We take the expectation of both sides to obtain

$$\mathbb{E}_q \|\boldsymbol{x}(t) - \boldsymbol{x}(s)\|^2$$

$$\le 4\left\{ D(t-s) + (t-s)^2 \sup_{0 \le u \le T_{\max}} \mathbb{E}_q \|\boldsymbol{f}_q(\boldsymbol{x}(u), u)\|^2 + \mathbb{E}_q \left\| \int_s^t \boldsymbol{A}(u)(\boldsymbol{x}(u) - \boldsymbol{x}(s)) du \right\|^2 \right.$$

$$\left. + \mathbb{E}_q \left\| \int_s^t (\{\boldsymbol{A}(u) - \boldsymbol{A}(s)\}\boldsymbol{x}_s + \{\boldsymbol{b}(u) - \boldsymbol{b}(s)\}) du \right\|^2 \right\}.$$

By applying Cauchy-Schwarz to each of the final two terms, we get

$$\mathbb{E}_q \left\| \int_s^t \boldsymbol{A}(u)(\boldsymbol{x}(u) - \boldsymbol{x}(s))du \right\|^2 \leq (t-s) \int_s^t \mathbb{E}_q \|\boldsymbol{A}(u)(\boldsymbol{x}(u) - \boldsymbol{x}(s))\|^2 du$$

$$\leq M^2(t-s) \int_s^t \mathbb{E}_q \|\boldsymbol{x}(u) - \boldsymbol{x}(s)\|^2 du$$

as well as

$$\mathbb{E}_q \left\| \int_s^t (\{\boldsymbol{A}(u) - \boldsymbol{A}(s)\}\boldsymbol{x}(s) + \{\boldsymbol{b}(u) - \boldsymbol{b}(s)\})du \right\|^2$$

$$\leq (t-s) \int_s^t \mathbb{E}_q \|\{\boldsymbol{A}(u) - \boldsymbol{A}(s)\}\boldsymbol{x}(s) + \{\boldsymbol{b}(u) - \boldsymbol{b}(s)\}\|^2 du$$

$$\leq 2(t-s) \int_s^t \mathbb{E}_q (\|\{\boldsymbol{A}(u) - \boldsymbol{A}(s)\}\boldsymbol{x}(s)\|^2 + \|\boldsymbol{b}(u) - \boldsymbol{b}(s)\|^2) du$$

$$\leq 2(t-s)^3 M^2 \int_s^t (\mathbb{E}_q \|\boldsymbol{x}(s)\|^2 + 1) du$$

$$\leq 2(t-s)^4 M^2 \left( \sup_{0 \leq u \leq T_{\max}} \mathbb{E}_q \|\boldsymbol{x}(u)\|^2 + 1 \right).$$

Combining our above bounds and invoking Grönwall's inequality, we get

$$\mathbb{E}_q \|\boldsymbol{x}(t) - \boldsymbol{x}(s)\|^2 \leq 4 \Big\{ D(t-s) + (t-s)^2 \sup_{0 \leq u \leq T_{\max}} \mathbb{E}_q \|\boldsymbol{f}_q(\boldsymbol{x}(u), u)\|^2$$

$$+ 2(t-s)^4 M^2 \left( \sup_{0 \leq u \leq T_{\max}} \mathbb{E}_q \|\boldsymbol{x}(u)\|^2 + 1 \right) \Big\} \exp\{M^2(t-s)^2\}.$$

Recognizing the leading term as $(t-s)$ and invoking the bound on $\sup_{0 \leq u \leq T_{\max}} \mathbb{E}_q \|\boldsymbol{x}(u)\|^2$ from Lemma 5 we obtain the result

$$\mathbb{E}_q \|\boldsymbol{x}(t) - \boldsymbol{x}(s)\|^2 \leq C_1(\mathbb{E}_q \|\boldsymbol{x}(0)\|^2, T_{\max}, M, D) \cdot (t-s).$$

$$\square$$

**Lemma 7:** Under $q$, the mean $\overline{\boldsymbol{m}}_{i,i+1}$ and covariance $\overline{\boldsymbol{S}}_{i,i+1}$ of $q(\boldsymbol{x}_{i+1}|\boldsymbol{x}_i)$ satisfy

$$\overline{\boldsymbol{m}}_{i,i+1} = \boldsymbol{x}_i + \Delta_i \boldsymbol{f}_q(\boldsymbol{x}_i, \tau_i) + (\Delta_i)^2 (\boldsymbol{r}_i^{(1)} \boldsymbol{x}_i + \boldsymbol{r}_i^{(2)})$$

$$\overline{\boldsymbol{S}}_{i,i+1} = \Delta_i \boldsymbol{I} + \frac{(\Delta_i)^2}{2} (\boldsymbol{A}(\tau_i) + \boldsymbol{A}(\tau_i)^\top) + (\Delta_i)^3 \boldsymbol{R}_i$$

where $\|\boldsymbol{r}_i^{(1)}\|, \|\boldsymbol{r}_i^{(2)}\|, \|\boldsymbol{R}_i\| \leq C_2(M)$ are independent of $0 \leq i \leq T$.

*Proof.* Since $q$ is the (strong) solution to the linear SDE eq. (7), then $q(\boldsymbol{x}_{i+1}|\boldsymbol{x}_i)$ is Gaussian with mean and covariance

$$\overline{\boldsymbol{m}}_{i,i+1} := \boldsymbol{\Psi}(\tau_{i+1}, \tau_i)\boldsymbol{x}_i + \int_0^{\Delta_i} \boldsymbol{\Psi}(\tau_i + \Delta_i, \tau_i + u)\boldsymbol{b}(u)du$$

$$\overline{\boldsymbol{S}}_{i,i+1} := \int_0^{\Delta_i} \boldsymbol{\Psi}(\tau_i + \Delta_i, \tau_i + u)\boldsymbol{\Psi}(\tau_i + \Delta_i, \tau_i + u)^\top du,$$

$$\boldsymbol{\Psi}(t, s) := \boldsymbol{I} + \sum_{k=1}^\infty \int_s^t \int_s^{u_1} \cdots \int_s^{u_{k-1}} \boldsymbol{A}(u_1) \cdots \boldsymbol{A}(u_k) du_1 \cdots du_k, \quad 0 \leq s \leq t \leq T_{\max} \quad (17)$$

where the series form of the state transition matrix is known as the Peano-Baker series [59]. Truncation of the Peano-Baker series at its second term yields

$$\boldsymbol{\Psi}(t, s) := \boldsymbol{I} + \int_s^t \boldsymbol{A}(u)du + \boldsymbol{R}^{(1)}(t, s)$$

where

$$\|\boldsymbol{R}^{(1)}(t,s)\| = \left\|\sum_{k=2}^{\infty} \int_s^t \int_s^{u_1} \cdots \int_s^{u_{k-1}} \boldsymbol{A}(u_1) \cdots \boldsymbol{A}(u_k) du_1 \cdots du_k\right\|$$

$$\overset{(i)}{\leq} \sum_{k=2}^{\infty} \left\|\int_s^t \int_s^{u_1} \cdots \int_s^{u_{k-1}} \boldsymbol{A}(u_1) \cdots \boldsymbol{A}(u_k) du_1 \cdots du_k\right\|$$

$$\overset{(ii)}{\leq} \sum_{k=2}^{\infty} \frac{(t-s)^k M^k}{k!}$$

$$= \exp((t-s)M) - 1 - (t-s) \leq \frac{M^2(t-s)^2}{2}\exp(M(t-s)).$$

In inequality (i) we used the fact that the series converges absolutely and for equality (ii) we used $\int_s^t \int_s^{u_1} \cdots \int_s^{u_{k-1}} du_1 \cdots du_k = (t-s)^k/k!$ as well as our assumption $\|\boldsymbol{A}(t)\| \leq M$. Moreover, by the Mean-Value Theorem,

$$\int_s^t \boldsymbol{A}(u)du = (t-s)\boldsymbol{A}(s) + \boldsymbol{R}^{(2)}(t,s), \quad \|\boldsymbol{R}^{(2)}(t,s)\| \leq M(t-s)^2 \tag{18}$$

where we used our assumption that $\boldsymbol{A}(t)$ is $M$-Lipschitz on $[0, T_{\max}]$, and hence its derivative is bounded in operator norm by $M$. Plugging this into our expression eq. (17) for the mean and covariance matrix,

$$\overline{\boldsymbol{m}}_{i,i+1} = \boldsymbol{x}_i + \Delta_i \boldsymbol{f}_q(\boldsymbol{x}_i, \tau_i) + (\Delta_i)^2(\boldsymbol{r}_i^{(1)}\boldsymbol{x}_i + \boldsymbol{r}_i^{(2)})$$

$$\overline{\boldsymbol{S}}_{i,i+1} = \Delta_i \boldsymbol{I} + \frac{(\Delta_i)^2}{2}(\boldsymbol{A}(\tau_i) + \boldsymbol{A}(\tau_i)^\top) + (\Delta_i)^3 \boldsymbol{R}_i$$

where $\|\boldsymbol{r}_i^{(1)}\|, \|\boldsymbol{r}_i^{(2)}\|, \|\boldsymbol{R}_i\| \leq C_2(M)$ are independent of the index $i$. $\qquad\square$

As an immediate consequence of Lemmas 5 and 6, we have for any $0 \leq s \leq t \leq T_{\max}$,

$$\mathbb{E}_q \|\boldsymbol{f}_q(\boldsymbol{x}(t), t) - \boldsymbol{f}_q(\boldsymbol{x}(s), s)\|^2$$

$$\leq 3\mathbb{E}_q \left\{\|\boldsymbol{A}(t)\|^2 \|\boldsymbol{x}(t) - \boldsymbol{x}(s)\|^2 + \|\boldsymbol{A}(t) - \boldsymbol{A}(s)\|^2 \|\boldsymbol{x}(s)\|^2 + \|\boldsymbol{b}(t) - \boldsymbol{b}(s)\|^2\right\}$$

$$\leq 3M^2 \mathbb{E}_q \|\boldsymbol{x}(t) - \boldsymbol{x}(s)\|^2 + (t-s)^2 M^2 \left\{1 + \sup_{0 \leq u \leq T_{\max}} \mathbb{E}_q \|\boldsymbol{x}(u)\|^2\right\}$$

$$\leq C_3(\mathbb{E}_q \|\boldsymbol{x}(0)\|^2, T_{\max}, M, D) \cdot (t-s).$$

We also have that for any $0 \leq t \leq T_{\max}$,

$$\mathbb{E}_q \|\gamma(\boldsymbol{x}(t), t)\|^2 \leq 2\left(M^2 \left\{1 + \sup_{0 \leq u \leq T_{\max}} \mathbb{E}_q \|\boldsymbol{x}(u)\|^2\right\} + \sup_{0 \leq u \leq T_{\max}} \mathbb{E}_q \|\boldsymbol{f}(\boldsymbol{x}(u), u)\|^2\right)$$

$$\leq C_4(\mathbb{E}_q \|\boldsymbol{x}(0)\|^2, T_{\max}, M, D).$$

For the second inequality we invoke the linear growth condition on $\boldsymbol{f}$ and Lemma 5.

Equipped with these results, we are finally prepared to handle the two terms in the decomposition eq. (16). For the first term,

$$\frac{1}{2}\left|\int_0^{T_{\max}} \mathbb{E}_q \|\boldsymbol{f}(\boldsymbol{x}(u), u) - \boldsymbol{f}_q(\boldsymbol{x}(u), u)\|^2 du - \sum_{i=0}^{T-1} \Delta_i \mathbb{E}_q \|\gamma(\boldsymbol{x}_i, \tau_i)\|^2\right|$$

$$= \frac{1}{2}\left|\sum_{i=0}^{T-1} \int_{\tau_i}^{\tau_{i+1}} \left\{\mathbb{E}_q \left\|\gamma^{(1)}(\boldsymbol{x}_i, \tau_i)\right\|^2 - \mathbb{E}_q \|\gamma^{(1)}(\boldsymbol{x}(u), u)\|^2\right\} du\right|$$

$$\leq \frac{1}{2}\sum_{i=0}^{T-1} \int_{\tau_i}^{\tau_{i+1}} \mathbb{E}_q \left\{(\|\gamma^{(1)}(\boldsymbol{x}_i, \tau_i)\| + \|\gamma^{(1)}(\boldsymbol{x}(u), u)\|)\|\gamma^{(1)}(\boldsymbol{x}_i, \tau_i) - \gamma^{(1)}(\boldsymbol{x}(u), u)\|\right\} du$$

$$\leq \frac{1}{2}\sum_{i=0}^{T-1}\int_{\tau_i}^{\tau_{i+1}} 2\sqrt{C_4}\sqrt{\mathbb{E}_q\|\gamma^{(1)}(\boldsymbol{x}_i,\tau_i)-\gamma^{(1)}(\boldsymbol{x}(u),u)\|^2}\,du$$

$$\leq \frac{1}{2}\sum_{i=0}^{T-1}\int_{\tau_i}^{\tau_{i+1}} 2\sqrt{2C_4}\sqrt{(2L_{\boldsymbol{f}}^2+C_3)\mathbb{E}_q\|\boldsymbol{x}_i-\boldsymbol{x}(u)\|^2+2L_{\boldsymbol{f}}^2(t-u)^2}\,du$$

$$\leq \sum_{i=0}^{T-1}\Delta_t^{3/2}C_5(\mathbb{E}_q\|\boldsymbol{x}_0\|^2,T_{\max},M,D,L_{\boldsymbol{f}})$$

$$\leq (\Delta t)^{1/2}C_5(\mathbb{E}_q\|\boldsymbol{x}_0\|^2,T_{\max},M,D,L_{\boldsymbol{f}}).$$

Altogether, we have proven that the first term in eq. (16) is $\mathcal{O}((\Delta t)^{1/2})$.

For the second term, we know that the strong solution $q$ to the linear SDE eq. (7) has Gaussian transition densities $q(\boldsymbol{x}_{i+1}|\boldsymbol{x}_i)$. By the KL formula for two Gaussian distributions

$$\mathbb{E}_q\log\frac{q(\boldsymbol{x}_{i+1}|\boldsymbol{x}_i)}{\tilde{p}(\boldsymbol{x}_{i+1}|\boldsymbol{x}_i)}=\frac{1}{2}\mathbb{E}_q\left(\frac{1}{\Delta_i}\mathrm{tr}(\overline{\boldsymbol{S}}_{i+1,i})-D+D\log\Delta_i-\log|\overline{\boldsymbol{S}}_{i,i+1}|+\frac{1}{\Delta_i}\|\overline{\boldsymbol{m}}_{i,i+1}-(\boldsymbol{x}_i+\Delta_i\boldsymbol{f}(\boldsymbol{x}_i,\tau_i))\|_2^2\right)$$

where $\overline{\boldsymbol{m}}_{i,i+1}$ and $\overline{\boldsymbol{S}}_{i,i+1}$ are defined as in Lemma 7. By Lemma 7, the mean term is

$$\frac{1}{2\Delta_i}\mathbb{E}_q\|\overline{\boldsymbol{m}}_{i,i+1}-(\boldsymbol{x}_i+\Delta_i\boldsymbol{f}(\boldsymbol{x}_i,\tau_i))\|_2^2$$
$$=\frac{\Delta_i}{2}\mathbb{E}_q\|\boldsymbol{f}_q(\boldsymbol{x}_i,\tau_i)-\boldsymbol{f}(\boldsymbol{x}_i,\tau_i)\|^2+\Delta_i^2 C_6(\mathbb{E}_q\|\boldsymbol{x}_0\|^2,M,D,L_{\boldsymbol{f}})$$
$$=\frac{\Delta_i}{2}\mathbb{E}_q\|\gamma(\boldsymbol{x}_i,\tau_i)\|^2+\Delta_i^2 C_6(\mathbb{E}_q\|\boldsymbol{x}_0\|^2,M,D,L_{\boldsymbol{f}})$$

In the second line, we expand the square and invoke the bound on $\sup_{0\leq t\leq T_{\max}}\mathbb{E}_q\|\boldsymbol{x}(t)\|^2$ from Lemma 5. Also by Lemma 7, the covariance term satisfies

$$\frac{1}{\Delta_i}\mathrm{tr}(\overline{\boldsymbol{S}}_{i,i+1})=D+\Delta_i\mathrm{tr}(\boldsymbol{A}(\tau_i)+\boldsymbol{A}(\tau_i)^\top)+\Delta_i^2\mathrm{tr}(\boldsymbol{R}_i)$$
$$\log|\overline{\boldsymbol{S}}_{i,i+1}|=D\log\Delta_i+\Delta_i\mathrm{tr}(\boldsymbol{A}(\tau_i)+\boldsymbol{A}(\tau_i)^\top)+\Delta_i^2 C_7(M).$$

Altogether, we have shown

$$\frac{\Delta_i}{2}\mathbb{E}_q\|\gamma(\boldsymbol{x}_i,\tau_i)\|^2-\mathbb{E}_q\log\frac{q(\boldsymbol{x}_{i+1}|\boldsymbol{x}_i)}{\tilde{p}(\boldsymbol{x}_{i+1}|\boldsymbol{x}_i)}=\Delta_i^2 C_8(\mathbb{E}_q\|\boldsymbol{x}_0\|^2,T_{\max},M,D,L_{\boldsymbol{f}}).$$

Summing across time steps yields

$$\left|\frac{\Delta_i}{2}\sum_{i=0}^{T-1}\mathbb{E}_q\|\gamma(\boldsymbol{x}_i,\tau_i)\|^2-\mathbb{E}_q\log\prod_{i=0}^{T-1}\frac{q(\boldsymbol{x}_{i+1}|\boldsymbol{x}_i)}{\tilde{p}(\boldsymbol{x}_{i+1}|\boldsymbol{x}_i)}\right|\leq(\Delta t)C_8(\mathbb{E}_q\|\boldsymbol{x}_0\|^2,T_{\max},M,D,L_{\boldsymbol{f}}).$$

$\square$

# F  Computing expectations of prior transition densities

Here, we provide a proof of Proposition 1 from Section 3.3, which says that $\mathbb{E}_{q(\boldsymbol{x})}[\log\tilde{p}(\boldsymbol{x}_{i+1}|\boldsymbol{x}_i,\boldsymbol{\Theta})]$ can be written as an expectation with respect to $q(\boldsymbol{x}_i)$ only. To do this, we first prove the following statement, which is a consequence of Stein's lemma.

**Lemma 8:**  Suppose $\begin{bmatrix}\boldsymbol{x}_i\\\boldsymbol{x}_{i+1}\end{bmatrix}\sim\mathcal{N}\left(\begin{bmatrix}\boldsymbol{m}_i\\\boldsymbol{m}_{i+1}\end{bmatrix},\begin{bmatrix}\boldsymbol{S}_i&\boldsymbol{S}_{i+1,i}^\top\\\boldsymbol{S}_{i+1,i}&\boldsymbol{S}_{i+1}\end{bmatrix}\right)$ where $\boldsymbol{x}_i,\boldsymbol{x}_{i+1}\in\mathbb{R}^D$. Then, for any $\boldsymbol{f}:\mathbb{R}^D\to\mathbb{R}^D$, $\mathbb{E}\left[\boldsymbol{f}(\boldsymbol{x}_i)(\boldsymbol{x}_{i+1}-\boldsymbol{m}_{i+1})^\top\right]=\mathbb{E}[\boldsymbol{J}_{\boldsymbol{f}}(\boldsymbol{x}_i)]\boldsymbol{S}_{i+1,i}^\top$, where $\boldsymbol{J}_{\boldsymbol{f}}(\boldsymbol{x}_i)$ is the Jacobian of $\boldsymbol{f}$ evaluated at $\boldsymbol{x}_i$.

*Proof.*  First, for notational simplicity, we denote

$$\tilde{\boldsymbol{x}}:=\begin{bmatrix}\boldsymbol{x}_i\\\boldsymbol{x}_{i+1}\end{bmatrix},\quad \boldsymbol{g}(\tilde{\boldsymbol{x}}):=\begin{bmatrix}\boldsymbol{f}(\boldsymbol{x}_i)\\\boldsymbol{f}(\boldsymbol{x}_{i+1})\end{bmatrix},\quad \tilde{\boldsymbol{m}}=\begin{bmatrix}\boldsymbol{m}_i\\\boldsymbol{m}_{i+1}\end{bmatrix},\quad \tilde{\boldsymbol{S}}=\begin{bmatrix}\boldsymbol{S}_i&\boldsymbol{S}_{i+1,i}^\top\\\boldsymbol{S}_{i+1,i}&\boldsymbol{S}_{i+1}\end{bmatrix}.$$

Next, we apply Stein's lemma to the Gaussian random vector $\tilde{x}$ and function $g(\cdot)$. This yields the identity,

$$\mathbb{E}_{q(\tilde{x})}\left[g(\tilde{x})(\tilde{x}-\tilde{m})^\top\right] = \mathbb{E}_{q(\tilde{x})}\left[J_g(\tilde{x})\right]\tilde{S}, \tag{19}$$

where $J_g(\tilde{x})$ is the Jacobian of $g$ evaluated at $\tilde{x}$:

$$J_g(\tilde{x}) = \begin{bmatrix} J_f(x_i) & 0 \\ 0 & J_f(x_{i+1}) \end{bmatrix}.$$

Reading off the bottom right block of matrices in eq. (19) gives,

$$\mathbb{E}\left[f(x_i)(x_{i+1}-m_{i+1})^\top\right] = \mathbb{E}\left[J_f(x_i)\right]S_{i+1,i}^\top.$$

$\square$

Using this lemma, we can now prove Proposition 1.

**Proposition 1:** The term $\mathbb{E}_q[\log\tilde{p}(x_{i+1}|x_i,\Theta)]$ can equivalently be written in terms of the mean parameters $(\mu_i, \mu_{i+1})$ and the expectations $\mathbb{E}_q\left[f(x_i|\Theta)\right], \mathbb{E}_q\left[f(x_i|\Theta)^\top\Sigma^{-1}f(x_i|\Theta)\right]$, and $\mathbb{E}_q\left[J_{f(\cdot|\Theta)}(x_i)\right]$, thereby reducing the dimensionality of integration from $\mathbb{R}^{2D}$ to $\mathbb{R}^D$.

*Proof.* Expanding the transition expectation term, we have

$$\mathbb{E}_q[\log\tilde{p}(x_{i+1}\mid x_i,\Theta)] = \mathbb{E}_q[\log\mathcal{N}(x_{i+1}\mid x_i+\Delta_i f(x_i|\Theta), \Delta_i\Sigma)]$$

$$= -\frac{D}{2}\log(2\pi\Delta_i) - \frac{1}{2\Delta_i}\mathbb{E}_q\left[\|x_{i+1}-x_i-\Delta_i f(x_i|\Theta)\|_{\Sigma^{-1}}^2\right].$$

We can rewrite the expectation in the above line as

$$\mathbb{E}_q\left[\|x_{i+1}-x_i-\Delta_i f(x_i|\Theta)\|_{\Sigma^{-1}}^2\right]$$

$$= \mathbb{E}_q[\|x_{i+1}\|_{\Sigma^{-1}}^2 + \|x_i\|_{\Sigma^{-1}}^2 + \Delta_i^2\|f(x_i|\Theta)\|_{\Sigma^{-1}}^2 - 2x_{i+1}^\top\Sigma^{-1}x_i - 2\Delta_i x_{i+1}^\top\Sigma^{-1}f(x_i|\Theta)$$

$$+ 2\Delta_i x_i^\top\Sigma^{-1}f(x_i|\Theta)]$$

$$= \mathrm{tr}(\Sigma^{-1}(S_{i+1}+m_{i+1}m_{i+1}^\top)) + \mathrm{tr}(\Sigma^{-1}(S_i+m_i m_i^\top)) + \Delta_i^2\mathbb{E}_q[f(x_i|\Theta)^\top\Sigma^{-1}f(x_i|\Theta)]$$

$$- 2\mathrm{tr}(\Sigma^{-1}(S_{i+1,i}^\top+m_i m_{i+1}^\top)) - 2\Delta_i\mathbb{E}_q\left[x_{i+1}^\top\Sigma^{-1}f(x_i|\Theta)\right] + 2\Delta_i\mathbb{E}_q\left[x_i^\top\Sigma^{-1}f(x_i|\Theta)\right]$$

Now, applying Lemma 8 to the second to last term, we have

$$\mathbb{E}_q\left[x_{i+1}^\top\Sigma^{-1}f(x_i|\Theta)\right] = \mathrm{tr}(\Sigma^{-1}(\mathbb{E}_q[f(x_i|\Theta)]m_{i+1}^\top + \mathbb{E}_q\left[J_{f(\cdot|\Theta)}(x_i)\right]S_{i+1,i}^\top)).$$

Applying Stein's lemma to the last term, we have

$$\mathbb{E}_q\left[x_i^\top\Sigma^{-1}f(x_i|\Theta)\right] = \mathrm{tr}(\Sigma^{-1}(\mathbb{E}_q[(f(x_i|\Theta)]m_i^\top + J_{f(\cdot|\Theta)}(x_i)S_i)).$$

Thus, we have shown that $\mathbb{E}_q[\log\tilde{p}(x_{i+1}|x_i,\Theta)]$ can be written in terms of $\mu_i$, $\mu_{i+1}$, and the expectations $\mathbb{E}_q\left[f(x_i|\Theta)\right], \mathbb{E}_q\left[f(x_i|\Theta)^\top\Sigma^{-1}f(x_i|\Theta)\right]$, and $\mathbb{E}_q\left[J_{f(\cdot|\Theta)}(x_i)\right]$. $\square$

## G   Parallelizing SING

### G.1   Sequential computation of log normalizer

Here, we describe the naive sequential approach of converting from natural parameters $\eta$ to mean parameters $\mu$ of $q(x_{0:T})$. The idea is to marginalize out one variable at a time in eq. (12), starting from $x_0$ and ending at $x_T$.

We'll first show how to marginalize out $x_0$ and $x_1$, and then use this to derive a general recursive update for this sequential approach. To marginalize out $x_0$, we focus on the following integral which contains all terms depending on $x_0$.

$$\int \exp\left(-\frac{1}{2}x_0^\top J_0 x_0 + (h_0 - L_0^\top x_1)^\top x_0\right)dx_0$$

$$= \underbrace{(2\pi)^{\frac{K}{2}}|J_0|^{-\frac{1}{2}}\exp\left(\frac{1}{2}h_0^\top J_0^{-1}h_0\right)}_{:=Z_0}\exp\left(h_1^{(p)\top}x_1 - \frac{1}{2}x_1^\top J_1^{(p)}x_1\right),$$

where $Z_0$ is the contribution to the log normalizer from marginalizing out $\boldsymbol{x}_0$, and

$$\boldsymbol{J}_1^{(p)} = -\boldsymbol{L}_0\boldsymbol{J}_0^{-1}\boldsymbol{L}_0^\top, \quad \boldsymbol{h}_1^{(p)} = -\boldsymbol{L}_0\boldsymbol{J}_0^{-1}\boldsymbol{h}_0.$$

Next, when marginalizing out $\boldsymbol{x}_1$, we should integrate the potential $\exp\left(\boldsymbol{h}_1^{(p)\top}\boldsymbol{x}_1 - \frac{1}{2}\boldsymbol{x}_1^\top\boldsymbol{J}_1^{(p)}\boldsymbol{x}_1\right)$ from the previous step multiplied by the rest of the terms in $\boldsymbol{A}(\boldsymbol{\eta})$ that depend on $\boldsymbol{x}_1$. This gives the following integral,

$$\int \exp\left(\boldsymbol{h}_1^{(p)\top}\boldsymbol{x}_1 - \frac{1}{2}\boldsymbol{x}_1^\top\boldsymbol{J}_1^{(p)}\boldsymbol{x}_1\right)\exp\left(-\frac{1}{2}\boldsymbol{x}_1^\top\boldsymbol{J}_1\boldsymbol{x}_1 - \boldsymbol{x}_2^\top\boldsymbol{L}_1\boldsymbol{x}_1 + \boldsymbol{h}_1^\top\boldsymbol{x}_1\right)d\boldsymbol{x}_1$$

$$= \int \exp\left(-\frac{1}{2}\boldsymbol{x}_1^\top\underbrace{(\boldsymbol{J}_1 + \boldsymbol{J}_1^{(p)})}_{:=\boldsymbol{J}_1^{(c)}}\boldsymbol{x}_1 + (\underbrace{\boldsymbol{h}_1 + \boldsymbol{h}_1^{(p)}}_{:=\boldsymbol{h}_1^{(c)}} - \boldsymbol{L}_1^\top\boldsymbol{x}_2)^\top\boldsymbol{x}_1\right)d\boldsymbol{x}_1$$

$$= \underbrace{(2\pi)^{\frac{K}{2}}|\boldsymbol{J}_1|^{-\frac{1}{2}}\exp\left(\frac{1}{2}\boldsymbol{h}_1^\top\boldsymbol{J}_1^{-1}\boldsymbol{h}_1\right)}_{:=Z_1}\exp\left(\boldsymbol{h}_2^{(p)\top}\boldsymbol{x}_2 - \frac{1}{2}\boldsymbol{x}_2^\top\boldsymbol{J}_2^{(p)}\boldsymbol{x}_2\right)$$

where $Z_1$ is the contribution to the log normalizer from marginalizing out $\boldsymbol{x}_1$, and

$$\boldsymbol{J}_2^{(p)} = -\boldsymbol{L}_1\boldsymbol{J}_1^{(c)-1}\boldsymbol{L}_1^\top, \quad \boldsymbol{h}_2^{(p)} = -\boldsymbol{L}_1\boldsymbol{J}_1^{(c)-1}\boldsymbol{h}_1^{(c)}.$$

Generalizing these recursions leads to the following sequential algorithm:

1. Initialize $\boldsymbol{J}_0^{(p)} = \boldsymbol{0}$, $\boldsymbol{h}_0^{(p)} = \boldsymbol{0}$, and $\log Z = 0$.
2. For $i = 0, \ldots, T$:
   (a) Update $\boldsymbol{J}_i^{(c)} \leftarrow \boldsymbol{J}_i + \boldsymbol{J}_i^{(p)}$ and $\boldsymbol{h}_i^{(c)} \leftarrow \boldsymbol{h}_i + \boldsymbol{h}_i^{(p)}$.
   (b) Update $\log Z \leftarrow \log Z + \log Z_i$.
   (c) Update $\boldsymbol{J}_{i+1}^{(p)} \leftarrow -\boldsymbol{L}_i\boldsymbol{J}_i^{(c)-1}\boldsymbol{L}_i^\top$ and $\boldsymbol{h}_{i+1}^{(p)} \leftarrow -\boldsymbol{L}_i\boldsymbol{J}_i^{(c)-1}\boldsymbol{h}_i^{(c)}$.
3. Return $A(\boldsymbol{\eta}) = \log Z$.

**Time complexity**    In each iteration $i$, this sequential algorithm has time complexity $\mathcal{O}(D^3)$ which comes from the matrix inversion and multiplication in $-\boldsymbol{L}_i\boldsymbol{J}_i^{(c)-1}\boldsymbol{L}_i^\top$. Since each iteration relies on values computed in the previous iteration, this algorithm must be implemented sequentially, leading to a total time complexity of $\mathcal{O}(D^3T)$.

### G.2    Parallelized computation of log normalizer

Here, we present our new method of converting from natural to mean parameters in *parallel time* using associative scans, leading to substantial computational speedups on modern parallel hardware. As discussed in Section 3.4, we accomplish this by defining elements in the scan as unnormalized Gaussian potentials $a_{i,j}$ and combine them using the binary associative operator,

$$a_{i,j} \bullet a_{j,k} = \int a_{i,j}a_{j,k}d\boldsymbol{x}_j. \tag{20}$$

Note that eq. (20) is indeed a binary associative operator, as it satisfies the identity:

$$(a_{i,j} \bullet a_{j,k}) \bullet a_{k,l} = \int \left(\int a_{i,j}a_{j,k}d\boldsymbol{x}_j\right)a_{k,l}d\boldsymbol{x}_k$$

$$= \int a_{i,j}\left(\int a_{j,k}a_{k,l}d\boldsymbol{x}_k\right)d\boldsymbol{x}_j$$

$$= a_{i,j} \bullet (a_{j,k} \bullet a_{k,l}).$$

In the following calculation, we show how to compute the binary associative operator between two potentials analytically. We will start with computing the operator between two *consecutive* potentials,

and use this calculation to generalize to computing any pair of potentials with a shared variable to be marginalized.

Let $a_{i-1,i}$ and $a_{i,i+1}$ be two consecutive potentials. Then,

$$a_{i-1,i} \bullet a_{i,i+1}$$

$$= \int \exp\left(-\frac{1}{2}\boldsymbol{x}_i^\top \boldsymbol{J}_i \boldsymbol{x}_i - \boldsymbol{x}_{i-1}^\top \boldsymbol{L}_{i-1}^\top \boldsymbol{x}_i + \boldsymbol{h}_i^\top \boldsymbol{x}_i\right) \exp\left(-\frac{1}{2}\boldsymbol{x}_{i+1}^\top \boldsymbol{J}_{i+1}\boldsymbol{x}_{i+1} - \boldsymbol{x}_i^\top \boldsymbol{L}_i^\top \boldsymbol{x}_{i+1} + \boldsymbol{h}_{i+1}^\top \boldsymbol{x}_{i+1}\right) d\boldsymbol{x}_i$$

$$= \exp\left(-\frac{1}{2}\boldsymbol{x}_{i+1}^\top \boldsymbol{J}_{i+1}\boldsymbol{x}_{i+1} + \boldsymbol{h}_{i+1}^\top \boldsymbol{x}_{i+1}\right) \int \exp\left(-\frac{1}{2}\boldsymbol{x}_i^\top \boldsymbol{J}_i \boldsymbol{x}_i + (\boldsymbol{h}_i - \boldsymbol{L}_{i-1}\boldsymbol{x}_{i-1} - \boldsymbol{L}_i^\top \boldsymbol{x}_{i+1})^\top \boldsymbol{x}_i\right) d\boldsymbol{x}_i$$

$$= \underbrace{(2\pi)^{\frac{D}{2}}|\boldsymbol{J}_i|^{-\frac{1}{2}}}_{:=Z_{\text{local}}} \exp\left(-\frac{1}{2}\begin{pmatrix}\boldsymbol{x}_{i-1}\\\boldsymbol{x}_{i+1}\end{pmatrix}^\top \begin{pmatrix}\boldsymbol{J}_1^{(p)} & \boldsymbol{L}^{(p)\top}\\\boldsymbol{L}^{(p)} & \boldsymbol{J}_2^{(p)}\end{pmatrix}\begin{pmatrix}\boldsymbol{x}_{i-1}\\\boldsymbol{x}_{i+1}\end{pmatrix} + \begin{pmatrix}\boldsymbol{x}_{i-1}\\\boldsymbol{x}_{i+1}\end{pmatrix}^\top \begin{pmatrix}\boldsymbol{h}_1^{(p)}\\\boldsymbol{h}_2^{(p)}\end{pmatrix}\right)$$

$$= a_{i-1,i+1}$$

where $Z_{\text{local}}$ is the contribution to the log normalizer from this operation, and

$$\boldsymbol{J}_1^{(p)} = -\boldsymbol{L}_{i-1}^\top \boldsymbol{J}_i^{-1}\boldsymbol{L}_{i-1}, \quad \boldsymbol{J}_2^{(p)} = \boldsymbol{J}_{i+1} - \boldsymbol{L}_i \boldsymbol{J}_i^{-1}\boldsymbol{L}_i^\top,$$

$$\boldsymbol{h}_1^{(p)} = -\boldsymbol{L}_{i-1}^\top \boldsymbol{J}_i^{-1}\boldsymbol{h}_i, \quad \boldsymbol{h}_2^{(p)} = \boldsymbol{h}_{i+1} - \boldsymbol{L}_i \boldsymbol{J}_i^{-1}\boldsymbol{h}_i,$$

$$\boldsymbol{L}^{(p)} = \boldsymbol{L}_i \boldsymbol{J}_i^{-1}\boldsymbol{L}_{i-1}.$$

From the calculation above, we observe that the element $a_{i,j}$, where $i$ and $j$ are not necessarily consecutive, takes the general form

$$a_{i,j} = Z_{\text{local}} \exp\left(-\frac{1}{2}\begin{pmatrix}\boldsymbol{x}_i\\\boldsymbol{x}_j\end{pmatrix}^\top \begin{pmatrix}\boldsymbol{J}_1 & \boldsymbol{L}^\top\\\boldsymbol{L} & \boldsymbol{J}_2\end{pmatrix}\begin{pmatrix}\boldsymbol{x}_i\\\boldsymbol{x}_j\end{pmatrix} + \begin{pmatrix}\boldsymbol{x}_i\\\boldsymbol{x}_j\end{pmatrix}^\top \begin{pmatrix}\boldsymbol{h}_1\\\boldsymbol{h}_2\end{pmatrix}\right).$$

Using this form, we now derive the analytical result of the binary associative operator between any two Gaussian potentials with a shared marginalization variable.

$$a_{i,j} \bullet a_{j,k}$$

$$= \int Z_{\text{local}} \exp\left(-\frac{1}{2}\begin{pmatrix}\boldsymbol{x}_i\\\boldsymbol{x}_j\end{pmatrix}^\top \begin{pmatrix}\boldsymbol{J}_1 & \boldsymbol{L}^\top\\\boldsymbol{L} & \boldsymbol{J}_2\end{pmatrix}\begin{pmatrix}\boldsymbol{x}_i\\\boldsymbol{x}_j\end{pmatrix} + \begin{pmatrix}\boldsymbol{x}_i\\\boldsymbol{x}_j\end{pmatrix}^\top \begin{pmatrix}\boldsymbol{h}_1\\\boldsymbol{h}_2\end{pmatrix}\right)$$

$$\cdot \tilde{Z}_{\text{local}} \exp\left(-\frac{1}{2}\begin{pmatrix}\boldsymbol{x}_j\\\boldsymbol{x}_k\end{pmatrix}^\top \begin{pmatrix}\tilde{\boldsymbol{J}}_1 & \tilde{\boldsymbol{L}}^\top\\\tilde{\boldsymbol{L}} & \tilde{\boldsymbol{J}}_2\end{pmatrix}\begin{pmatrix}\boldsymbol{x}_j\\\boldsymbol{x}_k\end{pmatrix} + \begin{pmatrix}\boldsymbol{x}_j\\\boldsymbol{x}_k\end{pmatrix}^\top \begin{pmatrix}\tilde{\boldsymbol{h}}_1\\\tilde{\boldsymbol{h}}_2\end{pmatrix}\right) d\boldsymbol{x}_j$$

$$= Z_{\text{local}}\tilde{Z}_{\text{local}} \exp\left(-\frac{1}{2}\boldsymbol{x}_i^\top \boldsymbol{J}_1 \boldsymbol{x}_i + \boldsymbol{x}_i^\top \boldsymbol{h}_1 - \frac{1}{2}\boldsymbol{x}_k^\top \tilde{\boldsymbol{J}}_2 \boldsymbol{x}_k + \boldsymbol{x}_k^\top \tilde{\boldsymbol{h}}_2\right)$$

$$\cdot \int \exp\left(-\frac{1}{2}\boldsymbol{x}_j^\top \underbrace{(\boldsymbol{J}_2 + \tilde{\boldsymbol{J}}_1)}_{:=\boldsymbol{J}^{(c)}}\boldsymbol{x}_j + \boldsymbol{x}_j^\top (-\boldsymbol{L}^{(p)}\boldsymbol{x}_i - \tilde{\boldsymbol{L}}^{(p)\top}\boldsymbol{x}_k + \underbrace{\boldsymbol{h}_2 + \tilde{\boldsymbol{h}}_1}_{:=\boldsymbol{h}^{(c)}})\right) d\boldsymbol{x}_j$$

$$= Z_{\text{local}}^{(p)} \exp\left(-\frac{1}{2}\begin{pmatrix}\boldsymbol{x}_i\\\boldsymbol{x}_k\end{pmatrix}^\top \begin{pmatrix}\boldsymbol{J}_1^{(p)} & \boldsymbol{L}^{(p)\top}\\\boldsymbol{L}^{(p)} & \boldsymbol{J}_2^{(p)}\end{pmatrix}\begin{pmatrix}\boldsymbol{x}_i\\\boldsymbol{x}_k\end{pmatrix} + \begin{pmatrix}\boldsymbol{x}_i\\\boldsymbol{x}_k\end{pmatrix}^\top \begin{pmatrix}\boldsymbol{h}_1^{(p)}\\\boldsymbol{h}_2^{(p)}\end{pmatrix}\right)$$

$$= a_{i,k}$$

where

$$\boldsymbol{J}_1^{(p)} = \boldsymbol{J}_1 - \boldsymbol{L}^\top \boldsymbol{J}^{(c)-1}\boldsymbol{L}, \quad \boldsymbol{J}_2^{(p)} = \tilde{\boldsymbol{J}}_2 - \tilde{\boldsymbol{L}}\boldsymbol{J}^{(c)-1}\tilde{\boldsymbol{L}}^\top$$

$$\boldsymbol{h}_1^{(p)} = \boldsymbol{h}_1 - \boldsymbol{L}^\top \boldsymbol{J}^{(c)-1}\boldsymbol{h}^{(c)}, \quad \boldsymbol{h}_2^{(p)} = \tilde{\boldsymbol{h}}_1 - \tilde{\boldsymbol{L}}\boldsymbol{J}^{(c)-1}\boldsymbol{h}^{(c)}$$

$$\boldsymbol{L}^{(p)} = -\tilde{\boldsymbol{L}}\boldsymbol{J}^{(c)-1}\boldsymbol{L},$$

$$Z_{\text{local}}^{(p)} = Z^{\text{local}}\tilde{Z}^{\text{local}}(2\pi)^{\frac{D}{2}}|\boldsymbol{J}^{(c)}|^{-\frac{1}{2}}\exp\left(\frac{1}{2}\boldsymbol{h}^{(c)\top}\boldsymbol{J}^{(c)-1}\boldsymbol{h}^{(c)}\right)$$

In practice, instead of computing $Z_{\text{local}}^{(p)}$ we instead compute $\log Z_{\text{local}}^{(p)}$. Since each binary operation marginalizes out one variable, the final result of the associative scan,

$$a_{-1,0} \bullet a_{0,1} \bullet \cdots \bullet a_{T,T+1},$$

will be the result of accumulating all $\log Z_{\text{local}}$ terms, and will hence be equal to the desired log normalizer $A(\boldsymbol{\eta})$.

**Time complexity** The time complexity of the associative scan, assuming $T$ processors, is $\mathcal{O}(C \log T)$ where $C$ represents the cost of matrix operations within each application of the operator [34, 48]. By the derivation above, $C = \mathcal{O}(D^3)$ which comes from the matrix inversion and multiplication in $\boldsymbol{L}^\top \boldsymbol{J}^{(c)-1} \boldsymbol{L}$ and analogous terms. Therefore, the parallel time complexity is $\mathcal{O}(D^3 \log T)$.

## H   Background on Archambeau et al., 2007 (VDP)

In this section, we describe the variational diffusion process (VDP) smoothing algorithm proposed by Archambeau et al. [7]. The namesake for this algorithm is adopted from Verma et al. [33].

Rather than first approximating the continuous-time ELBO $\mathcal{L}_{\text{cont}}$, eq. (8) with $\mathcal{L}_{\text{approx}}$, eq. (9) and then maximizing with respect to the variational parameters, the VDP algorithm directly maximizes $\mathcal{L}_{\text{cont}}$. In particular, the VDP algorithm involves iteratively solving (continuous-time) ODEs. When implemented numerically, though, these ODEs are solved using a discrete-time scheme, such as Euler's method. Therefore, like Verma et al. [33] and SING, VDP only approximately maximizes the true, continuous-time ELBO.

Archambeau et al. [7] rewrite the ELBO $\mathcal{L}_{\text{cont}}$ in terms of $(\boldsymbol{A}(t), \boldsymbol{b}(t))_{0 \le t \le T_{\max}}$ as well as $(\boldsymbol{m}(t), \boldsymbol{S}(t))_{0 \le t \le T_{\max}}$ the marginal mean and covariance of $q(\boldsymbol{x})$ at each time $0 \le t \le T_{\max}$. These parameters are related by the Fokker-Planck equations for linear SDEs

$$\frac{d\boldsymbol{m}(t)}{dt} = \boldsymbol{A}(t)\boldsymbol{m}(t) + \boldsymbol{b}(t) \tag{21}$$

$$\frac{d\boldsymbol{S}(t)}{dt} = \boldsymbol{A}(t)\boldsymbol{S}(t) + \boldsymbol{S}(t)\boldsymbol{A}(t)^\top + \boldsymbol{\Sigma}. \tag{22}$$

To ensure that, at the optimum, equations eq. (21) and eq. (22) are satisfied, VDP introduces the Lagrange multipliers $(\boldsymbol{\lambda}(t))_{0 \le t \le T_{\max}}$ and $(\boldsymbol{\Psi}(t))_{0 \le t \le T_{\max}}$ and incorporate eq. (21) and eq. (22) into the objective

$$\mathcal{L}_{\text{cont}}((\boldsymbol{A}(t), \boldsymbol{b}(t), \boldsymbol{m}(t), \boldsymbol{S}(t))_{0 \le t \le T_{\max}}, \boldsymbol{\Theta}) + \int_0^{T_{\max}} \boldsymbol{\lambda}(t)^\top \left( \frac{d\boldsymbol{m}(t)}{dt} - \boldsymbol{A}(t)\boldsymbol{m}(t) - \boldsymbol{b}(t) \right) dt$$

$$+ \int_0^{T_{\max}} \text{tr} \left( \boldsymbol{\Psi}(t)^\top \left\{ \frac{d\boldsymbol{S}(t)}{dt} - \boldsymbol{A}(t)\boldsymbol{S}(t) - \boldsymbol{S}(t)\boldsymbol{A}(t) - \boldsymbol{\Sigma} \right\} \right) dt.$$

Performing an integration by parts yields

$$\mathcal{L}_{\text{cont}}((\boldsymbol{A}(t), \boldsymbol{b}(t), \boldsymbol{m}(t), \boldsymbol{S}(t))_{0 \le t \le T_{\max}}, \boldsymbol{\Theta}) - \int_0^{T_{\max}} \boldsymbol{\lambda}(t)^\top \left( \boldsymbol{A}(t)\boldsymbol{m}(t) + \boldsymbol{b}(t) \right) dt$$

$$- \int_0^{T_{\max}} \text{tr} \left( \boldsymbol{\Psi}(t)^\top \left\{ \boldsymbol{A}(t)\boldsymbol{S}(t) + \boldsymbol{S}(t)\boldsymbol{A}(t) + \boldsymbol{\Sigma} \right\} \right) dt - \int_0^{T_{\max}} \left( \text{tr} \left\{ \frac{d\boldsymbol{\Psi}(t)^\top}{dt} \boldsymbol{S}(t) \right\} + \frac{d\boldsymbol{\lambda}(t)^\top}{dt} \boldsymbol{m}(t) \right) dt$$

$$+ \text{tr}(\boldsymbol{\Psi}(T_{\max})^\top \boldsymbol{S}(T_{\max})) - \text{tr}(\boldsymbol{\Psi}(0)^\top \boldsymbol{S}(0)) + \boldsymbol{\lambda}(T_{\max})^\top \boldsymbol{m}(T_{\max}) - \boldsymbol{\lambda}(0)^\top \boldsymbol{m}(0). \tag{23}$$

The authors proceed by performing coordinate ascent on this objective with respect to $(\boldsymbol{A}(t), \boldsymbol{b}(t), \boldsymbol{m}(t), \boldsymbol{S}(t), \boldsymbol{\lambda}(t), \boldsymbol{\Psi}(t))_{0 \le t \le T_{\max}}$. However, when taking the (variational) derivatives with respect to the functions $\boldsymbol{A}(t)$, $\boldsymbol{b}(t)$, $\boldsymbol{m}(t)$, and $\boldsymbol{S}(t)$, the dependence of $\boldsymbol{A}(t)$ and $\boldsymbol{b}(t)$ on $\boldsymbol{m}(t)$ and $\boldsymbol{S}(t)$, and vice versa, is ignored.

Taking the derivative of eq. (23) with respect to $\boldsymbol{m}(t)$ and $\boldsymbol{S}(t)$ yields the updates for the Lagrange multipliers

$$\frac{d\boldsymbol{\Psi}(t)}{dt} = -\boldsymbol{\Psi}(t)\boldsymbol{A}(t) - (\boldsymbol{A}(t))^\top \boldsymbol{\Psi}(t) + \frac{d}{d\boldsymbol{S}(t)}(-\text{D}_{\text{KL}}\left(q \parallel p\right)), \quad \boldsymbol{\Psi}(T_{\max}) = \boldsymbol{0} \tag{24}$$

$$\frac{d\boldsymbol{\lambda}(t)}{dt} = -\boldsymbol{A}(t)^\top \boldsymbol{\lambda}(t) + \frac{d}{d\boldsymbol{m}(t)}(-\text{D}_{\text{KL}}\left(q \parallel p\right)), \quad \boldsymbol{\lambda}(T_{\max}) = \boldsymbol{0} \tag{25}$$

subject to the jump conditions at observation times $\{t_i\}_{i=1}^T$

$$\boldsymbol{\Psi}(t_i^+) = \boldsymbol{\Psi}(t_i) + \frac{d}{d\boldsymbol{S}(t_i)}\mathbb{E}_q \log p(\boldsymbol{y}(t_i)|\boldsymbol{x}(t_i)) \tag{26}$$

$$\boldsymbol{\lambda}(t_i^+) = \boldsymbol{\lambda}(t_i) + \frac{d}{d\boldsymbol{m}(t_i)}\mathbb{E}_q \log p(\boldsymbol{y}(t_i)|\boldsymbol{x}(t_i)). \tag{27}$$

Taking the derivative of eq. (23) with respect to $\boldsymbol{A}(t)$ and $\boldsymbol{b}(t)$ yields the updates

$$\boldsymbol{A}(t) = \mathbb{E}_{q(\boldsymbol{x}(t))}\left[\frac{d\boldsymbol{f}(\boldsymbol{x}|\boldsymbol{\Theta})}{d\boldsymbol{x}}\right] - 2\boldsymbol{\Sigma}\boldsymbol{\Psi}(t) \tag{28}$$

$$\boldsymbol{b}(t) = \mathbb{E}_{q(\boldsymbol{x}(t))}[\boldsymbol{f}(\boldsymbol{x}|\boldsymbol{\Theta})] - \boldsymbol{A}(t)\boldsymbol{m}(t) - \boldsymbol{\Sigma}\boldsymbol{\lambda}(t). \tag{29}$$

Finally, for $p(\boldsymbol{x}(0)) = \mathcal{N}(\boldsymbol{\nu}, \boldsymbol{V})$, taking the derivative of eq. (23) with respect to $\boldsymbol{\nu}$ and $\boldsymbol{V}$ yields

$$\boldsymbol{m}(0) = \boldsymbol{\nu} - \boldsymbol{V}\boldsymbol{\lambda}(0) \tag{30}$$

$$\boldsymbol{S}(0) = (2\boldsymbol{\Psi}(0) + \boldsymbol{V}^{-1})^{-1}. \tag{31}$$

Altogether, the VDP variational inference algorithm alternates between the following steps, until convergence

1. Solve for $(\boldsymbol{m}(t), \boldsymbol{S}(t))_{0 \leq t \leq T_{\max}}$ according to eq. (21) and eq. (22).
2. Update $(\boldsymbol{\lambda}(t), \boldsymbol{\Psi}(t))_{0 \leq t \leq T_{\max}}$, starting from $\boldsymbol{\lambda}(T_{\max}) = \boldsymbol{0}$ and $\boldsymbol{\Psi}(T_{\max}) = \boldsymbol{0}$, according to eq. (25) and eq. (24) and taking into account the jump conditions eq. (27) and eq. (26) at observation times $\{t_i\}_{i=1}^T$.
3. Update $(\boldsymbol{A}(t), \boldsymbol{b}(t))_{0 \leq t \leq T_{\max}}$ according to eq. (28) and eq. (29).
4. Update $\boldsymbol{m}(0)$ and $\boldsymbol{S}(0)$ according to eq. (30) and eq. (31).

For stability, Archambeau et al. [7, 8] propose replacing the update for $(\boldsymbol{A}(t), \boldsymbol{b}(t))_{0 \leq t \leq T_{\max}}$ at iteration $\ell$ with the "soft" updates

$$\begin{aligned} \boldsymbol{A}^{(\ell+1)}(t) &= \boldsymbol{A}^{(\ell)}(t) + \omega(\widetilde{\boldsymbol{A}}^{(\ell)}(t) - \boldsymbol{A}^{(\ell)}(t)) \\ \boldsymbol{b}^{(\ell+1)}(t) &= \boldsymbol{b}^{(\ell)}(t) + \omega(\widetilde{\boldsymbol{b}}^{(\ell)}(t) - \boldsymbol{b}^{(\ell)}(t)), \end{aligned} \tag{32}$$

where $\widetilde{\boldsymbol{A}}^{(\ell)}(t)$ and $\widetilde{\boldsymbol{b}}^{(\ell)}(t)$ are defined as in eq. (28) and eq. (29). In our experiments, we observe that choosing $\omega$ to be small (e.g., $10^{-3}$) is often necessary to ensure convergence of VDP.

Unlike Verma et al. [33] and SING, VDP does not benefit from convergence guarantees. Indeed in Appendix J, we show that even in the setting where the prior SDE is linear and the observation model is Gaussian, VDP will not recover the true posterior in one step.

# I  Comparison of SING to Verma et al., 2024

## I.1  Overview of Verma et al., 2024

In this section, we outline Verma et al. [33], the work upon which SING is built, and discuss key differences between the two variational EM algorithms.

Like SING, Verma et al. [33] maximizes the approximate ELBO $\mathcal{L}_{\text{approx}}$ from eq. (9) with respect the variational parameters $(\boldsymbol{A}(t), \boldsymbol{b}(t))_{0 \leq t \leq T_{\max}}$ as well as the model parameters $\boldsymbol{\Theta}$. To accomplish this, they perform a change-of-measure

$$\tilde{p}(\boldsymbol{x}_{0:T}|\boldsymbol{\Theta}) = p_L(\boldsymbol{x}_{0:T})\frac{\tilde{p}(\boldsymbol{x}_{0:T}|\boldsymbol{\Theta})}{p_L(\boldsymbol{x}_{0:T})},$$

where $p_L(\boldsymbol{x}_{0:T}) = q(\boldsymbol{x}_{0:T}|\boldsymbol{\eta}_L)$ belongs to the same exponential family as the variational posterior and has natural parameters $\boldsymbol{\eta}_L = (\boldsymbol{h}^L, \boldsymbol{J}^L, \boldsymbol{L}^L)$. Suppose $q(\boldsymbol{x}_{0:T}|\boldsymbol{\eta})$ is initialized at $\boldsymbol{\eta}^{(0)} = \boldsymbol{\eta}_L$. Then the corresponding NGVI update steps can be written as

$$\begin{aligned} \tilde{\boldsymbol{h}}_i^{(j+1)} &= (1-\rho)\tilde{\boldsymbol{h}}_i^{(j)} + \rho\nabla_{\boldsymbol{\mu}_{i,1}}\mathbb{E}_{q^{(j)}}[\log p(\boldsymbol{y}_i|\boldsymbol{x}_i)\delta_i + \Delta_i V(\boldsymbol{x}_i, \boldsymbol{x}_{i+1}) + \Delta_i V(\boldsymbol{x}_{i-1}, \boldsymbol{x}_i)] \\ \tilde{\boldsymbol{J}}_i^{(j+1)} &= (1-\rho)\tilde{\boldsymbol{J}}_i^{(j)} + \rho\nabla_{\boldsymbol{\mu}_{i,2}}\mathbb{E}_{q^{(j)}}[\log p(\boldsymbol{y}_i|\boldsymbol{x}_i)\delta_i + \Delta_i V(\boldsymbol{x}_i, \boldsymbol{x}_{i+1}) + \Delta_i V(\boldsymbol{x}_{i-1}, \boldsymbol{x}_i)] \\ \tilde{\boldsymbol{L}}_i^{(j+1)} &= (1-\rho)\tilde{\boldsymbol{L}}_i^{(t)} + \rho\nabla_{\boldsymbol{\mu}_{i,3}}\mathbb{E}_{q^{(j)}}[\Delta_i V(\boldsymbol{x}_i, \boldsymbol{x}_{i+1})] \\ \boldsymbol{h}_i^{(j+1)} &= \tilde{\boldsymbol{h}}_i^{(j+1)} + \boldsymbol{h}^L, \quad \boldsymbol{J}_i^{(j+1)} = \tilde{\boldsymbol{J}}_i^{(j+1)} + \boldsymbol{J}^L, \quad \boldsymbol{L}_i^{(j+1)} = \tilde{\boldsymbol{L}}_i^{(j+1)} + \boldsymbol{L}^L \end{aligned} \tag{33}$$

Here, $\Delta_i V(\boldsymbol{x}_i, \boldsymbol{x}_{i+1}) = \log\left(\tilde{p}(\boldsymbol{x}_{i+1}|\boldsymbol{x}_i, \boldsymbol{\Theta})/p_L(\boldsymbol{x}_{i+1}|\boldsymbol{x}_i)\right), 0 \le i \le T-1$ is the log ratio of the transition probabilities from $\boldsymbol{x}_i$ to $\boldsymbol{x}_{i+1}$ under $\tilde{p}$ and $p_L$. The NGVI updates that appear in Verma et al. [33] are written in this unusual form to separate the contribution of $p_L$, which belongs to the same family as $q$, from the other terms that appear in the ELBO.

The authors propose alternating between (i) performing numerous NGVI updates (33) and (ii) updating $p_L$ at iteration $j$ according to a statistical linearization step. Specifically, statistical linearization of the prior drift computes the best linear approximation to the nonlinear prior drift under the current variational posterior according to

$$\boldsymbol{A}_i^L, \boldsymbol{b}_i^L = \arg\min \mathbb{E}_{q(\boldsymbol{x}_i|\boldsymbol{\eta}^{(j)})} \|\boldsymbol{A}_i^L \boldsymbol{x}_i + \boldsymbol{b}_i^L - \boldsymbol{f}(\boldsymbol{x}_i|\boldsymbol{\Theta})\|_2^2$$
$$\implies \boldsymbol{A}_i^L = \mathbb{E}_{q(\boldsymbol{x}_i|\boldsymbol{\eta}^{(\ell)})}\left[\frac{d\boldsymbol{f}(\boldsymbol{x}_i|\boldsymbol{\Theta})}{d\boldsymbol{x}}\right], \quad \boldsymbol{b}_i^L = \mathbb{E}_{q(\boldsymbol{x}_i|\boldsymbol{\eta}^{(j)})}[\boldsymbol{f}(\boldsymbol{x}_i|\boldsymbol{\Theta})] - \boldsymbol{A}_i^L \boldsymbol{m}_i, \quad i = 0, \dots T-1. \tag{34}$$

By the log normalizer computation discussed in Appendix G, one can use $(\boldsymbol{A}_i^L, \boldsymbol{b}_i^L)_{i=1}^{T-1}$ to obtain a new set of natural parameters $\boldsymbol{\eta}^{L,\text{new}} = (\boldsymbol{h}^{L,\text{new}}, \boldsymbol{J}^{L,\text{new}}, \boldsymbol{L}^{L,\text{new}})$. One then updates the natural parameters by setting

$$\tilde{\boldsymbol{h}}_i^{(j)} \leftarrow \tilde{\boldsymbol{h}}_i^{(j)} + (\boldsymbol{h}_i^L - \boldsymbol{h}_i^{L,\text{new}}), \ \tilde{\boldsymbol{J}}_i^{(j)} \leftarrow \tilde{\boldsymbol{J}}_i^{(j)} + (\boldsymbol{J}_i^L - \boldsymbol{J}_i^{L,\text{new}}), \ \tilde{\boldsymbol{L}}_i^{(j)} \leftarrow \tilde{\boldsymbol{L}}_i^{(j)} + (\boldsymbol{L}_i^L - \boldsymbol{L}_i^{L,\text{new}}) \tag{35}$$

and lastly $\boldsymbol{\eta}^L \leftarrow \boldsymbol{\eta}^{L,\text{new}}$. The updates eq. (35) ensure that updating the parameters $\boldsymbol{\eta}^L$ of $p_L$ does not change the parameters of the $q$.

Importantly, the statistical linearization step appearing in Verma et al. [33] does not matter for inference, since it is apparent that by "undoing" the change-of-measure, one can rewrite the NGVI update (33) to exactly match the SING update (11). In fact, we point out that repeatedly performing statistical linearization, as is suggested by Verma et al. [33], is unnecessary: one will obtain the same $\boldsymbol{\eta}^L$ by performing only a single statistical linearization step at the end of inference.

The reason for introducing $p_L$ is that the authors contend that it aids in hyperparameter learning, insofar as it allows one to couple the E and M-steps via $\boldsymbol{\eta}_L$, which from eq. (34), evidently depends on $\boldsymbol{\Theta}$. In other words, during the M-step Verma et al. [33] solve

$$\arg\max_{\boldsymbol{\Theta}} \mathcal{L}(\boldsymbol{\eta}_L(\boldsymbol{\Theta}) + \tilde{\boldsymbol{\eta}}, \boldsymbol{\Theta}), \quad \tilde{\boldsymbol{\eta}} = (\tilde{\boldsymbol{h}}, \tilde{\boldsymbol{J}}, \tilde{\boldsymbol{L}}).$$

This idea is adopted from Adam et al. [60]. One can imagine there are many functions $\vartheta : \mathbb{R}^{|\boldsymbol{\Theta}|} \to \boldsymbol{\Xi}$ that couple the E and M-steps by writing $\boldsymbol{\eta} = \vartheta(\boldsymbol{\Theta}) + (\boldsymbol{\eta} - \vartheta(\boldsymbol{\Theta}))$. The authors argue that defining $\vartheta$ as in eq. (34) is desirable insofar as optimizing $\boldsymbol{\eta} - \vartheta(\boldsymbol{\Theta})$ with respect to $\boldsymbol{\eta}$ for $\boldsymbol{\Theta}$ fixed is less dependent on $\boldsymbol{\Theta}$ than for alternate $\vartheta$. One issue with this argument is that, unlike Verma et al. [33], we are also optimizing the parameters of the likelihood $p(\boldsymbol{y}|\boldsymbol{x}, \boldsymbol{\Theta})$, and so it is not evident that statistical linearization of the prior eq. (34) best couples the E and M-steps.

## I.2 Differences between SING and Verma et al., 2024

SING aims to simplify and improve upon the algorithm presented by Verma et al. [33]. First, Verma et al. [33] does not address how to compute the term

$$\mathbb{E}_q V(\boldsymbol{x}_i, \boldsymbol{x}_{i+1}) = \frac{1}{\Delta_i} \mathbb{E}_q \log \frac{p(\boldsymbol{x}_{i+1}|\boldsymbol{x}_i, \boldsymbol{\Theta})}{p_L(\boldsymbol{x}_{i+1}|\boldsymbol{x}_i)} = \mathbb{E}_q\left\{ -\frac{1}{2}\|\boldsymbol{A}_t^L \boldsymbol{x}_i + \boldsymbol{b}_i^L - \boldsymbol{f}(\boldsymbol{x}_i|\boldsymbol{\Theta})\|_{\boldsymbol{\Sigma}^{-1}}^2 \right.$$
$$\left. + \left\langle \frac{\boldsymbol{x}_{i+1} - \boldsymbol{x}_i}{\Delta_i} - (\boldsymbol{A}_t^L \boldsymbol{x}_i + \boldsymbol{b}_i^L), \boldsymbol{f}(\boldsymbol{x}_i|\boldsymbol{\Theta}) - (\boldsymbol{A}_t^L \boldsymbol{x}_i + \boldsymbol{b}_i^L) \right\rangle_{\boldsymbol{\Sigma}^{-1}} \right\},$$

which, a priori, requires approximating expectations in $2D$ dimensions. In Appendix F, we explicitly discuss how to compute analogous quantities for SING using $D$-dimensional expectations, which makes our proposed inference algorithm both faster and more memory-efficient. Moreover, as we previously discussed, changing the base measure from $p$ to $p_L$ is unnecessary for inference, and so we remove it. We do not explore coupling the E and M-steps with SING.

Additionally, the inference algorithm proposed by Verma et al. [33] requires, for each NGVI step, sequentially converting between natural and mean parameters, which has linear time complexity and can be computationally expensive when the number of steps is large. In Appendix G.2, we propose

a parallelized algorithm for computing the log normalizer of the variational family, which leads to significant speedup in the runtime for inference (Figure 5).

Using the sparse variational GP framework from Titsias [36], we propose an extension of SING, SING-GP, to the Bayesian model that includes a Gaussian process prior on the latent SDE drift. Previous works that perform approximate inference in GP-SDE models, including Duncker et al. [13] and Hu et al. [14], perform inference on the latent variables $(\boldsymbol{x}(t))_{0 \leq t \leq T_{\max}}$ using VDP, and hence are inherently slow to converge and unstable for large step sizes $\Delta_i$ (Figure 2 and Figure 3).

## J  VDP and SING in the conjugate setting

**Proposition 2:** Suppose $\boldsymbol{f}(\boldsymbol{x}, t | \boldsymbol{\Theta}) = \boldsymbol{A}^{\text{prior}}(t)\boldsymbol{x}(t) + \boldsymbol{b}^{\text{prior}}(t)$ and $p(\boldsymbol{y}(t_i) | \boldsymbol{x}(t_i), \boldsymbol{\Theta}) = \mathcal{N}(\boldsymbol{C}\boldsymbol{x}(t_i) + \boldsymbol{d}, \boldsymbol{R})$. Then, when replacing $\tilde{p}$ with $p$ in $\mathcal{L}_{\text{approx}}$, SING performs exact inference on $\boldsymbol{\tau}$ when $\rho = 1$. Specifically, after one full natural gradient step, $q(\boldsymbol{x}_{0:T} | \boldsymbol{\eta}^{(1)}) = p(\boldsymbol{x}_{0:T} | \mathcal{D}, \boldsymbol{\Theta})$.

*Proof.* First, we point out that we are able to replace $\tilde{p}$ with $p$ in $\mathcal{L}_{\text{approx}}$ since $\boldsymbol{f}$ is linear. From here, we write

$p(\boldsymbol{x}_{0:T})p(\boldsymbol{y} | \boldsymbol{x}_{0:T})$

$$\propto \exp\left(-\frac{1}{2}\sum_{i=0}^{T} \boldsymbol{x}_i^\top \boldsymbol{J}_i^{\text{prior}} \boldsymbol{x}_i - \sum_{i=0}^{T-1} \boldsymbol{x}_{i+1}^\top \boldsymbol{L}_i^{\text{prior}} \boldsymbol{x}_i + \sum_{i=0}^{T} (\boldsymbol{h}_i^{\text{prior}})^\top \boldsymbol{x}_i - \frac{1}{2}\sum_{i=0}^{n} \|\boldsymbol{y}(t_i) - \boldsymbol{C}\boldsymbol{x}(t_i) - \boldsymbol{d}\|_{\boldsymbol{R}^{-1}}^2\right)$$

$$\propto \exp\left(-\frac{1}{2}\sum_{i=0}^{T} \boldsymbol{x}_i^\top (\boldsymbol{J}_i^{\text{prior}} + \delta_i \boldsymbol{C}^\top \boldsymbol{R}^{-1} \boldsymbol{C})\boldsymbol{x}_i - \sum_{i=0}^{T-1} \boldsymbol{x}_{i+1}^\top \boldsymbol{L}_i^{\text{prior}} \boldsymbol{x}_i + \sum_{i=0}^{T} (\boldsymbol{h}_i^{\text{prior}} + \delta_i \boldsymbol{C}^\top \boldsymbol{R}^{-1}\{\boldsymbol{y}(\tau_i) - \boldsymbol{d}\})^\top \boldsymbol{x}_i\right)$$

where $\delta_i$ is the indicator denoting whether an observation occurs at $\tau_i$. It is straightforward to see that this coincides with the posterior over paths $p(\boldsymbol{x} | \mathcal{D})$, marginalized to the grid $\boldsymbol{\tau}$, written $p(\boldsymbol{x}_{0:T} | \mathcal{D})$. Hence, $p(\boldsymbol{x}_{0:T} | \mathcal{D})$ is Gaussian with natural parameters $\boldsymbol{\eta}^{\text{true}} = \left[\{\boldsymbol{h}_i^{\text{prior}} + \delta_i \boldsymbol{C}^\top \boldsymbol{R}^{-1}(\boldsymbol{y}(\tau_i) - \boldsymbol{d}), -\frac{1}{2}(\boldsymbol{J}_i^{\text{prior}} + \delta_i \boldsymbol{C}^\top \boldsymbol{R}^{-1} \boldsymbol{C})\}_{i=0}^{T}, \{-\boldsymbol{L}_i^{\text{prior}}\}_{i=0}^{T-1}\right]$.

If we write $\mathcal{L}_{\text{approx}}$ with $p$ substituted for $\tilde{p}$ as

$$\mathcal{L}_{\text{approx}}(\boldsymbol{\eta}, \boldsymbol{\Theta}) = \mathbb{E}_q \log \frac{p(\boldsymbol{x}_{1:T})p(\boldsymbol{y} | \boldsymbol{x}_{1:T})}{q(\boldsymbol{x}_{1:T})}$$

then the NGVI update eq. (11) with $\rho = 1$ becomes

$$\boldsymbol{\eta}^{(1)} = \boldsymbol{\eta}^{(0)} + \nabla_{\boldsymbol{\mu}} \mathcal{L}_{\text{approx}}(\boldsymbol{\eta}^{(0)}, \boldsymbol{\Theta}) \stackrel{(i)}{=} \boldsymbol{\eta}^{(0)} + (\boldsymbol{\eta}^{\text{true}} - \boldsymbol{\eta}^{(0)}) = \boldsymbol{\eta}^{\text{true}}.$$

In equality (i), we used the identity from Appendix B.2, Lemma 4 for the natural gradient of the KL divergence between two distributions belonging to the same exponential family. $\square$

**Proposition 3:** Consider the same setting as in Proposition 2. Then VDP algorithm described in Appendix H does not recover $p(\boldsymbol{x} | \mathcal{D})$ in one step.

*Proof.* For simplicity of presentation, we will assume $\boldsymbol{\Sigma} = \boldsymbol{I}$.

Suppose we have $(\boldsymbol{m}(t), \boldsymbol{S}(t))_{0 \leq t \leq T_{\max}}$, the marginal means of covariances of the variational posterior $q$, as well as $(\boldsymbol{A}(t), \boldsymbol{b}(t))_{0 \leq t \leq T_{\max}}$ that specify the drift of the variational posterior. After the first step of the Archambeau et al. [7] VDP algorithm, these are consistent.

In order to analyze VDP in this setting, we first compute the (variational) derivative of each term in the ELBO $\mathcal{L}_{\text{cont}}$ with respect to $\boldsymbol{m}(t)$ and $\boldsymbol{S}(t)$. To this end,

$$\frac{1}{2}\mathbb{E}_q \|\{\boldsymbol{A}^{\text{prior}}(t)\boldsymbol{x}(t) + \boldsymbol{b}^{\text{prior}}(t)\} - \{\boldsymbol{A}(t)\boldsymbol{x}(t) + \boldsymbol{b}(t)\}\|_2^2$$

$$= \mathbb{E}_q \left\{\frac{1}{2}\boldsymbol{x}(t)^\top (\boldsymbol{A}^{\text{prior}}(t) - \boldsymbol{A}(t))^\top (\boldsymbol{A}^{\text{prior}}(t) - \boldsymbol{A}(t))\boldsymbol{x}(t) + \boldsymbol{x}(t)^\top (\boldsymbol{A}^{\text{prior}}(t) - \boldsymbol{A}(t))^\top (\boldsymbol{b}^{\text{prior}}(t) - \boldsymbol{b}(t))\right\} + \text{const}$$

$$= \frac{1}{2}\text{tr}\left((\boldsymbol{A}^{\text{prior}}(t) - \boldsymbol{A}(t))^\top (\boldsymbol{A}^{\text{prior}}(t) - \boldsymbol{A}(t))\{\boldsymbol{S}(t) + \boldsymbol{m}(t)\boldsymbol{m}(t)^\top\}\right)$$
$$+ \boldsymbol{m}(t)^\top (\boldsymbol{A}^{\text{prior}}(t) - \boldsymbol{A}(t))^\top (\boldsymbol{b}^{\text{prior}}(t) - \boldsymbol{b}(t)) + \text{const}$$

as well as

$$\mathbb{E}_q \log \mathcal{N}(\boldsymbol{y}(t_i)|\boldsymbol{C}\boldsymbol{x}(t_i) + \boldsymbol{d}, \boldsymbol{R})$$

$$=\mathbb{E}_q\left\{-\frac{1}{2}(\boldsymbol{y}(t_i) - \{\boldsymbol{C}\boldsymbol{x}(t_i) + \boldsymbol{d}\})^\top \boldsymbol{R}^{-1}(\boldsymbol{y}(t_i) - \{\boldsymbol{C}\boldsymbol{x}(t_i) + \boldsymbol{d}\})\right\} + \text{const}$$

$$=\mathbb{E}_q\left\{-\frac{1}{2}\boldsymbol{x}(t_i)^\top \boldsymbol{C}^\top \boldsymbol{R}^{-1}\boldsymbol{C}\boldsymbol{x}(t_i) + \boldsymbol{x}(t_i)^\top \boldsymbol{C}^\top \boldsymbol{R}^{-1}\boldsymbol{y}(t_i) - \boldsymbol{x}(t_i)^\top \boldsymbol{C}^\top \boldsymbol{R}^{-1}\boldsymbol{d}\right\} + \text{const}$$

$$=-\frac{1}{2}\text{tr}\left(\boldsymbol{C}^\top \boldsymbol{R}^{-1}\boldsymbol{C}(\boldsymbol{S}(t_i) + \boldsymbol{m}(t_i)\boldsymbol{m}(t_i)^\top)\right) + \boldsymbol{m}(t_i)^\top \boldsymbol{C}^\top \boldsymbol{R}^{-1}(\boldsymbol{y}(t_i) - \boldsymbol{d}) + \text{const}$$

where const represents a generic constant with respect to $(\boldsymbol{m}(t), \boldsymbol{S}(t))_{0 \leq t \leq T_{\max}}$.

From Appendix H, these imply the following updates for the Lagrange multipliers

$$\frac{d\boldsymbol{\lambda}(t)}{dt} = -\boldsymbol{A}(t)^\top \boldsymbol{\lambda}(t) - (\boldsymbol{A}^{\text{prior}}(t) - \boldsymbol{A}(t))^\top (\boldsymbol{A}^{\text{prior}}(t) - \boldsymbol{A}(t))\boldsymbol{m}(t)$$
$$- (\boldsymbol{A}^{\text{prior}}(t) - \boldsymbol{A}(t))^\top (\boldsymbol{b}^{\text{prior}}(t) - \boldsymbol{b}(t))$$
$$\frac{d\boldsymbol{\Psi}(t)}{dt} = -\boldsymbol{A}(t)^\top \boldsymbol{\Psi}(t) - \boldsymbol{\Psi}(t)\boldsymbol{A}(t) - \frac{1}{2}(\boldsymbol{A}^{\text{prior}}(t) - \boldsymbol{A}(t))^\top (\boldsymbol{A}^{\text{prior}}(t) - \boldsymbol{A}(t))$$

with the jump conditions at times $\{t_i\}_{i=1}^n$

$$\boldsymbol{\lambda}(t_i^+) = \boldsymbol{\lambda}(t_i) + \boldsymbol{C}^\top \boldsymbol{R}^{-1}(\boldsymbol{y}(t_i) - \boldsymbol{d}) - \boldsymbol{C}^\top \boldsymbol{R}^{-1}\boldsymbol{C}\boldsymbol{m}(t_i)$$
$$\boldsymbol{\Psi}(t_i^+) = \boldsymbol{\Psi}(t_i) - \frac{1}{2}\boldsymbol{C}^\top \boldsymbol{R}^{-1}\boldsymbol{C}.$$

WLOG, we assume $t_i < T_{\max}$ for $1 \leq i \leq n$ and set $\boldsymbol{\lambda}(T_{\max}) = \boldsymbol{0}$, $\boldsymbol{\Psi}(T_{\max}) = \boldsymbol{0}$ (otherwise, the terminal condition is the same as the jump condition at time $T_{\max}$). One can check these match the updates appearing on Archambeau et al. [7], page 7. The last step in the VDP algorithm is to update the variational parameters according to

$$\boldsymbol{A}(t) = \boldsymbol{A}^{\text{prior}}(t) - 2\boldsymbol{\Psi}(t)$$
$$\boldsymbol{b}(t) = \boldsymbol{A}^{\text{prior}}(t)\boldsymbol{m}(t) + \boldsymbol{b}^{\text{prior}}(t) - \boldsymbol{A}(t)\boldsymbol{m}(t) - \boldsymbol{\lambda}(t) = (\boldsymbol{A}^{\text{prior}}(t) - \boldsymbol{A}(t))\boldsymbol{m}(t) + \boldsymbol{b}^{\text{prior}}(t) - \boldsymbol{\lambda}(t)$$

Next, we determine the analytical form of the posterior drift in this setting. In particular, the Kushner-Stratonovich-Pardoux equations [29–31] give a closed-form expression for the SDE satisfied by the posterior path measure

$$q : d\boldsymbol{x}(t) = \left\{\boldsymbol{A}^{\text{prior}}\boldsymbol{x}_t + \boldsymbol{b}^{\text{prior}} + \nabla_{\boldsymbol{x}} \log p(\boldsymbol{y}_{\geq t}|\boldsymbol{x}(t))\right\} dt + d\boldsymbol{w}(t).$$

where $p(\boldsymbol{y}_{\geq t}|\boldsymbol{x}(t))$ represents the conditional probability of observations $\boldsymbol{y}(t_i)$, $t_i \geq t$ conditioned on $\boldsymbol{x}(t) = \boldsymbol{x}$. Observe also that $p(\boldsymbol{y}_{\geq t}|\boldsymbol{x}(t))$ will have jumps occurring at observation times.

In general, we cannot compute $p(\boldsymbol{y}_{\geq t}|\boldsymbol{x}(t))$ in closed-form, and hence do not have an analytical expression for the posterior drift. In the linear case, though, we can compute this explicitly. In order to do so, we make a couple of observations. First, for $t_{i-1} < t \leq t_i$, we see that by the Markov property

$$p(\boldsymbol{y}_{\geq t}|\boldsymbol{x}(t)) \propto p(\boldsymbol{x}(t_i)|\boldsymbol{x}(t))\left(\prod_{k=i}^{n-1} p(\boldsymbol{x}(t_{k+1})|\boldsymbol{x}(t_k))p(\boldsymbol{y}(t_k)|\boldsymbol{x}(t_k))\right) p(\boldsymbol{y}(t_n)|\boldsymbol{x}(t_n)).$$

Since each of these factors is Gaussian, $p(\boldsymbol{y}_{\geq t}|\boldsymbol{x}(t))$ will again be Gaussian. Let us write

$$p(\boldsymbol{y}_{\geq t}|\boldsymbol{x}(t)) = \exp\left\{-\frac{1}{2}\boldsymbol{x}(t)^\top \boldsymbol{G}(t)\boldsymbol{x}(t) + \boldsymbol{g}(t)^\top \boldsymbol{x}(t) + c(t)\right\}, \quad t_{i-1} < t \leq t_i.$$

Second, $\psi(\boldsymbol{x}, t) = p(\boldsymbol{y}_{\geq t}|\boldsymbol{x}(t))$ satisfies the Backward Kolmogorov Equation (BKE)

$$\frac{\partial}{\partial t}\psi(\boldsymbol{x}, t) + (\boldsymbol{A}^{\text{prior}}(t)\boldsymbol{x}(t) + \boldsymbol{b}^{\text{prior}}(t))^\top \nabla_{\boldsymbol{x}}\psi(\boldsymbol{x}, t) + \frac{1}{2}\Delta_{\boldsymbol{x}}\psi(\boldsymbol{x}, t) = 0$$

in between observation times $t_{i-1} < t \le t_i$. Plugging

$$\psi(\boldsymbol{x}, t) = \exp\left\{-\frac{1}{2}\boldsymbol{x}^\top \boldsymbol{G}(t)\boldsymbol{x} + \boldsymbol{g}(t)^\top \boldsymbol{x} + c(t)\right\}$$

into the BKE, we get

$$\psi(\boldsymbol{x}, t)(\boldsymbol{A}^{\text{prior}}(t)\boldsymbol{x} + \boldsymbol{b}^{\text{prior}}(t))^\top(\boldsymbol{g}(t) - \boldsymbol{G}(t)\boldsymbol{x}) + \psi(\boldsymbol{x}, t)\frac{1}{2}\left\{(\boldsymbol{g}(t) - \boldsymbol{G}(t)\boldsymbol{x})^\top(\boldsymbol{g}(t) - \boldsymbol{G}(t)\boldsymbol{x}) - \text{tr}(\boldsymbol{G}(t))\right\}$$

$$= -\psi(\boldsymbol{x}, t)\left(-\frac{1}{2}\boldsymbol{x}^\top\frac{d}{dt}\boldsymbol{G}(t)\boldsymbol{x} + \frac{d}{dt}\boldsymbol{g}(t)^\top\boldsymbol{x} + \frac{d}{dt}c(t)\right)$$

Simplifying,

$$(\boldsymbol{A}^{\text{prior}}(t)\boldsymbol{x} + \boldsymbol{b}^{\text{prior}}(t))^\top(\boldsymbol{G}(t)\boldsymbol{x} - \boldsymbol{g}(t)) + \frac{1}{2}\left\{(\boldsymbol{G}(t)\boldsymbol{x} - \boldsymbol{g}(t))^\top(\boldsymbol{g}(t) - \boldsymbol{G}(t)\boldsymbol{x}) + \text{tr}(\boldsymbol{G}(t))\right\}$$

$$= \left(-\frac{1}{2}\boldsymbol{x}^\top\frac{d}{dt}\boldsymbol{G}(t)\boldsymbol{x} + \frac{d}{dt}\boldsymbol{g}(t)^\top\boldsymbol{x} + \frac{d}{dt}c(t)\right)$$

$$\implies \frac{d}{dt}\boldsymbol{G}(t) = -(\boldsymbol{A}^{\text{prior}}(t))^\top\boldsymbol{G}(t) - \boldsymbol{G}(t)\boldsymbol{A}^{\text{prior}}(t) + \boldsymbol{G}(t)^\top\boldsymbol{G}(t)$$

$$\implies \frac{d}{dt}\boldsymbol{g}(t) = -(\boldsymbol{A}^{\text{prior}}(t))^\top\boldsymbol{g}(t) + \boldsymbol{G}(t)^\top\boldsymbol{b}^{\text{prior}}(t) + \boldsymbol{G}(t)^\top\boldsymbol{g}(t)$$

where we used the facts that $2\boldsymbol{x}^\top(\boldsymbol{A}^{\text{prior}}(t))^\top\boldsymbol{G}(t)\boldsymbol{x} = \boldsymbol{x}^\top(\boldsymbol{A}^{\text{prior}}(t))^\top\boldsymbol{G}(t)\boldsymbol{x} + \boldsymbol{x}^\top\boldsymbol{G}(t)^\top\boldsymbol{A}^{\text{prior}}\boldsymbol{x}$ and that if $\boldsymbol{G}(T_{\max})$ is symmetric, then $(\boldsymbol{G}(t))_{0 \le t \le T_{\max}}$ will be symmetric.

In other words, we have shown that $p(\boldsymbol{y}_{\ge t}|\boldsymbol{x}(t))$ satisfies the Riccati equation

$$\frac{d}{dt}\boldsymbol{G}(t) = -(\boldsymbol{A}^{\text{prior}}(t))^\top\boldsymbol{G}(t) - \boldsymbol{G}_t\boldsymbol{A}^{\text{prior}}(t) + \boldsymbol{G}(t)^\top\boldsymbol{G}(t)$$

$$\frac{d}{dt}\boldsymbol{g}(t) = -(\boldsymbol{A}^{\text{prior}}(t))^\top\boldsymbol{g}(t) + \boldsymbol{G}(t)^\top\boldsymbol{b}^{\text{prior}} + \boldsymbol{G}(t)^\top\boldsymbol{g}(t)$$

with jump conditions

$$\boldsymbol{G}(t_i) = \boldsymbol{G}(t_i^+) + \boldsymbol{C}^\top\boldsymbol{R}^{-1}\boldsymbol{C}$$
$$\boldsymbol{g}(t_i) = \boldsymbol{g}(t_i^+) + \boldsymbol{C}^\top\boldsymbol{R}^{-1}(\boldsymbol{y}(t_i) - \boldsymbol{d})$$

and initial conditions $\boldsymbol{G}(T_{\max}) = \boldsymbol{0}$, $\boldsymbol{g}(T_{\max}) = \boldsymbol{0}$. The posterior drift is then given by

$$\boldsymbol{A}^{\text{prior}}(t)\boldsymbol{x}(t) + \boldsymbol{b}^{\text{prior}}(t) + \nabla_{\boldsymbol{x}}\log p(\boldsymbol{y}_{\ge t}|\boldsymbol{x}(t)) = (\boldsymbol{A}^{\text{prior}}(t) - \boldsymbol{G}(t))\boldsymbol{x}(t) + (\boldsymbol{b}^{\text{prior}}(t) + \boldsymbol{g}(t)).$$

Now, though, we can directly compare $2\boldsymbol{\Psi}(t)$ to $\boldsymbol{G}(t)$ and $(\boldsymbol{A}^{\text{prior}}(t) - \boldsymbol{A}(t))\boldsymbol{m}(t) + \boldsymbol{\lambda}(t)$ to $\boldsymbol{g}(t)$. In particular,

$$\frac{d(2\boldsymbol{\Psi}(t))}{dt} = -\boldsymbol{A}(t)^\top(2\boldsymbol{\Psi}(t)) - (2\boldsymbol{\Psi}(t))\boldsymbol{A}(t) - (\boldsymbol{A}^{\text{prior}}(t) - \boldsymbol{A}(t))^\top(\boldsymbol{A}^{\text{prior}}(t) - \boldsymbol{A}(t)),$$
$$2\boldsymbol{\Psi}(t_i) = 2\boldsymbol{\Psi}(t_i^+) + \boldsymbol{C}^\top\boldsymbol{R}^{-1}\boldsymbol{C},$$

which is not equivalent to the Riccati equation satisfied by $(\boldsymbol{G}(t))_{0 \le t \le T_{\max}}$.

Suppose, for example, $(\boldsymbol{A}(t), \boldsymbol{b}(t))_{0 \le t \le T_{\max}}$ are initialized to $(\boldsymbol{A}^{\text{prior}}(t), \boldsymbol{b}^{\text{prior}}(t))_{0 \le t \le T_{\max}}$. Then

$$\frac{d(2\boldsymbol{\Psi}(t))}{dt} = -(\boldsymbol{A}^{\text{prior}}(t))^\top(2\boldsymbol{\Psi}(t)) - (2\boldsymbol{\Psi}(t))\boldsymbol{A}^{\text{prior}}(t), \quad 2\boldsymbol{\Psi}(t_i) = 2\boldsymbol{\Psi}(t_i^+) + \boldsymbol{C}^\top\boldsymbol{R}^{-1}\boldsymbol{C}.$$

$\square$

## K   SING-GP: Drift estimation with Gaussian process priors

Here, we present the GP-SDE generative model and full inference algorithm using SING-GP from Section 3.5. Our presentation of the GP-SDE model primarily follows that of Duncker et al. [13] and Hu et al. [14].

### K.1 GP-SDE generative model

The GP-SDE model consists of the latent SDE model from Section 2.1 combined with Gaussian process priors on the output dimensions of the drift function $\boldsymbol{f}(\boldsymbol{x}(t))$. More formally, we have

$$d\boldsymbol{x}(t) = \boldsymbol{f}(\boldsymbol{x}(t))dt + \boldsymbol{\Sigma}^{\frac{1}{2}}d\boldsymbol{w}(t), \quad \mathbb{E}[\boldsymbol{y}(t_i)|\boldsymbol{x}] = g(\boldsymbol{C}\boldsymbol{x}(t_i) + \boldsymbol{d})$$

where

$$f_d(\cdot) \overset{\text{iid}}{\sim} \mathcal{GP}(0, \kappa_{\boldsymbol{\theta}}(\cdot, \cdot)), \quad d = 1, \dots, D.$$

Here, $\boldsymbol{f}(\cdot) = [f_1(\cdot), \dots, f_D(\cdot)]^{\top}$, $\kappa_{\boldsymbol{\theta}}$ is the kernel covariance function, and $\boldsymbol{\theta}$ are kernel hyperparameters. As in Section 2.1, we denote all model hyperparameters as $\boldsymbol{\Theta} = \{\boldsymbol{\theta}, \boldsymbol{C}, \boldsymbol{d}\}$. Throughout this section, we assume $\boldsymbol{\Sigma} = \boldsymbol{I}$ without loss of generality. In addition to our primary goal of performing inference over $\boldsymbol{x}(t)$, we would also like to perform inference over $\boldsymbol{f}(\cdot)$. In Appendix K.2, we outline an approach from Duncker et al. [13] that uses sparse GP approximations for inference over $\boldsymbol{f}(\cdot)$.

### K.2 Sparse GP approximations for posterior drift inference

Following Duncker et al. [13], we augment the generative model from Appendix K.1 with *sparse inducing points* [36], which enable tractable approximate inference of $\boldsymbol{f}(\cdot)$. This technique allows the computational complexity of inference of $\boldsymbol{f}(\cdot)$ evaluated at a batch of $L$ locations to be reduced from $\mathcal{O}(L^3)$ to $\mathcal{O}(LM^2)$, where $M \ll L$ is the number of inducing points. Let us denote the inducing locations as $\{\boldsymbol{z}_m\}_{m=1}^M \subset \mathbb{R}^N$ and corresponding inducing values as $\{\boldsymbol{u}_d\}_{d=1}^D \subset \mathbb{R}^M$. After introducing these variables, the full generative model from Appendix K.1 becomes

$$p(\boldsymbol{y}, \boldsymbol{x}, \boldsymbol{f}, \boldsymbol{u} \mid \boldsymbol{\Theta}) = p(\boldsymbol{y} \mid \boldsymbol{x}, \boldsymbol{\Theta})p(\boldsymbol{x} \mid \boldsymbol{f})\prod_{d=1}^D p(f_d \mid \boldsymbol{u}_d, \boldsymbol{\Theta})p(\boldsymbol{u}_d \mid \boldsymbol{\Theta}).$$

Treating $\boldsymbol{u}_d$ as pseudo-observations, we assume the following prior on inducing points which derives from the GP kernel:

$$p(\boldsymbol{u}_d \mid \boldsymbol{\Theta}) = \mathcal{N}(\boldsymbol{u}_d \mid \boldsymbol{0}, \boldsymbol{K}_{zz})$$

where $\boldsymbol{K}_{zz}$ is the Gram matrix corresponding to the batch of points $\{\boldsymbol{z}_m\}_{m=1}^M$. By properties of conditional Gaussian distributions, this implies that $f_d(\cdot) \mid \boldsymbol{u}_d$ is another GP,

$$f_d(\cdot) \mid \boldsymbol{u}_d \sim \mathcal{GP}(\mu_{f_d \mid \boldsymbol{u}_d}(\cdot), \kappa_{f_d \mid \boldsymbol{u}_d}(\cdot, \cdot))$$

where

$$\mu_{f_d \mid \boldsymbol{u}_d}(\boldsymbol{x}) = \boldsymbol{k}_{xz}\boldsymbol{K}_{zz}^{-1}\boldsymbol{u}_d$$
$$\kappa_{f_d \mid \boldsymbol{u}_d}(\boldsymbol{x}, \boldsymbol{x}') = \kappa_{\boldsymbol{\theta}}(\boldsymbol{x}, \boldsymbol{x}') - \boldsymbol{k}_{xz}\boldsymbol{K}_{zz}^{-1}\boldsymbol{k}_{zx}.$$

Above, $\boldsymbol{k}_{xz} := [\kappa_{\boldsymbol{\theta}}(\boldsymbol{x}, \boldsymbol{z}_1), \dots, \kappa_{\boldsymbol{\theta}}(\boldsymbol{x}, \boldsymbol{z}_M)]$ and $\boldsymbol{k}_{zx} = \boldsymbol{k}_{xz}^{\top}$.

To perform VI in the GP-SDE model, we follow Duncker et al. [13] and define an augmented variational family of the form

$$q(\boldsymbol{x}, \boldsymbol{f}, \boldsymbol{u}) = q(\boldsymbol{x})\prod_{d=1}^D p(f_d \mid \boldsymbol{u}_d, \boldsymbol{\Theta})q(\boldsymbol{u}_d).$$

For the variational family on inducing points, we choose

$$q(\boldsymbol{u}_d) = \mathcal{N}(\boldsymbol{u}_d \mid \boldsymbol{m}_u^{d*}, \boldsymbol{S}_u^{d*})$$

so that $\boldsymbol{m}_u^{d*}$ and $\boldsymbol{S}_u^{d*}$ have convenient closed-form updates, as we describe in the next section.

We use this augmented posterior to derive a continuous-time ELBO analogous to $\mathcal{L}_{\text{cont}}$ in eq. (8) of the main text. By Jensen's inequality,

$$
\begin{aligned}
&\log p(\boldsymbol{y} \mid \boldsymbol{\Theta}) \\
&= \log \int p(\boldsymbol{y} \mid \boldsymbol{x}, \boldsymbol{\Theta}) p(\boldsymbol{x} \mid \boldsymbol{f}) p(\boldsymbol{f} \mid \boldsymbol{u}, \boldsymbol{\Theta}) p(\boldsymbol{u} \mid \boldsymbol{\Theta}) d\boldsymbol{x} d\boldsymbol{f} d\boldsymbol{u} \\
&\geq \int q(\boldsymbol{x}, \boldsymbol{f}, \boldsymbol{u}) \log \frac{p(\boldsymbol{y} \mid \boldsymbol{x}, \boldsymbol{\Theta}) p(\boldsymbol{x} \mid \boldsymbol{f}) p(\boldsymbol{f} \mid \boldsymbol{u}, \boldsymbol{\Theta}) p(\boldsymbol{u} \mid \boldsymbol{\Theta})}{q(\boldsymbol{x}, \boldsymbol{f}, \boldsymbol{u})} d\boldsymbol{x} d\boldsymbol{f} d\boldsymbol{u} \\
&= \int q(\boldsymbol{x}, \boldsymbol{f}, \boldsymbol{u}) \log \frac{p(\boldsymbol{y} \mid \boldsymbol{x}, \boldsymbol{\Theta}) p(\boldsymbol{x} \mid \boldsymbol{f}) \prod_{d=1}^{D} p(\boldsymbol{u}_d \mid \boldsymbol{\Theta})}{q(\boldsymbol{x}) \prod_{d=1}^{D} q(\boldsymbol{u}_d)} d\boldsymbol{x} d\boldsymbol{f} d\boldsymbol{u} \\
&= \mathbb{E}_{q(\boldsymbol{x})}[\log p(\boldsymbol{y} \mid \boldsymbol{x}, \boldsymbol{\Theta})] - \mathbb{E}_{q(\boldsymbol{f})}[\mathrm{D}_{\mathrm{KL}}\left(q(\boldsymbol{x}) \,\|\, p(\boldsymbol{x} \mid \boldsymbol{f})\right)] - \sum_{d=1}^{D} \mathrm{D}_{\mathrm{KL}}\left(q(\boldsymbol{u}_d) \,\|\, p(\boldsymbol{u}_d \mid \boldsymbol{\Theta})\right) \\
&:= \mathcal{L}_{\text{cont-GP}}(q(\boldsymbol{x}), q(\boldsymbol{u}), \boldsymbol{\Theta}),
\end{aligned}
$$

where

$$
q(\boldsymbol{f}) = \prod_{d=1}^{D} \int p(f_d \mid \boldsymbol{u}_d, \boldsymbol{\Theta}) q(\boldsymbol{u}_d) d\boldsymbol{u}_d. \tag{36}
$$

Since $\mathcal{L}_{\text{cont-GP}}(q(\boldsymbol{x}), q(\boldsymbol{u}), \boldsymbol{\Theta})$ is intractable to compute directly, as we discuss in Section 2.3, we introduce an approximation to the continuous-time ELBO that is analogous to $\mathcal{L}_{\text{cont}}$ in eq. (9). This objective is

$$
\mathcal{L}_{\text{approx-GP}}(q(\boldsymbol{x}_{0:T}), q(\boldsymbol{u}), \boldsymbol{\Theta}) :=
$$

$$
\mathbb{E}_{q(\boldsymbol{x})}[\log p(\boldsymbol{y} \mid \boldsymbol{x}, \boldsymbol{\Theta})] - \mathbb{E}_{q(\boldsymbol{f})}[\mathrm{D}_{\mathrm{KL}}\left(q(\boldsymbol{x}_{0:T}) \,\|\, \tilde{p}(\boldsymbol{x}_{0:T} \mid \boldsymbol{f}, \boldsymbol{\Theta})\right)] - \sum_{d=1}^{D} \mathrm{D}_{\mathrm{KL}}\left(q(\boldsymbol{u}_d) \,\|\, p(\boldsymbol{u}_d \mid \boldsymbol{\Theta})\right). \tag{37}
$$

In eq. (37), we use definitions from Section 2.3: $q(\boldsymbol{x}_{0:T})$ is the finite-dimensional variational posterior on time grid $\boldsymbol{\tau}$ and $\tilde{p}(\boldsymbol{x}_{0:T})$ is the corresponding prior process on the same time grid discretized using Euler–Maruyama. We note that $\mathcal{L}_{\text{approx-GP}}$ has two main differences from $\mathcal{L}_{\text{approx}}$: (1) the second term now involves an expectation with respect to $q(\boldsymbol{f})$, which comes from the Gaussian process prior on the drift, and (2) $\mathcal{L}_{\text{approx-GP}}$ has an additional third term regularizing the posterior on inducing points towards its GP-derived prior.

### K.3   Inference and learning using SING-GP

In a GP-SDE model, the two inference goals are (1) inferring a posterior over the latent states $q(\boldsymbol{x}_{0:T})$ and (2) inferring a posterior over the drift function (i.e., dynamics) $q(\boldsymbol{f})$. In SING-GP, we propose to use SING as described in Section 3 to perform latent state inference, and we derive novel closed-form updates for $q(\boldsymbol{u}_d)$ based on $\mathcal{L}_{\text{approx-GP}}$. In the next section, we will show how Gaussian conjugacy allows for straightforward inference of $q(\boldsymbol{f})$ given $q(\boldsymbol{u}_d)$.

To update $q(\boldsymbol{u}_d)$, we would like to maximize $\mathcal{L}_{\text{approx-GP}}(q(\boldsymbol{x}_{0:T}), q(\boldsymbol{u}), \boldsymbol{\Theta})$ with respect to the variational parameters $\boldsymbol{m}_u^{d*}, \boldsymbol{S}_u^{d*}$ given the current estimates of $q(\boldsymbol{x}_{0:T})$ and $\boldsymbol{\Theta}$. By direct differentiation of $\mathcal{L}_{\text{approx-GP}}(q(\boldsymbol{x}_{0:T}), q(\boldsymbol{u}), \boldsymbol{\Theta})$ with respect to $\boldsymbol{m}_u^{d*}$ and $\boldsymbol{S}_u^{d*}$, one can derive the following analytical updates for $\boldsymbol{m}_u^{d*}$ and $\boldsymbol{S}_u^{d*}$:

$$
\boldsymbol{S}_u^{d*} = \boldsymbol{K}_{zz}\left(\boldsymbol{K}_{zz} + \Delta t \sum_{i=0}^{T} \mathbb{E}_{q(\boldsymbol{x}_i)}[\boldsymbol{k}_{zx_i}\boldsymbol{k}_{x_i z}]\right)^{-1}\boldsymbol{K}_{zz}
$$

$$
\boldsymbol{m}_u^* = \boldsymbol{S}_u^{d*}\boldsymbol{K}_{zz}^{-1}\left(\Delta t \sum_{i=0}^{T-1} \mathbb{E}_{q(\boldsymbol{x}_i)}[\boldsymbol{k}_{zx_i}]\left(\frac{\boldsymbol{m}_{i+1} - \boldsymbol{m}_i}{\Delta t}\right) + \Delta t \sum_{i=0}^{T-1} \mathbb{E}_{q(\boldsymbol{x}_i)}\left[\frac{d\boldsymbol{k}_{zx}}{d\boldsymbol{x}}\right]\left(\frac{\boldsymbol{S}_{i,i+1} - \boldsymbol{S}_i}{\Delta t}\right)\right). \tag{38}
$$

In the above equation, $\boldsymbol{m}_u^* \in \mathbb{R}^{M \times D}$ contains $\boldsymbol{m}_u^{d*}$ in each column. The required kernel expectations in eq. (38) have analytical expressions for, e.g., the commonly used radial basis function (RBF) kernel, and otherwise can be approximated via Gaussian quadrature for specialized kernels. Moreover,

observe that since the dependence on $T$ only appears in sums, this update can be computed in parallel across time steps.

Finally, the learning goal in the GP-SDE is to optimize the kernel hyperparameters $\boldsymbol{\theta}$. Prior work using GP-SDEs with structured GP kernels [14] found that alternating between updating $q(\boldsymbol{u})$ and optimizing $\boldsymbol{\theta}$ was prone to getting stuck in local maxima of the ELBO. Hence, we adopt the kernel hyperpararmeter learning approach proposed by Hu et al. [14], which uses the update

$$\boldsymbol{\theta}^* = \arg\max_{\boldsymbol{\theta}} \left\{ \max_{q(\boldsymbol{u})} \mathcal{L}_{\text{approx-GP}}(q(\boldsymbol{x}_{0:T}), q(\boldsymbol{u}), \boldsymbol{\theta}) \right\}.$$

The inner maximization can be performed via the closed-form updates as discussed above. We perform the outer maximization, which depends on both $q(\boldsymbol{u})$ and $\mathcal{L}_{\text{approx-GP}}(q(\boldsymbol{x}_{0:T}), q(\boldsymbol{u}), \boldsymbol{\theta})$, via Adam optimization. In practice, we find that this kernel hyperparameter learning method performs well on both synthetic and real datasets.

### K.4  Posterior inference of the drift at unseen points

The sparse approximation $q(\boldsymbol{u})$ allows us to compute the posterior distribution $\boldsymbol{f}^* := \boldsymbol{f}(\boldsymbol{x}^*)$ at any unseen latent space location $\boldsymbol{x}^* \in \mathbb{R}^D$. Recall that

$$q(\boldsymbol{f}^*) = \prod_{d=1}^{D} p(f_d^* \mid \boldsymbol{u}_d, \boldsymbol{\Theta}) q(\boldsymbol{u}_d) d\boldsymbol{u}_d.$$

Since $q(\boldsymbol{u}_d)$ is Gaussian and $p(f_d^* \mid \boldsymbol{u}_d, \boldsymbol{\Theta})$ is conditionally Gaussian, then $q(\boldsymbol{f}^*)$ is also a closed-form Gaussian distribution. This takes the form,

$$q(f_d^*) = \mathcal{N}(f_d^* \mid \boldsymbol{k}_{x^*z} \boldsymbol{K}_{zz}^{-1} \boldsymbol{m}_u^d, \kappa_{\boldsymbol{\theta}}(\boldsymbol{x}^*, \boldsymbol{x}^*) - \boldsymbol{k}_{x^*z} \boldsymbol{K}_{zz}^{-1} \boldsymbol{k}_{zx^*} + \boldsymbol{k}_{x^*z} \boldsymbol{K}_{zz}^{-1} \boldsymbol{S}_u^d \boldsymbol{K}_{zz}^{-1} \boldsymbol{k}_{zx^*}).$$

### K.5  Incorporating inputs

For our application of SING-GP to real neural data in Section 5.4, we incorporate external inputs that may affect the underlying latent trajectories. Following Hu et al. [14], we model this as

$$d\boldsymbol{x}(t) = (\boldsymbol{f}(\boldsymbol{x}(t)) + \boldsymbol{B}\boldsymbol{v}(t))dt + \boldsymbol{\Sigma}^{\frac{1}{2}} d\boldsymbol{w}(t)$$

where $\boldsymbol{v}(t) \in \mathbb{R}^I$ is a time-varying, known input signal and $\boldsymbol{B} \in \mathbb{R}^{D \times I}$ is a linear transformation of the input into the latent space. We denote $\boldsymbol{v}_i := \boldsymbol{v}(\tau_i)$ where $\tau_i$ is a time point in the grid $\boldsymbol{\tau}$. We derive the optimal update $\boldsymbol{B}^*$ that maximizes $\mathcal{L}_{\text{approx-GP}}$ as

$$\boldsymbol{B}^* = \left( \sum_{i=0}^{T-1} \left( \boldsymbol{m}_{i+1} - \boldsymbol{m}_i - \Delta t \mathbb{E}_{q(\boldsymbol{f}), q(\boldsymbol{x}_i)}[\boldsymbol{f}(\boldsymbol{x}_i)] \right) \boldsymbol{v}_i^\top \right) \left( \Delta t \sum_{i=0}^{T-1} \boldsymbol{v}_i \boldsymbol{v}_i^\top \right)^{-1}.$$

## L  Experiment details and additional results

For all experiments, we fit our models using either an NVIDIA A100 GPU or NVIDIA H100 GPU.

### L.1  Inference on synthetic data

Here, we provide additional experimental details to supplement Section 5.1. For this section only, we will abuse notation and use $\boldsymbol{x}_i(t)$ to denote individual coordinates of $\boldsymbol{x}(t) \in \mathbb{R}^D$, for $0 \le t \le T_{\max}$. This not to be confused with time steps of the grid $\boldsymbol{\tau}$, as introduced in Section 2.3.

#### L.1.1  Linear dynamics, Gaussian observations

**Details on simulated data**  We generate 30 independent trials from an SDE with linear dynamics

$$\boldsymbol{f}(\boldsymbol{x}(t)) = \boldsymbol{A}\boldsymbol{x}(t), \quad \boldsymbol{A} = \frac{1}{\Delta t} \left( 0.997 \begin{bmatrix} \cos\theta & -\sin\theta \\ \sin\theta & \cos\theta \end{bmatrix} - \boldsymbol{I} \right)$$

with $\theta = \pi/250$. To simulate these trials, we use an Euler–Maruyama discretization with $\Delta t = 0.001$ and $T_{\max} = 1$. Initial conditions are sampled uniformly in $[-2, 2]^2$. We then simulate observations $\boldsymbol{y}_i \sim \mathcal{N}(\boldsymbol{C}\boldsymbol{x}_i + \boldsymbol{d}, \boldsymbol{R})$ at every time point $\tau_i$, where the entries of $\boldsymbol{C}, \boldsymbol{d}$ are sampled as *i.i.d.* standard normal variables and we set $\boldsymbol{R} = 0.35\boldsymbol{I}$.

**Details on model fitting**   For our main result in Figure 2 (top row), we fit SING with 20 iterations and step size $\rho = 1$, with hyperparameters $\boldsymbol{\Theta}$ fixed to their true generative values. As expected, convergence in this fully conjugate setting occurs after only 1 iteration. We use analogous settings for VDP: 20 variational EM iterations, each consisting of a single forward/backward pass to update variational parameters. For our Kalman smoothing baseline, we use the Dynamax package [54]. For Adam-based optimization, we sweep over initial learning rates $\{10^{-4}, 5 \cdot 10^{-4}, 10^{-3}, 5 \cdot 10^{-3}, 10^{-2}\}$, fit for 20 iterations, and choose the best fit according to the final ELBO value, which in this case corresponded to learning rate $5 \cdot 10^{-3}$.

**Computing latents RMSE**   The latents RMSE metric from Figure 2D and H are computed as follows:

$$\text{latents RMSE} = \left( \frac{1}{T+1} \sum_{i=0}^{T} \mathbb{E}_{q(\boldsymbol{x}_i)}[\|\boldsymbol{x}_i - \boldsymbol{x}_i^{\text{true}}\|_2^2] \right)^{1/2}.$$

Since $q(\boldsymbol{x}_i) = \mathcal{N}(\boldsymbol{x}_i \mid \boldsymbol{m}_i, \boldsymbol{S}_i)$, this can be computed analytically using the identity

$$\mathbb{E}_{q(\boldsymbol{x}_i)}[\|\boldsymbol{x}_i - \boldsymbol{x}_i^{\text{true}}\|_2^2] = \text{tr}(\boldsymbol{S}_i) + \|\boldsymbol{m}_i - \boldsymbol{x}_i^{\text{true}}\|_2^2.$$

### L.1.2   Place cell model

**Details on simulated data**   We generate 30 independent trials from an SDE whose drift is characterized by a Van der Pol oscillator,

$$f_1(\boldsymbol{x}(t)) = \tau \mu \left( \boldsymbol{x}_1(t) - \frac{1}{3} \boldsymbol{x}_1(t)^3 - \boldsymbol{x}_2(t) \right)$$

$$f_2(\boldsymbol{x}(t)) = \tau \frac{\boldsymbol{x}_1(t)}{\mu}$$

with $\tau = 10$ and $\mu = 2$. To simulate these trials, we use an Euler–Maruyama discretization with $\Delta t = 0.001$ and $T_{\max} = 2$. Initial conditions are sampled uniformly in $[-3, 3]^2$.

For our observations, we use a Poisson count model inspired by spiking activity in place cells. To model the tuning curves (i.e., expected firing rates) for this model, we use radial basis functions. For $N = 8$ neurons, these take the form

$$r_n(\boldsymbol{x}) = a \exp \left( -\frac{1}{2\ell^2} \|\boldsymbol{x} - \boldsymbol{c}_n\|_2^2 \right) + a_0, \quad n = 1, \ldots, N$$

where $\{\boldsymbol{c}_n\}_{n=1}^{N} \subset \mathbb{R}^D$ are centers placed along the latent trajectories, $\ell$ controls the width of the curve, and $a, a_0$ control the peak and background firing rates respectively. For our experiments, we set $\ell = 0.5, a = 2.5$, and $a_0 = 0.25$. We then generate Poisson spike counts for each time bin $\tau_i$ according to

$$y_n(\tau_i) \sim \text{Pois}(r_n(\boldsymbol{x}(\tau_i))).$$

In this likelihood model, the expected log-likelihood term $\mathbb{E}_q[\log p(\boldsymbol{y}|\boldsymbol{x})]$ is not available in closed form, so we approximate it using Gauss-Hermite quadrature.

**Details on model fitting**   For our main result in Figure 2 (bottom row), we fit SING for 500 iterations. We use the following step size schedule inspired by [25]: $\rho$ is log-linearly increased from $10^{-3}$ to $10^{-1.5}$ for the first 10 iterations, and then kept at $10^{-1.5}$ for the rest of the iterations. We fit VDP for 500 iterations and use the "soft" updates in eq. (32) to ensure numerical stability. After manually tuning the learning rate, we find that VDP performs best with $\omega = 0.05$. For the conditional moments Gaussian smoother (CMGS), we use an implementation from Dynamax [54]. For Adam-based optimization, we sweep over initial learning rates $\{10^{-4}, 5 \cdot 10^{-4}, 10^{-3}, 5 \cdot 10^{-3}, 10^{-2}\}$, fit for 500 iterations, and choose the best fit according to the final ELBO value, which in this case corresponded to learning rate $10^{-2}$.

### L.1.3 Embedded Lorenz attractor

**Details on simulated data**  We generate 30 independent trials from a $D$-dimensional latent SDE whose first 3 dimensions evolve according to a Lorenz attractor drift, and the remaining dimensions are a random walk. These drifts are characterized by

$$
\begin{aligned}
f_1(\boldsymbol{x}(t)) &= \alpha(\boldsymbol{x}_2(t) - \boldsymbol{x}_1(t)) \\
f_2(\boldsymbol{x}(t)) &= \boldsymbol{x}_1(t)(\rho - \boldsymbol{x}_3(t)) - \boldsymbol{x}_2(t) \\
f_3(\boldsymbol{x}(t)) &= \boldsymbol{x}_1(t)\boldsymbol{x}_2(t) - \beta\boldsymbol{x}_3(t) \\
f_d(\boldsymbol{x}(t)) &= 0, \quad d > 3.
\end{aligned}
\tag{39}
$$

where $(\alpha, \rho, \beta) = (10, 28, 8/3)$. To simulate these trials, we use an Euler–Maruyama discretization with $\Delta t = 0.001$ and $T_{\max} = 1$. Initial conditions are sampled from independent standard normal distributions. We then simulate 100-dimensional observations $\boldsymbol{y}_i \sim \mathcal{N}(\boldsymbol{Cx}_i + \boldsymbol{d}, \boldsymbol{R})$, where the entries of $\boldsymbol{C}$ and $\boldsymbol{d}$ are sampled *i.i.d.* from $\mathcal{N}(0, 0.01)$ and we set $\boldsymbol{R} = 0.05\boldsymbol{I}$.

**Details on model fitting**  For our results in Figure 4, we fit SING for $D = 3, 5, 10, 20, 50$ using two different methods of approximating transition expectations: (1) Monte Carlo and (2) Gauss-Hermite quadrature. For each method, we fit SING for 1000 iterations, keeping output parameters and prior drift parameters fixed to their true values. The step size $\rho$ is log-linearly increased from $10^{-3}$ to $10^{-2}$ for the first 10 iterations, and then kept at $10^{-2}$ for the rest of the iterations. For the Monte Carlo approach, we use a single Monte Carlo sample per expectation. For the quadrature approach, we use 6 quadrature nodes per latent dimension for $D = 3$ and 5; beyond $D = 5$, quadrature results in out-of-memory errors on a NVIDIA A100 GPU.

## L.2 Comparison to neural-SDE variational posterior

**Background on SDE Matching**  SDE Matching, proposed by Bartosh et al. [41], is an algorithm for performing joint (approximate) inference and learning in the latent SDE model. The authors approximate the intractable posterior distribution with a neural stochastic differential equation

$$
\tilde{q}(\boldsymbol{x}) : d\boldsymbol{x}(t) = \tilde{\boldsymbol{f}}(\boldsymbol{x}(t)|\phi)dt + \boldsymbol{\Sigma}^{1/2}d\boldsymbol{w}(t), \quad \boldsymbol{x}(0) \sim \mathcal{N}(\boldsymbol{m}_\phi, \boldsymbol{S}_\phi), \quad 0 \leq t \leq T_{\max},
$$

where $\phi$ are the parameters of the variational posterior and $\tilde{\boldsymbol{f}}(\cdot|\phi)$ is a neural network drift function. Unlike SING, SDE Matching also applies to models with state-dependent diffusion coefficient. Compared to prior works that perform VI with neural-SDE variational families, SDE Matching is advantageous since it does not require drawing samples from the variational posterior.

Also dissimilar from SING, SDE Matching is an amortized VI algorithm, meaning that the variational approximation is "shared" across samples $(\boldsymbol{x}^{(\ell)}(t))_{0 \leq t \leq T_{\max}}$. In particular, the posterior drift $\tilde{\boldsymbol{f}}(\boldsymbol{x}^{(\ell)}(t)|\phi)$ is modeled also as a function of the observations $\{\boldsymbol{y}^{(\ell)}(t_i)\}_{i=1}^n$.

**Details on simulated data**  We compare SING to SDE Matching [41] on the three-dimensional stochastic Lorenz attractor benchmark dataset as described in Li et al. [11]. In particular, we generate latents according to the three-dimensional SDE on $0 \leq t \leq T_{\max}$, $T_{\max} = 1$ with drift function

$$
\begin{aligned}
f_1(\boldsymbol{x}(t)) &= \alpha(\boldsymbol{x}_2(t) - \boldsymbol{x}_1(t)) \\
f_2(\boldsymbol{x}(t)) &= \boldsymbol{x}_1(t)(\rho - \boldsymbol{x}_3(t)) - \boldsymbol{x}_2(t) \\
f_3(\boldsymbol{x}(t)) &= \boldsymbol{x}_1(t)\boldsymbol{x}_2(t) - \beta\boldsymbol{x}_3(t)
\end{aligned}
\tag{40}
$$

for $(\alpha, \rho, \beta) = (10, 28, 8/3)$, diffusion coefficient $\sigma = 0.15$, and initial condition $\mathcal{N}(\boldsymbol{x}(0)|\boldsymbol{0}, \boldsymbol{I})$. We sample the SDE according to the Euler–Maruyama discretization with $\Delta t = 0.00025$. We observe $\boldsymbol{y}(t_i)$ according to $\mathcal{N}(\boldsymbol{y}(t_i)|\boldsymbol{Cx}(t_i) + \boldsymbol{d}, (0.01)^2\boldsymbol{I})$ on an evenly spaced grid of size 0.025, where $\boldsymbol{C}$ and $\boldsymbol{d}$ are also chosen as in Li et al. [11]. We sample 1024 total trials.

**Details on learning**  Following Bartosh et al. [41], we model the prior $p(\boldsymbol{x})$ as a neural-SDE whose neural network drift has a single softplus activation. However, unlike Li et al. [11], Bartosh et al. [41] who consider a four-dimensional latent space, we consider a three-dimensional latent space so that we can directly assess recovery of the ground truth Lorenz attractor dynamics.

For SDE Matching, we use the authors' publicly available codebase, which includes code for evaluation on the same benchmark; we do not further tune any hyperparameters. For SING, we perform $10^3$ iterations (10 E-steps, 10 M-steps each), and for SDE Matching we perform $10^4$ iterations; these choices ensure that the number of gradient updates is the same for both algorithms.

To perform the E-step of the vEM algorithm with SING, we define the time grid $\boldsymbol{\tau}$ using evenly spaced points with $\Delta t = 0.0025$. We increase the NGVI step size $\rho$ on a log-linear scale for 100 iterations from $10^{-5}$ to $10^{-3}$ and hold it constant for subsequent iterations. To perform the M-step, we update both the output parameters and the drift parameters using Adam. We hold the learning rate for the drift parameters constant at $10^{-3}$. And for learning the output parameters, we decay the learning rate from $10^{-3}$ to $10^{-7}$ for 500 iterations and then hold it constant. To compute expectations of the prior drift under the variational posterior, we use a single Monte Carlo sample.

| # Samples | Latents RMSE | | Dynamics RMSE | |
|---|---|---|---|---|
| | **SING** | **SDE Matching** | **SING** | **SDE Matching** |
| 10 | .50 ± .05 | .99 ± .07 | .25 ± .04 | .49 ± .05 |
| 30 | .52 ± .05 | .78 ± .03 | .45 ± .38 | .25 ± .04 |
| 50 | .49 ± .05 | .70 ± .02 | .31 ± .06 | .23 ± .03 |
| 100 | .50 ± .04 | .67 ± .03 | .29 ± .14 | .19 ± .01 |
| 300 | .46 ± .05 | .60 ± .01 | .25 ± .05 | .17 ± .01 |
| 500 | .48 ± .05 | .59 ± .01 | .25 ± .06 | .17 ± .01 |
| 1024 | .58 ± .08 | .57 ± .01 | .56 ± .44 | .17 ± .01 |

Figure 7: **Comparison of SING and SDE Matching.** A comparison of latents and dynamics RMSE on the stochastic Lorenz attractor dataset for SING and SDE Matching. Errors represent 2SE computed over 10 random initializations of the prior drift and output parameters and random seeds for the algorithms.

**Evaluation metrics**   We compare the performance of SING and SDE Matching using two metrics: the latents and normalized dynamics root mean-squared error. We compute the latents RMSE exactly as in Appendix L.1.1. And for the normalized dynamics RMSE, we 1024 draw independent samples $(\boldsymbol{x}^{(j)}(t))_{0 \leq t \leq T_{\max}}$ from the ground-truth stochastic Lorenz attractor and estimate

$$\left( \int_0^{T_{\max}} \mathbb{E}\left[ \frac{\|\boldsymbol{f}(\boldsymbol{x}(t)) - \boldsymbol{f}(\boldsymbol{x}(t)|\boldsymbol{\Theta})\|^2}{\|\boldsymbol{f}(\boldsymbol{x}(t))\|^2} \right] \right)^{1/2} \approx \left( \frac{1}{\text{trials} \cdot \bar{T}} \sum_{i=1}^{\bar{T}} \frac{\|\boldsymbol{f}(\boldsymbol{x}(\bar{\tau}_i)) - \boldsymbol{f}(\boldsymbol{x}(\bar{\tau}_i)|\boldsymbol{\Theta})\|^2}{\|\boldsymbol{f}(\boldsymbol{x}(\bar{\tau}_i))\|^2} \right)^{1/2}.$$

where $\bar{\boldsymbol{\tau}} = \{\bar{\tau}_i\}_{i=1}^{\bar{T}}$ is the Euler–Maruyama sampling grid. Note that the samples on which the normalized dynamics RMSE is computed are different from those used to perform joint inference and learning (with either SING or SDE Matching).

**Results**   Our experimental results, summarized in Figure 7, provide compelling evidence that the structured Gaussian process variational approximation, combined with the fast and numerically stable SING inference algorithm, leads to more accurate recovery of the ground truth latents compared to SDE Matching. In particular, SDE Matching requires approximately 500 samples to match the latents RMSE that SING achieves with only 10 trials. This is likely because at each iteration SDE Matching updates the posterior using only a single time $t$ in $[0, T_{\max}]$, whereas SING updates the variational posterior on the entire grid $\boldsymbol{\tau}$.

However, SDE Matching outperforms SING in recovering the ground truth Lorenz attractor dynamics. In our experiments, we observe that while performing vEM with SING can achieve dynamics RMSE comparable to that of SDE Matching, the dynamics RMSE varies greatly depending on the random initialization of the prior neural network and output parameters. Improving learning with SING remains a future research direction.

### L.3 Drift estimation on synthetic data

**Details on simulated data** The Duffing equation is a second-order ODE described by

$$\frac{d^2 v}{dt^2} + \gamma \frac{dv}{dt} - \alpha v + \beta v^3 = 0, \quad 0 \le t \le T_{\max}.$$

By setting $\boldsymbol{x}_1 = v$, $\boldsymbol{x}_2 = \frac{dv}{dt}$, we can convert the Duffing equation into a system of first-order ODEs. Moreover, we add an independent Brownian motion with noise $\sigma = 0.2$ to each coordinate. Altogether, this yields the two-dimensional SDE with drift

$$\begin{aligned}
f_1(\boldsymbol{x}(t)) &= \boldsymbol{x}_2(t) \\
f_2(\boldsymbol{x}(t)) &= \alpha \boldsymbol{x}_1(t) - \beta \boldsymbol{x}_1(t)^3 - \gamma \boldsymbol{x}_2(t).
\end{aligned} \tag{41}$$

It is straightforward to verify that the Duffing equation has three fixed points when $\alpha/\beta > 0$: one at the origin and two at $(\pm\sqrt{\alpha/\beta}, 0)$. We generate four trajectories from the Duffing SDE eq. (41) with parameters $(\alpha, \beta, \gamma) = (2, 1, 0.1)$ and initial condition $\boldsymbol{x}(0) = (\sqrt{\alpha/\beta} - 0.1, 0.1)$, i.e., near the rightmost fixed point. We sample the solution to the SDE using the Euler–Maruyama discretization with $T_{\max} = 15$ and $\Delta t = 0.015$. All $10^3$ time steps are taken to define the grid $\boldsymbol{\tau}$. A visualization of two of the sampled trajectories is provided in Figure 3A.

From the $10^3$ total time steps, we sample $3 \times 10^2$ at random at which to include Gaussian observations $\mathcal{N}(\boldsymbol{y}(t_i)|\boldsymbol{C}\boldsymbol{x}(t_i) + \boldsymbol{d}, \boldsymbol{R})$. The true emissions parameters $\boldsymbol{C} \in \mathbb{R}^{N \times D}$ and $\boldsymbol{d} \in \mathbb{R}^D$ are chosen by sampling *i.i.d.* standard normal random variables. $\boldsymbol{R}$ is fixed to a diagonal matrix with entries $0.1$.

**Details on learning** We simultaneously perform inference and learning in the latent SDE model according to the variational EM (vEM) algorithm. For each variational EM step, we take 10 E-steps (according to either VDP or SING) as well as 50 M-steps. In each M-step, we update both the drift parameters $\boldsymbol{\theta}$ as well as the output parameters $\{\boldsymbol{C}, \boldsymbol{d}\}$. We update the drift parameters by performing Adam on the approximate ELBO $\mathcal{L}_{\text{approx}}$, and we use a closed-form update for the output parameters (see next section). In the case of the GP prior drift, the only learnable parameters are the length-scale and output scale of the RBF kernel. The updates to the posterior over inducing points $p(\boldsymbol{u})$ are performed during the E-step.

For the GP prior drift we use a $12 \times 12$ grid of inducing points on $[-6, 6]^2$. To choose the size of this grid, we looked at the first two principal component scores of the observations $\{\boldsymbol{y}(t_i)\}_{i=1}^n$. For the neural-SDE, we use a neural network with two hidden layers of size 64 and ReLU activations. And for the polynomial basis drift, we consider all polynomials up to order three $\{1, \boldsymbol{x}_1, \boldsymbol{x}_2, \boldsymbol{x}_1\boldsymbol{x}_2, \boldsymbol{x}_1^2, \boldsymbol{x}_2^2, \dots\}$. Note that by the form of the Duffing equation eq. (41), this polynomial drift function can, in theory, perfectly represent the true drift.

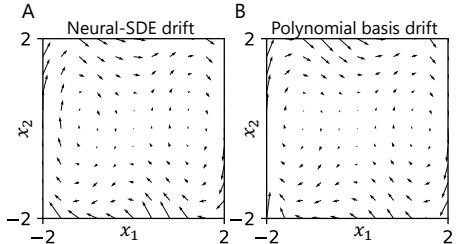

Figure 8: Prior drift functions learned via vEM with SING and Adam. **A**: Learned neural network drift. **B**: Learned polynomial basis drift.

We initialize the output parameters $\boldsymbol{C}$ and $\boldsymbol{d}$ by taking the first two principal components of the observations and the mean of the observations, respectively. The GP length scales and output scales are initialized to one; the neural network weight matrices are initialized by *i.i.d.* draws from a truncated normal distribution and the biases are initialized to zero; the coefficients on the polynomial basis are initialized by *i.i.d.* draws from $\mathcal{N}(0, (0.1)^2)$.

When using SING during the E-step, we increase $\rho$ on a log-linear scale from $10^{-3}$ to $10^{-1}$ during the first ten iterations, and we then fix it at $10^{-1}$ for the remaining 40 vEM iterations. We found this procedure of starting with $\rho$ small and subsequently increasing it on a log-linear scale during the first $k$ iterations to result in effective inference and learning [25]. As for VDP, we choose the soft update parameter $\omega$ to be as large as possible while maintaining the stability of the algorithm. For the GP prior and neural network drift classes, this selection procedure yielded $\omega = 0.05$. For the polynomial basis drift, we were not able to obtain convergence with any reasonable choice of $\omega$ (see Appendix H for a discussion of the parameter $\omega$). For this reason, it is omitted from Figure 3.

In Figure 8, we visualize the learned dynamics corresponding to the neural network and polynomial basis prior drift functions. The posterior mean and variance of the GP prior drift are visualized in Figure 3B. When performing inference with SING, both parameteric drift functions, in addition to the GP posterior mean, match the drift eq. (41) of the Duffing equation (Figure 3D). We measure the discrepancy between the learned and true drift by computing the dynamics root mean-squared error

$$
\text{dynamics RMSE} = \left( \sum_{\boldsymbol{v} \in \boldsymbol{V}} w(\boldsymbol{v}) \| \boldsymbol{f}(\boldsymbol{v}) - \boldsymbol{f}(\boldsymbol{v}|\boldsymbol{\Theta}) \|_2^2 \right)^{1/2}. \tag{42}
$$

Here, we take a fine grid of cells, each with equal $D$-dimensional volume, covering the latent space; we denote the collection of their centers by $\boldsymbol{V}$. Then, we compute the proportion of true latent trajectories that fall into the cell with center $\boldsymbol{v}$, written $w(\boldsymbol{v})$, such that $\sum_{\boldsymbol{v} \in \boldsymbol{V}} w(\boldsymbol{v}) = 1$. Mathematically, eq. (42) approximates

$$
\left( \frac{1}{\text{trials} \cdot T_{\max}} \sum_{j=1}^{\text{trials}} \int_0^{T_{\max}} \| \boldsymbol{f}(\boldsymbol{x}^{(j)}(t)) - \boldsymbol{f}(\boldsymbol{x}^{(j)}(t)|\boldsymbol{\Theta}) \|_2^2 dt \right)^{1/2},
$$

where trials represents the number of trials and $(\boldsymbol{x}^{(j)}(t))_{0 \leq t \leq T_{\max}}$ is the true latent trajectory corresponding to the $j$th trial.

**Closed-form updates for output parameters**   Next, we derive closed-form updates for the output parameters $\boldsymbol{C} \in \mathbb{R}^{N \times D}, \boldsymbol{d} \in \mathbb{R}^N, \boldsymbol{R} \in \mathbb{R}^{N \times N}$ in the Gaussian observation model

$$
p(\boldsymbol{y}|\boldsymbol{x}, \boldsymbol{\Theta}) = \prod_{i=1}^n \mathcal{N}(\boldsymbol{y}(t_i)|\boldsymbol{C}\boldsymbol{x}(t_i) + \boldsymbol{d}, \boldsymbol{R}),
$$

where $\boldsymbol{R}$ is assumed to be diagonal. These same update equations appear in Hu et al. [14]. Although not directly mentioned in the main text, we can also learn the entries of the emissions covariance matrix $\boldsymbol{R}$.

Writing out the expected log-likelihood of $p(\boldsymbol{y}|\boldsymbol{x})$ under $q(\boldsymbol{x})$, we have

$$
\mathbb{E}_q[\log p(\boldsymbol{y}|\boldsymbol{x}, \boldsymbol{\Theta})] = -\frac{n}{2}\log|\boldsymbol{R}| + \sum_{i=1}^n \mathbb{E}_q\left[-\frac{1}{2}\|\boldsymbol{y}(t_i) - \boldsymbol{C}\boldsymbol{x}(t_i) - \boldsymbol{d}\|_{\boldsymbol{R}^{-1}}^2\right] + \text{const}
$$

$$
= -\frac{n}{2}\log|\boldsymbol{R}| - \frac{n}{2}\boldsymbol{d}^\top \boldsymbol{R}^{-1}\boldsymbol{d} + \sum_{i=1}^n \left( \boldsymbol{y}(t_i)^\top \boldsymbol{R}^{-1}(\boldsymbol{C}\boldsymbol{m}(t_i) + \boldsymbol{d}) - \frac{1}{2}\boldsymbol{y}(t_i)^\top \boldsymbol{R}^{-1}\boldsymbol{y}(t_i) \right.
$$

$$
\left. - \frac{1}{2}\text{tr}(\boldsymbol{C}^\top \boldsymbol{R}^{-1}\boldsymbol{C}\left\{\boldsymbol{S}(t_i) + \boldsymbol{m}(t_i)\boldsymbol{m}(t_i)^\top\right\}) - \boldsymbol{d}^\top \boldsymbol{R}^{-1}\boldsymbol{C}\boldsymbol{m}(t_i) \right) + \text{const},
$$

where const represents a generic constant with respect to $(\boldsymbol{C}, \boldsymbol{d}, \boldsymbol{R})$. Differentiating with respect to $\boldsymbol{C}, \boldsymbol{d}$, and $\boldsymbol{R}^{-1}$ yields the update equations

$$
\boldsymbol{C}^*\left(\sum_{i=1}^n (\boldsymbol{S}(t_i) + \boldsymbol{m}(t_i)\boldsymbol{m}(t_i)^\top)\right) = \sum_{i=1}^n (\boldsymbol{y}(t_i) - \boldsymbol{d}^*)\boldsymbol{m}(t_i)^\top \tag{43}
$$

$$
\boldsymbol{d}^* = \frac{1}{n}\sum_{i=1}^n (\boldsymbol{y}(t_i) - \boldsymbol{C}^*\boldsymbol{m}(t_i)) \tag{44}
$$

$$
\boldsymbol{R}_{j,j}^* = \frac{1}{n}\sum_{i=1}^n \left( \boldsymbol{y}_j(t_i)^2 - 2\boldsymbol{y}_j(t_i)(\boldsymbol{C}_j^*)^\top \boldsymbol{m}(t_i) + ((\boldsymbol{C}_j^*)^\top \boldsymbol{m}(t_i))^2 + (\boldsymbol{C}_j^*)^\top \boldsymbol{S}(t_i)(\boldsymbol{C}_j^*) \right) - (\boldsymbol{d}_j^*)^2, \ 1 \leq j \leq D
$$

where $\boldsymbol{R}_{j,j}$ is the $j$th diagonal entry of $\boldsymbol{R}$ and $\boldsymbol{C}_j$ is the $j$th row of $\boldsymbol{C}$.

By defining $\bar{\boldsymbol{y}} = n^{-1} \sum_{i=1}^{n} \boldsymbol{y}(t_i)$ as well as $\bar{\boldsymbol{m}} = n^{-1} \sum_{i=1}^{n} \boldsymbol{m}(t_i)$ and plugging eq. (44) into eq. (43), we obtain

$$\boldsymbol{C}^* \left( \sum_{i=1}^{n} (\boldsymbol{S}(t_i) + (\boldsymbol{m}(t_i) - \bar{\boldsymbol{m}})(\boldsymbol{m}(t_i) - \bar{\boldsymbol{m}})^\top) + n\bar{\boldsymbol{m}}\bar{\boldsymbol{m}}^\top \right)$$

$$= \sum_{i=1}^{n} (\boldsymbol{y}(t_i) - \bar{\boldsymbol{y}})(\boldsymbol{m}(t_i) - \bar{\boldsymbol{m}})^\top + n\boldsymbol{C}^*\bar{\boldsymbol{m}}\bar{\boldsymbol{m}}^\top$$

$$\implies \boldsymbol{C}^* = \left( \sum_{i=1}^{n} (\boldsymbol{y}(t_i) - \bar{\boldsymbol{y}})(\boldsymbol{m}(t_i) - \bar{\boldsymbol{m}})^\top \right) \left( \sum_{i=1}^{n} ((\boldsymbol{m}(t_i) - \bar{\boldsymbol{m}})(\boldsymbol{m}(t_i) - \bar{\boldsymbol{m}})^\top + \boldsymbol{S}(t_i)) \right)^{-1}.$$

From the equation for $\boldsymbol{C}^*$, it is straightforward to solve for $\boldsymbol{d}^*$ and $\boldsymbol{R}^*_{j,j}$, $1 \leq j \leq D$.

**Discretization experiments**  In Figure 3E, we investigate the effect of the discretization size $\Delta t$ on the performance of the vEM algorithm with SING and VDP. Recall that our true latent trajectories are sampled with Euler–Maruyama step size 0.015, and so we would not expect $\mathcal{L}_{\text{approx}}$ to better approximate $\mathcal{L}_{\text{cont}}$ for smaller discretizations $\boldsymbol{\tau}$. In the case of VDP, though, we demonstrate that taking $\Delta t$ smaller than 0.015 leads to improvements in the stability of the algorithm. Moreover, for large $\Delta t$, performing vEM with VDP is highly unstable and may diverge depending on the neural network initialization. For this reason, we exclude the dynamics RMSE corresponding to $\Delta t > 0.15$.

In order to coarsen i.e., downsample the original grid $\boldsymbol{\tau}$, we fill in grid points where observations were not observed with zeros. Then, we consider each $j$th observation in $\boldsymbol{\tau}$. If that observation is zero (i.e., there is not an observation at that grid point), we take the nearest observation, should there exist an observation in the original grid that lies between consecutive points in the coarsened grid. This procedure increases $\Delta t$ to $j \cdot \Delta t$. In order to make the original grid $\boldsymbol{\tau}$ finer, i.e., upsample, we add $j$ zeros between the existing observations, again treating missing observations as zeros. This procedure decreases $\Delta t$ to $\Delta t/(j + 1)$. Note that coarsening the grid may lead to fewer observations, but making it finer will neither increase nor decrease the number of observations.

## L.4   Runtime comparisons

**Experiment details**  We perform runtime comparisons on simulated datasets consisting of a 2D linear dynamical system (LDS) with Gaussian observations. Each point in Figures 5 and 9 represents the average wall clock time (in seconds) over 5 runs of each method on randomly sampled dynamical systems. All runtime comparisons were performed on an NVIDIA A100 GPU.

We compare two methods: (1) 'parallel' SING, which uses associative scans as described in Section 3.4 to perform natural-to-mean parameter conversion, and (2) 'sequential' SING, which uses the standard sequential algorithm to perform this conversion, as we detail in Appendix G.1.

For each run, we randomly generate the data as follows. We construct a linear drift function of the form

$$\boldsymbol{f}(\boldsymbol{x}(t)) = \boldsymbol{A}\boldsymbol{x}(t) + \boldsymbol{b}$$

where $\boldsymbol{A} \in \mathbb{R}^{2 \times 2}$ is constrained to have negative real eigenvalues for stability and $\boldsymbol{b} \in \mathbb{R}^2$ has small non-zero entries. Using Euler–Maruyama discretization with $\Delta t = 0.001$, we sample for $T$ time steps from this 2D linear SDE. Then, for each time step $\tau_i$, we sample 10D Gaussian observations,

$$\boldsymbol{y}(\tau_i) \sim \mathcal{N}(\boldsymbol{C}\boldsymbol{x}(\tau_i) + \boldsymbol{d}, \boldsymbol{R})$$

where $\boldsymbol{C}$ and $\boldsymbol{d}$ are randomly sampled with *i.i.d.* standard normal entries and $\boldsymbol{R} = 0.35\boldsymbol{I}$.

For each run, we fit the corresponding method (either 'parallel' or 'sequential') for 20 iterations, keeping hyperparameters $\boldsymbol{\Theta}$ fixed.

**Additional results on different batch sizes**  In Figure 9, we provide additional results from runtime experiments on varying batch sizes. Here, we show wall clock time for batch sizes of 1, 20, 50, and 100 for each method. We find that for a single trial, parallel SING exhibits nearly constant scaling with sequence length, while sequential SING scales linearly in time, as expected.

For larger batch sizes, our implementations of parallel SING and sequential SING both parallelize over the batch dimension. Thus, it is expected that parallel resources on a single GPU may be

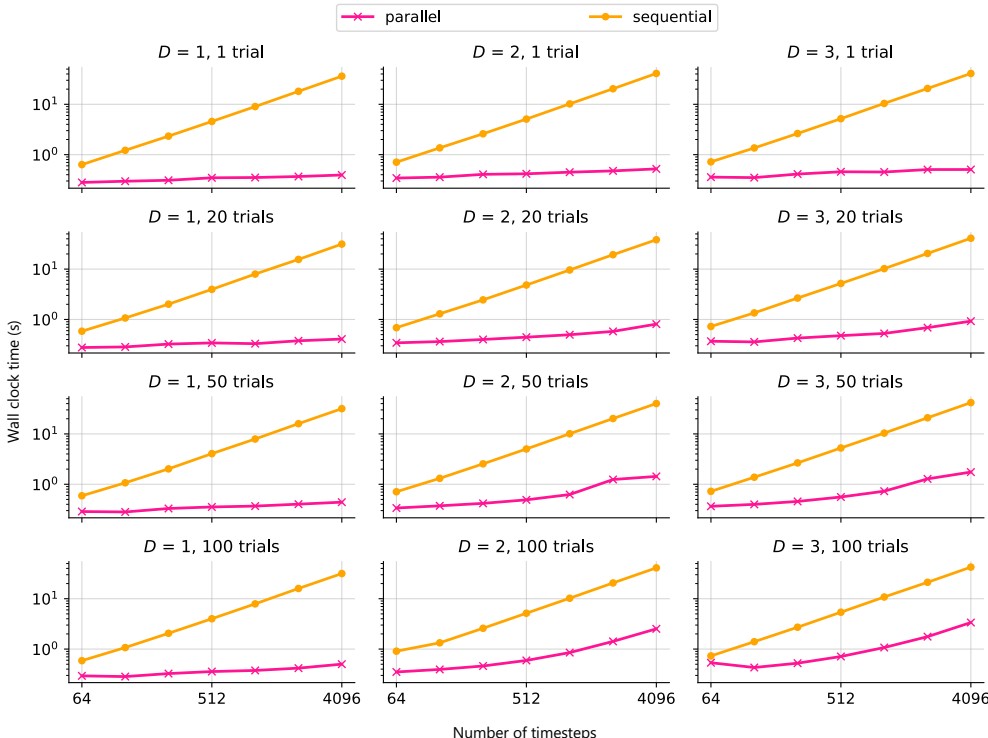

Figure 9: Runtime comparisons between parallel SING (using associative scans) and its sequential counterpart on an NVIDIA A100 GPU. Parallel SING scales favorably with sequence length compared to sequential SING on different latent dimensions $D$ and batch sizes up to 100 trials.

constrained for models that operate on both large batch sizes and long sequences. In Figure 9, we show that despite these constraints, parallel SING still scales favorably with respect to sequence length compared to its sequential counterpart for batch size up to 100 trials. We also note that our implementation of SING supports mini-batching on smaller batch sizes in order to maintain this favorable scaling behavior, even on datasets with many (i.e., hundreds or thousands of) sequences.

### L.5 Application to modeling neural dynamics during aggression

**Dataset specifications**   We apply our method to a publicly available dataset from Vinograd et al. [55], which can be found at: https://dandiarchive.org/dandiset/001037. We use the dataset labeled as 'sub-M0L' which consists of neural activity from one trial of a mouse exhibiting aggressive behavior towards two consecutive intruders. This dataset consists of calcium imaging from ventromedial hypothalamus neurons. In total, 56 neurons were imaged over 420.1 seconds at 10 Hz, resulting in a data matrix with $T = 4201$ and $N = 56$. Before fitting models, we $z$-score the data such that each neuron's activity has mean 0 and standard deviation 1.

**GP-SDE model details**   We fit GP-SDE models with kernel function chosen to be the "smoothly switching linear" kernel from Hu et al. [14]. This kernel takes the form

$$\kappa(\boldsymbol{x}, \boldsymbol{x}') = \sum_{j=1}^{J} \underbrace{\left((\boldsymbol{x} - \boldsymbol{c}_j)^\top \boldsymbol{M} (\boldsymbol{x}' - \boldsymbol{c}_j) + \sigma_0^2\right)}_{\kappa_{\mathrm{lin}}^{(j)}(\boldsymbol{x}, \boldsymbol{x}')} \underbrace{\pi_j(\boldsymbol{x}) \pi_j(\boldsymbol{x}')}_{\kappa_{\mathrm{part}}^{(j)}(\boldsymbol{x}, \boldsymbol{x}')}$$

where

$$\pi_j(\boldsymbol{x}) = \frac{\exp(\boldsymbol{w}_j^\top \phi(\boldsymbol{x})/\tau)}{1 + \sum_{j=1}^{J-1} \exp(\boldsymbol{w}_j^\top \phi(\boldsymbol{x})/\tau)}.$$

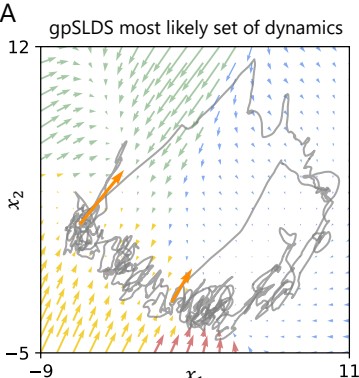
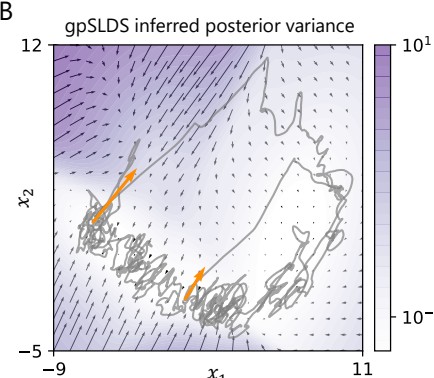

Figure 10: Additional results for the same SING-GP model fit as in Figure 6C. **A**: Inferred latent trajectory, dynamics, and learned input effect using SING-GP. The dynamics arrows are colored by most likely linear dynamical state, as learned by the smoothly switching linear kernel from Hu et al. [14]. The orange arrow represent the direction of learned inputs (i.e. columns of $\boldsymbol{B}$) capturing the effect of the intruders' entrances in the mouse's environment. As expected, these arrows explain the fast change in latent trajectories during intruder entrances. **B**: Same inferred dynanics as panel A, but the latent space is colored by inferred posterior variance of $q(\boldsymbol{f})$ from SING-GP model fit. The model expresses low uncertainty in regions with data and high uncertainty in regions without data.

Intuitively, this kernel encodes prior structure in the dynamics such that it "switches" between different linear dynamical systems based on location in latent space, but maintains smooth dynamics between different linear systems. This is enforced by the linear kernel $\kappa_{\text{lin}}^{(j)}(\boldsymbol{x}, \boldsymbol{x}')$ and partition kernel $\kappa_{\text{part}}^{(j)}(\boldsymbol{x}, \boldsymbol{x}')$ for each linear regime $j$. Above, $J$ is the total number of discrete states that the function switches between; we choose $J = 4$ and $D = 2$ to be consistent with prior work [56, 55, 14]. We also choose $\phi(\boldsymbol{x}) = \begin{bmatrix} 1 & \boldsymbol{x}_1 & \boldsymbol{x}_2 \end{bmatrix}^\top$. The kernel hyperparameters are $\boldsymbol{\theta} = \{\boldsymbol{M}, \sigma_0^2, \{\boldsymbol{c}_j, \boldsymbol{w}_j\}, \tau\}$, which we optimize via Adam [52] in variational EM. In particular, for learning we use the approach proposed by Hu et al. [14], which optimizes hyperparameters jointly with the closed-form posterior drift update. Finally, we model observations as

$$\boldsymbol{y}(t_i) \sim \mathcal{N}(\boldsymbol{C}\boldsymbol{x}(t_i) + \boldsymbol{d}, \boldsymbol{R})$$

for every observation time $t_i$. Unless otherwise specified, we fit models whose discretization matches the imaging rate; i.e., $\Delta t = 0.1$.

**Model fitting details** To fit GP-SDE models using both SING-GP and VDP-GP, we chose inducing points to be a uniformly spaced $8 \times 8$ grid in $[-8, 10] \times [-4, 12]$. These bounds were chosen by examining two-dimensional latent trajectories obtained by principal component analysis (PCA) on the neural data matrix. Similarly, we initialize $\boldsymbol{C}$ and $\boldsymbol{d}$ using PCA and keep them fixed during model fitting. We initialize kernel hyperparameters randomly from *i.i.d.* standard Gaussian distributions.

For both SING-GP and VDP-GP, we perform variational EM for 30 iterations. In each E-step we perform 15 updates (natural gradient steps for SING-GP and forward/backward passes for VDP-GP). In each M-step we perform 50 Adam iterations with initial learning rate 1e-4 to learn kernel hyperparameters.

For SING-GP, we again use a suggested learning rate schedule from [25]. In particular, we set the learning rates for the first 10 iterations to be log-linearly spaced between $10^{-1}$ and $10^0$. We then keep the $10^0$ learning rate for the rest of fitting. For VDP-GP, we use "soft updates" (eq. (32)) and manually tune $\omega$ to the largest value which still maintains numerical stability; in this case $\omega = 0.005$.

Since the smoothly-switching linear kernel from Hu et al. [14] has several hyperparameters, we repeat our experiments for 5 different random initializations of $\Theta$. In Figure 6D-E, we report average metrics and 2-standard error bars across these 5 initializations.

**Computing posterior probability of slow points** In Figure 6C, we use the probabilistic formulation of the drift in the GP-SDE to identify high-probability regions with slow points. To do this, we

compute a "posterior probability of slow point" metric for a fine grid of locations in latent space. For a particular $\boldsymbol{x}^* \in \mathbb{R}^D$, we define this as:

$$\mathbb{P}(\boldsymbol{x}^*\text{is a slow point}) = \prod_{d=1}^{D} \mathbb{P}_{q(f_d(\boldsymbol{x}^*))}[|f_d(\boldsymbol{x}^*)| < \epsilon]$$

where $\epsilon > 0$ is chosen to be a small threshold. Intuitively, this probability is large if the inferred drift at $\boldsymbol{x}^*$ is close to 0 in the posterior. Recall from Appendix K.4 that $q(f_d(\boldsymbol{x}^*))$ is a closed-form Gaussian distribution, so this quantity can be easily computed.

**Experiment details for Figure 6D** In Figure 6D, we compare SING-GP and VDP-GP on two metrics as a function of discretization $\Delta_t$ ranging from 0.06 to 0.1. We fit both methods using 100 variational EM iterations.

The first metric for model comparison is final expected log-likelihood; precisely, this is $\mathbb{E}_q[\log p(\boldsymbol{y} \mid \boldsymbol{x})]$ at iteration 100. We choose this metric to be able to directly compare the goodness-of-fit for each model; in contrast, the ELBOs are not directly comparable due to their slightly different formulations of the term $\mathrm{D}_{\mathrm{KL}}\left(q(\boldsymbol{x}) \| p(\boldsymbol{x}|\boldsymbol{f})\right)$.

The second metric for model comparison is the number of iterations until convergence. Here, we say that the model has converged at iteration $j$ if $|\mathcal{L}(q^{(j)}, \boldsymbol{\Theta}) - \mathcal{L}(q^{(j-1)}, \boldsymbol{\Theta})| < 200$ nats, where $\mathcal{L}(\cdot)$ denotes each model's respective ELBO function. We note that other choices of convergence threshold yielded similar trends as the plot in Figure 6D.

**Forward simulation experiment** Here, we provide details on the forward simulation experiment in Figure 6E. To compute the predictive $R^2$ metric in Figure 6E, we carry out the following steps.

(a) For a given model fit and discretization $\Delta t$, select 20 uniformly spaced time points $t_0$ along the inferred latent trajectory, as well as their corresponding marginal means, $\boldsymbol{m}(t_0) := \mathbb{E}_q[\boldsymbol{x}(t_0)]$.

(b) For each $t_0$ and $\boldsymbol{m}(t_0)$, use the posterior mean of inferred dynamics $q(\boldsymbol{f})$ to simulate forward the latent state for $s$ seconds using discretization $\Delta t$, where $s = 1, 2, \ldots, 15$.

(c) Forward simulation will yield a predicted latent state, $\hat{\boldsymbol{x}}(t_0 + s)$. We compute the predictive $R^2$ for each choice of $s$ as,

$$R^2 = 1 - \frac{\sum_{t_0} \|\hat{\boldsymbol{y}}(t_0 + s) - \boldsymbol{y}(t_0 + s))\|_2^2}{\sum_{t_0} \|\boldsymbol{y}(t_0 + s) - \bar{y}\|_2^2},$$

where $\hat{\boldsymbol{y}}(t_0 + s) = \boldsymbol{C}\hat{\boldsymbol{x}}(t_0 + s) + \boldsymbol{d}$, $\bar{y}$ is the mean of all observations in the trial, and the sums are taken over all 20 initial times $t_0$.

**Additional results** In Figure 10, we present additional experimental results to supplement the SING-GP inferred flow field and line attractor from the main text, Figure 6C. These results provide new angles of analysis of the SING-GP model fit. Recall from above that the smoothly switching linear kernel from Hu et al. [14] encourages smoothly switching linear dynamics to be inferred.

In Figure 10A, flow field arrows are colored by most likely linear state. That is, each set of colored arrows represents a different linear dynamical system. Encoding this structure into the kernel allows for convenient downstream analysis of these inferred dynamics, in particular using established properties of linear systems. Moreover, the orange arrows in Figure 10 represent the columns of the learned input effect matrix $\boldsymbol{B} \in \mathbb{R}^{D \times 2}$. We find that these input directions correspond to the direction of the latent trajectory at both timepoints at which an intruder enters.

In Figure 10B, we plot the inferred posterior variance of the same SING-GP model. We find that model uncertainty is low in the regions traversed by the inferred latent trajectory and high otherwise. In fact, this posterior variance is exactly the quantity that we use to compute posterior probability of slow points in Figure 6C. This demonstrates how the probabilistic formulation of the GP-SDE, which we can now perform reliable inference and learning in using SING-GP, allows us to better understand and interpret inferred flow fields.

