# OpenReview forum: "SING: SDE Inference via Natural Gradients"
_NeurIPS.cc/2025/Conference — NeurIPS 2025 poster_

### Official Review · Reviewer_c8GD · 2025-06-18

**Clarity:** 4
**Significance:** 3
**Originality:** 3
**Rating:** 5
**Confidence:** 4

**Summary:**

This paper introduces SDE Inference via Natural Gradients (SING) as a method that leverages natural gradient VI to improve inference in latent SDE models. The paper also describes a method for parallelisation, further improving efficiency and extends the approach to GP-SDEs. The paper provides extensive theoretical support and proofs as well as solid experimental support via validation on simulated and real-world data.

**Questions:**

In related work, the authors should also describe alternative approaches to latent SDE estimation based on flow matching. See e.g. https://arxiv.org/abs/2502.02472. As these methods significantly improve on VI it would be important to contrast against these. Ideally the authors show competitive performance with these alternative approaches that have been introduced to address the limitations of VI.

**Ethical Concerns:**

["NO or VERY MINOR ethics concerns only"]

**Final Justification:**

I would like to thank the authors for their response and additional experiments. This paper is an important contribution to the community.

**Limitations:**

The assumption that the diffusion term is known can be acknowledged as a limitation if this cannot be addressed in the current setup.

**Paper Formatting Concerns:**

No.

**Quality:**

4

**Strengths And Weaknesses:**

This paper provides a solid contribution to the field by introducing SING as a way to enable fast and reliable inference in latent SDE models. The claims are supported by solid proofs. The parallel approach yields convincing improvements. The experimental validation clearly demonstrate its benefits compared to methods like VDP. The paper is very well written in general.

There are a couple of points that warrant attention.

The diffusion term Sigma is assumed given. However, it can also be considered a model parameter. Can the theory be extended to incorporate fitting of Sigma? If yes, I suggest to incorporate this. If no, please explain why and explain why this is not an issue when fitting models on real-world data where the process noise is unknown.

The authors state that the method is closest to that of Verma et al. However, comparisons are with VDP. While the authors clearly describe the differences in the appendix I would have expected an explicit comparison in terms of empirical results. Please add these comparisons to the plots or show this using a targeted comparison in the appendix or make more explicit why such a comparison is irrelevant.

---

> ### Author Rebuttal · Authors · 2025-07-31
>
> We thank the reviewer for taking the time to read our submission and for their thoughtful response! We are especially glad that the reviewer appreciated the theoretical and empirical contributions of our method, and found our paper to be well-written.
>
> ## Weaknesses
>
> > Assumptions on the diffusion term $\Sigma$
>
> We thank the reviewer for their comment and point out that for the purpose of inference, SING can handle arbitrary fixed, positive-definite diffusion coefficient $\Sigma^{1/2}$. Moreover, when learning the output parameters $C, d$ together with the prior drift $f$, the diffusion coefficient should be fixed to $\Sigma^{1/2} = I$ to ensure identifiability of the model.
>
> Indeed, suppose our prior over latents is $p(x): dx(t) = f(x(t)) dt + \Sigma^{1/2} dw(t)$. Then by applying the transformation $z(t) = \Sigma^{-1/2}x(t)$, the generative model becomes $p(z): dz(t) = \Sigma^{-1/2} f(\Sigma^{1/2} z(t)) dt + dw(t)$, $\mathbb{E}[y(t)] = g(C\Sigma^{1/2} z(t) + d)$. And the posterior process becomes $q(z(t)): dz(t) = \Sigma^{-1/2}( A(t) \Sigma^{1/2} z(t) + b(t) )dt +dw(t)$, which is again a linear, time-varying SDE. Moreover, it is a standard result (see e.g., [1] Lemma E.2.1) that the continuous-time ELBO $\mathcal{L}_{\text{cont}}$ remains invariant under invertible transformations. As a consequence, for any prior process with non-white noise $\Sigma^{1/2}$, one can always perform an appropriate transformation to whiten the noise, perform approximate inference with SING, and then apply the inverse transformation to recover the inferred latents in the original space.
>
> If model class remains the same under the mapping $f \to \Sigma^{-1/2} f(\Sigma^{1/2})$ (e.g., neural networks, linear maps, linear combination of polynomial basis functions) and $C, d$ are learned together with the drift $f$, then the prior noise $\Sigma$ is non-identifiable in the model. For this reason, we fix $\Sigma = I$ when performing learning.
>
> Lastly, we point out that for certain state-dependent diffusion coefficients, it remains possible to perform both learning and inference with SING. This is done using a similar approach to that outlined above: whitening the noise of the prior process and variational posterior, applying inference and learning with SING, and then performing the inverse transformation. Precisely, we transform the prior SDE using the Lamperti transform (see e.g., [2] Section 3.6).
>
> > Comparison to Verma et al., 2024
>
> We appreciate the reviewer’s suggestion and agree that an empirical comparison with Verma et al., 2024 would be ideal and have explored this direction. However, we encountered persistent issues running their publicly released code. We carefully followed the installation instructions provided in their repository and additionally attempted several manual interventions, such as modifying version constraints and testing across different environments. Despite these efforts, we were unable to get the code running due to unresolved dependency conflicts. We have alerted the authors to inform them of this issue.
>
> In light of this, we provide a more explicit comparison between SING and Verma et al., 2024 which we will also include in the appendix. Verma et al., 2024 propose an inference algorithm that repeatedly factorizes the prior on latents as $p(x) = p_L(x) v(x)$ where $p_L$ is a linear SDE and $v(x) = p(x) / p_L(x)$. During inference, they alternate between this factorization, performing natural gradient descent to update the variational posterior and applying a statistical linearization step to determine a new $p_L$. In contrast, SING avoids these repeated linearization and factorization steps by leveraging the fact that NGVI is parametrization invariant; that is, the particular factorization of the generative model does not affect the resulting updates. As a result, SING achieves equivalent inference while reducing computational cost.
>
> Moreover, SING introduces a novel theoretical contribution not present in Verma et al., 2024: a bound on the approximation error between the discretized ELBO optimized in practice and the true continuous-time ELBO. This result provides theoretical justification for applying SING to continuous-time latent SDE models, bridging an important gap between practice and theory that was not addressed in prior work.
>
> We also acknowledge that Verma et al., 2024 propose a modified M-step that couples inference and learning by updating model parameters and variational parameters simultaneously. While this approach may reduce the number of M-step iterations required for convergence, it increases the computational cost per iteration. More importantly, the practical benefits of this approach are difficult to assess without empirical comparison, which is not feasible at this time due to the technical limitations described above.
>
> Finally, we would like to emphasize that SING introduces several additional practical and computational improvements which we believe will make our method more readily usable and scalable in real-world applications. These include simpler but equivalent natural gradient updates during inference, parallelizable updates which scale logarithmically with respect to sequence length, and tractable computations for expectations in the ELBO. We have developed and plan to release a modular and efficient codebase that implements SING and its extensions, including support for Gaussian process drift models.
>
> ## Questions
>
> > Comparison to SDE Matching
>
> We thank the reviewer for their suggestion to compare SING to SDE Matching [3]. SDE Matching performs amortized variational inference by using normalizing flows to model the posterior marginals. These marginals are not necessarily Gaussian. This is similar to another related work [4], which still assumes Gaussian posterior marginals but also applies an amortized posterior encoder.
>
> We compare the methods on the stochastic Lorenz attractor benchmark dataset, in which 3D latents evolve according to Lorenz attractor dynamics on $t \in [0, 1]$ with parameters $(\rho, \sigma, \beta) = (28, 10, 8/3)$ and diffusion coefficient $0.15$. Observations are made every $\Delta t = 0.025$ are generated as $y(t) = \mathcal{N}(Cx(t) + d, 0.01^2)$. The full experimental setup is detailed in [5]. For SDE Matching, we used the authors’ codebase. We compare SING with $10^3$ vEM iterations, each consisting of 10 SING E-steps and 10 M-steps, to SDE Matching with $10^4$ iterations; the number of gradient updates is the same for both methods. SING updates are performed with a single Monte Carlo sample. We model the prior for both methods with a 3D neural-SDE, whose drift is a neural network with softplus activation.
>
> We compare the methods using two metrics: the RMSE between the inferred and ground truth latent (‘latents RMSE’) and the root mean squared relative error between the learned prior drift and ground truth Lorenz attractor dynamics (‘dynamics RMSE’).
>
> **Latents RMSE**
> | n_trials | SING  | SDE Matching |
> |----------|-------|--------------|
> | 10       | 0.443 | 1.173        |
> | 30       | 0.443 | 0.807        |
> | 50       | 0.442 | 0.711        |
> | 100      | 0.440 | 0.719       |
> | 300      | 0.440 | 0.637       |
> | 500      | 0.440 | 0.567       |
> | 1024     | 0.440 | 0.549       |
>
> **Dynamics RMSE**
> | n_trials | SING  | SDE Matching |
> |----------|-------|--------------|
> | 10       | 0.272 | 0.551        |
> | 30       | 0.265 | 0.329        |
> | 50       | 0.300 | 0.232        |
> | 100      | 0.265 | 0.239       |
> | 300      | 0.330 | 0.179       |
> | 500      | 0.270 | 0.161       |
> | 1024     | 0.297 | 0.164      |
>
> We observe that SING outperforms SDE Matching on both metrics in data-limited settings ($< 30$ trials). Moreover, even up to $1024$ trials, SING recovers the latents more accurately than SDE Matching. We hypothesize that this is because SING explicitly models the entire latent trajectory during inference, whereas SDE Matching performs updates using the inferred posterior at a single time point. However, SING is not able to effectively refine its prior estimate of Lorenz dynamics as the number of trials increases; we will investigate improvements to learning with SING following the rebuttal period.
>
> **Thank you again for your positive comments and thoughtful response!** If you have any further questions, we would be happy to answer them.
>
> – The SING authors
>
> [1] Dupuis and Ellis. A weak convergence approach to the theory of large deviations. John Wiley and Sons, 1997.
> [2] Pavliotis. Stochastic processes and applications. Texts in Applied Mathematics, 2014.
> [3] Bartosh et al. SDE Matching: Scalable and simulation-free training of latent stochastic differential equations. ICML, 2025.
> [4] Course and Nair. Amortized reparametrization: efficient and scalable variational inference for latent SDEs. NeurIPS, 2023.
> [5] Li et al. Scalable gradients for stochastic differential equations. AISTATS, 2020.

---

### Official Review · Reviewer_Q7Cf · 2025-06-27

**Clarity:** 4
**Significance:** 2
**Originality:** 2
**Rating:** 4
**Confidence:** 3

**Summary:**

The authors consider a method for estimating latent stochastic differential equations (SDEs) through a natural gradient estimator. The natural gradient provides gradient updates that adhere to the geometry of the KL divergence. The model is assumed to have observations of a known link function applied to an affine transformation of a latent SDE, and the objective is to recover the affine transformation as well as the posterior latent distribution. The authors consider a family of linear SDEs for the variational distribution, which provides Gaussian time marginals and is amenable for use with natural gradient variational inference. The authors test the proposed method on a number of different experiments and illustrate the performance of the natural gradient estimator versus baselines.

**Questions:**

How limiting is the variational distribution being a linear SDE? For example, since we approximate objects of the form $g(Cx+d)$, this would imply a limitation on the volatility, for example, of the process of $y$.

Is computing the Jacobian in the ELBO expensive? Can the authors describe some more details on this.

Is there a way to estimate the parameters of $g$ using this method?

For other SDEs (for example, GBM) where the volatility is a function of state, are there other ways to extend the method to these cases? The GBM case has log normal marginal distributions, so I assume something similar could work by reformulating in terms of the parameters of a log normal, but is this generally extensible?

Many of the experiments suggest that the true drift can be recovered but I was wondering if a more concrete theoretical statement could be made regarding this? Along the lines of the previous questions, how does the constrained variational distribution affect this?

**Ethical Concerns:**

["NO or VERY MINOR ethics concerns only"]

**Final Justification:**

After reading the response, I believe the method provides useful insights into the some topics regarding estimating SDEs and would be an interesting addition to the conference. I am still not certain about the novelty of the method since the main technical results are largely combinations of existing work. Additionally, requiring an SDE to satisfy the Lamperti transform can be quite difficult, I think for a general SDE estimation method this is something that should be studied in greater detail.

**Limitations:**

Yes

**Quality:**

3

**Strengths And Weaknesses:**

**Strengths:**

The authors provide a bound on the ELBO under a discrete setting to the continuous time setting that has the same order of convergence as the Euler Maruyama method.

The method empirically displays good performance on a variety of tasks. A number of comparisons are particularly impressive, for example the Adam vs NGVI plots are very informative.

The derivation of this formulation is neat and a slightly different take than many other papers along the lines of latent SDEs.

**Weaknesses:**

The authors mention a variety of related work but do not consider comparisons to them. It would be interesting to see how these methods compare in both computation speed and memory usage.

It would be useful to get more context as to where using the natural gradient is more appropriate versus using standard gradient descent with an ELBO with a more flexible variational distribution.

The variational distribution being a linear SDE to be a major limitation especially since the volatility needs to be constrained quite heavily. On that note, the main techniques involve making the variational distribution Guassian and applying NGVI is not entirely original.

This field has a lot of related work and has been well studied from a variety of different angles, so it would be good to get more context on the contributions made in the paper compared with prior work. While this work focuses on the natural gradient variational inference under a linear SDE variational distribution for discovering SDE parameters, others have considered adjacent strategies. Just as an example of some related work missing:

Latent SDEs on Homogeneous Spaces, NeurIPS 2023

Neural Structure Learning with Stochastic Differential Equations, ICLR 2024

Identifying Latent Stochastic Differential Equations, IEEE Transactions on Signal Processing 2021

Learning interpretable dynamics of stochastic complex systems from experimental data, Nature Communications 2024

Computing high-dimensional invariant distributions from noisy data, Journal of Computational Physics 2023

---

> ### Author Rebuttal · Authors · 2025-07-31
>
> We thank the reviewer for their positive and thoughtful response! We especially appreciate the reviewer’s comments on the derivation of our ELBO bound and the strong empirical performance of SING across tasks.
>
> ## Weaknesses
> > Comparisons to related work, in particular computation speed and memory usage
>
> We have conducted additional experiments comparing SING to SDE Matching [1], a recent work that uses amortized normalizing flows to model posterior marginals, which are not necessarily Gaussian. This is similar to [2], which still assumes Gaussian posterior marginals but also applies an amortized encoder.
>
> We compare the methods on a stochastic Lorenz benchmark [3], where 3D latents evolve on $t \in [0, 1]$ with $(\rho, \sigma, \beta) = (28, 10, 8/3)$ and diffusion $0.15$. Observations are generated every $\Delta t = 0.025$ as $y(t) = \mathcal{N}(Cx(t) + d, 0.01^2)$. For SDE Matching, we used the authors’ codebase. We ran SING for $10^3$ vEM iterations (10 E-steps, 10 M-steps each) and SDE Matching for $10^4$; the number of gradient updates is the same. Both use a neural-SDE prior with softplus activation. We evaluate methods using the RMSE between the inferred and ground truth latents (‘latents RMSE’) and between the learned prior drift and ground truth dynamics (‘dynamics RMSE’).
>
> **Latents RMSE**
> | n_trials | SING  | SDE Matching |
> |----------|-------|--------------|
> | 10 | 0.443 | 1.173|
> | 30 | 0.443 | 0.807|
> | 50 | 0.442 | 0.711|
> | 100 | 0.440 | 0.719|
> | 300 | 0.440 | 0.637|
> | 500 | 0.440 | 0.567|
> | 1024 | 0.440 | 0.549|
>
> **Dynamics RMSE**
> | n_trials | SING  | SDE Matching |
> |----------|-------|--------------|
> | 10 | 0.272 | 0.551|
> | 30 | 0.265 | 0.329|
> | 50 | 0.300 | 0.232|
> | 100 | 0.265 | 0.239|
> | 300 | 0.330 | 0.179|
> | 500 | 0.270 | 0.161|
> | 1024 | 0.297 | 0.164|
>
> SING outperforms SDE Matching in data-limited settings ($< 30$ trials) and recovers latents more accurately in up to $1024$ trials. We hypothesize that this is because SING explicitly models the entire latent trajectory during inference, whereas SDE Matching performs updates using the inferred posterior at a single time point. However, SING does not effectively refine its prior estimate of Lorenz dynamics with more trials; we will investigate improvements to learning with SING following the rebuttal period.
>
> SDE Matching is preferable in runtime and memory, as it updates the amortized posterior at individual time points. SING’s complexity depends on $T$, the length of the time grid at which inference is performed. Still, SING parallelizes its updates across $T$, yielding $\mathcal{O}(D^3 \log T)$ complexity.
>
> > More context on natural vs. standard gradient with a more flexible variational distribution
>
> SING’s variational posterior factorizes across trials $i$ as $q = \prod_{i=1}^N q(x^{(i)})$, where each $q(x^{(i)})$ is represented by a linear time-varying SDE with parameters $A^{(i)}(t), b^{(i)}(t)$. Consequently, if the latent trajectories can be well-explained by per-trial locally linear dynamics, then using a linear time-varying SDE as the variational posterior is reasonable. In this case, SING exploits the exponential family structure of the problem to achieve faster convergence than standard gradient descent or even Adam, as we showed in our experiments. Moreover, in data-limited regimes, using a linear SDE could be more statistically efficient relative to more flexible variational posteriors. That said, for latent SDEs exhibiting sharp nonlinearities, more flexible variational distributions may yield better estimates, provided enough data is available.
>
> > Linear SDE is a major limitation, especially since the volatility needs to be constrained
>
> As mentioned above, the variational posterior factorizes across trials $i$. Thus, as long as each trajectory $x^{(i)}$ is well-approximated by time-varying linear dynamics, we would expect our variational approximation to be accurate. This claim is supported by our experiments showing that SING performs accurate inference in the Duffing oscillator (Fig 3) and stochastic Lorenz attractor.
>
> As to volatility, we outline our approach to allowing for state-dependent diffusion coefficients below.
>
> > Main techniques are not entirely original
>
> We agree with the reviewer that natural gradient methods have been well-studied. The goal of our work is to develop a robust and scalable natural gradient method for latent SDEs, along with theoretical guarantees on its approximation error to the continuous-time setting. Our contribution lies in deriving closed-form natural gradient updates specifically for latent SDEs, enabling parallelization of these updates for scalability to long sequences, providing theoretical bounds on the ELBO approximation error, and extending this approach to inference in models with nonparametric (Gaussian process) drift.
>
> > Additional related work
>
> We thank the reviewer for bringing these to our attention and will include them in our revised paper.
>
> ## Questions
>
> > Limitation on the volatility of the process of $y$
>
> While the variational posterior of $x(t)$ is assumed to be a linear SDE, its learned time-dependent linear coefficients $A(t)$ allow the marginal distribution of $y(t)$ to have variance that varies in time. This is because the marginal variance of $x(t)$ evolves according to the ODE $dS(t) = A(t)S(t) + S(t)A(t)^\top + \Sigma$.
>
> > Is computing the Jacobian in the ELBO expensive?
>
> The ELBO does not require computing a Jacobian. The author may be referring to the naive NGVI update (eq. 4), which requires computing the inverse Fisher information matrix, i.e. the Jacobian of the mapping from natural to mean parameters. Naively, this would have complexity $O(D^3 T^3)$. However, we do not actually compute the inverse Fisher explicitly; due to the exponential family structure of our variational posterior, the NGVI update can be equivalently written as eq. (6). We show that SING can compute these updates in $O(D^3 \log T)$ time, enabling scalability to long time series.
>
> > Is there a way to estimate the parameters of $g$?
>
> $g$ is a pre-determined inverse link function, but the parameters $C$ and $d$ of the affine mapping can be learned.
>
> > For other SDEs (for example, GBM) where the volatility is a function of state, are there other ways to extend the method to these cases?
>
> We thank the reviewer for their insight regarding how SING can be applied to latent-SDE models with state-dependent diffusion. The example of the GBM can indeed be generalized to a larger class of state-dependent diffusion coefficients via the Lamperti transform (e.g., [4] Section 3.6). For a 1D process $dx(t) = f(x(t)) dt + \sigma(x(t)) dw(t)$, the Lamperti transform is defined as
> $$
> h(x) = \int_{x_0}^x 1/\sigma(s) ds
> $$
> for $x_0$ an arbitrary fixed point in the state space. Applying Itô’s rule to the map $z(t) = h(x(t))$ and process $x(t)$ yields a diffusion process with identity noise and modified drift. The observation model also transforms accordingly through $h^{-1}$.
>
> Once we have transformed the prior and likelihood models, we can leverage SING to perform inference in the transformed space $z$. One can then apply Itô’s rule to $h^{-1}(z(t))$ to recover the approximate posterior in the original space $x$. Because the continuous-time ELBO $\mathcal{L}_{\text{cont}}$ remains invariant under reparameterization (e.g., [5] Lemma E.2.1), the optimization problem in the transformed space is equivalent to that in the original space.
>
> We note that a Gaussian process variational approximation in the transformed space $z$ is not appropriate when the Lamperti transform takes values in only a subset of $\mathbb{R}$. For example, for the Cox-Ingerson-Ross equation $dx(t) = a(b - x(t)) dt + \sigma \sqrt{x(t)}dw(t)$, the Lamperti transform $h(x) = \frac{2}{\sigma} \sqrt{x}$ takes nonnegative values.
>
> We conducted additional experiments validating our proposed approach on latents that evolve according to the 1D GBM $dx(t) = \mu x(t) dt + \sigma x(t) dw(t)$ on $[0, 2]$ with $\mu = 1$ and $\sigma = 1$. We have observations at every $\Delta t = 0.02$ that are equal to the latents perturbed by Gaussian noise with SD $0.1$, and we use $1024$ trials. We choose our prior to be a GBM with unknown $\mu$, initialized at $0$. The Lamperti transform is $h(x) = \frac{1}{\sigma}\log(x)$. We fix the output parameters $C, d$ as well as $\sigma$ and aim to perform inference and learn $\mu$. We find that SING recovers the latents in the true latent space ($x$) with MSE $(0.084)^2$ as well as $\hat{\mu} \approx 1.08$.
>
> > More concrete theoretical statement on true drift recovery? How does the constrained variational distribution affect this?
>
> We agree that a more concrete theoretical analysis on drift recovery would be valuable, particularly with respect to the choice of variational family. While our work focuses primarily on empirical drift recovery rather than formal guarantees, we note that the choice of a linear time-varying SDE can still capture nonlinearities in the latent processes (as above) and encourage learning of a nonlinear drift.
>
> **Thank you again for your positive comments and thoughtful response!** If you have any further questions, we would be happy to answer them.
>
> – The SING authors
>
> References:
> [1] Bartosh et al. SDE Matching: Scalable and simulation-free training of latent stochastic differential equations. ICML, 2025.
> [2] Course and Nair. Amortized reparametrization: efficient and scalable variational inference for latent SDEs. NeurIPS, 2023.
> [3] Li et al. Scalable gradients for stochastic differential equations. AISTATS, 2020.
> [4] Pavliotis. Stochastic processes and applications. Texts in Applied Mathematics, 2014.
> [5] Dupuis and Ellis. A weak convergence approach to the theory of large deviations. John Wiley and Sons, 1997.

---

### Official Review · Reviewer_bEC3 · 2025-07-01

**Clarity:** 4
**Significance:** 3
**Originality:** 3
**Rating:** 5
**Confidence:** 3

**Summary:**

The authors propose to use natural gradient VI for inference in latent SDEs, proposing an algorithm SING where a variational family of linear SDEs is prescribed over the latent dynamics, and the parameters of the drift and link function are learnable. The authors provide several numerical exploits to speed up inference with parallelization and provide illustrative and quantitative examples of their method in simulated and real data.

**Questions:**

- I am concerned with the linear family for the posterior drift, since this would preclude dynamics that for instance exhibit multiple modes. Similarly, it seems to me that if the prior SDE allow for bistable behavior (e.g. bifurcations in the trajectories), using a linear variational family cannot model this.

- On the other hand, the prior family on the drift could admit bifurcations for some values of $\theta$. How does this reconcile with the linear variational family for the latent SDE? In particular, should one expect any connection between (1) the latent linear SDE learned by NGVI and (2) the learned vector field?

- The authors mention several related works that use more general families of processes for inference of trajectories. Presumably these are more computationally intensive or less accurate overall. However, are there some scenarios (such as bifurcations) where more flexible model classes may help? Some discussion of this would be valuable I think.

- Can anything be said in the case where the diffusion coefficient has some dependency on the state $x$? For instance, SDEs of the Cox-Ingersoll-Ross type. Such systems may have applications in some biophysics domains, e.g. [1]

- As I understand it, SING infers a (variational) posterior over latent trajectories, but MAP estimate for the parameters $\Theta$. Is extension to the fully Bayesian setting straightforward, or would it require new techniques?

- For the question on the dimensionality of the latent dynamics, I think it would be valuable (and potentially not too much work) for some empirical results comparing the use of Gaussian quadrature to Monte Carlo for estimation of expectations. It seems to me that the most glaring limitation in the present work is that all examples are 2D; at least in the synthetic examples it would be good to see some results for moderate dimensions (2-5 dims).


[1] Frishman A, Ronceray P. Learning force fields from stochastic trajectories. Physical Review X. 2020 Apr 1;10(2):021009.

**Ethical Concerns:**

["NO or VERY MINOR ethics concerns only"]

**Final Justification:**

The authors' rebuttal helped clarify a key misunderstanding I had about the work, namely that the linear SDE parameters are learned for each trajectory rather than across trajectories. I am also satisfied with the additional results on scaling up to higher latent dimensions, the results look very promising and would aid the paper in making its point. I therefore keep original high score.

**Limitations:**

yes

**Quality:**

4

**Strengths And Weaknesses:**

Strengths:
The paper is well written and compelling. It provides a pedagogical introduction to Bayesian inference problems for SDEs, natural gradient VI, etc.

The formulation of SING is rigorous and the use of a linear SDE variational family clearly leads to a statistically efficient and elegant model. This is evident from the mathematical derivations as well as the numerical experiments.

The extension to SING-GP is a nice addition and in my view it significantly broadens the relevance of the work, since in many applications the underlying class of drift functions would not be known.

The numerical results presented are polished and clearly illustrate the favourable behavior of the proposed algorithm.

Weaknesses:

In all the examples shown, the latent dynamics are two-dimensional. The authors mention that this is a limitation of their method. I understand that this arises from the need to compute expectations in dimension $d$. The authors mention they use Gaussian quadrature, but it would be good to have an idea of what would happen if Monte Carlo estimation were used instead. The estimates would be unbiased, so some commentary on whether this stochastic approach is likely to work would be nice.
It is furthermore unclear from the related work section whether the baseline methods would also have the same scalability issues to higher dimensional problems.

---

> ### Author Rebuttal · Authors · 2025-07-31
>
> We thank the reviewer for taking the time to read our submission and for their thoughtful response! We especially appreciate the comments highlighting our work as “a pedagogical introduction” to the topic, as well as the rigor and applicability of our approach.
>
> ## Weaknesses
> > The authors mention they use Gaussian quadrature, but it would be good to have an idea of what would happen if Monte Carlo estimation were used instead.
>
> We thank the reviewer for their suggestion and have conducted additional experiments demonstrating that our method scales well with increasing latent dimension by using Monte Carlo.
>
> We apply SING with Monte Carlo to systems with latent dimension $D = 3, 5, 10, 20, 50$. For all of these settings we simulate ground-truth latents from an “embedded Lorenz attractor” whose first 3 dimensions evolve according to an SDE $dx(t) = f(x(t)) dt + dw(t)$ where $f(x)$ is a Lorenz attractor with parameters $(\rho, \sigma, \beta) = (28, 10, 8/3)$, and whose other dimensions $d > 3$ evolve according to a random walk with identity diffusion. Then, we simulate observations $y_n(t) \sim \mathcal{N}(c_n^\top x(t) + d_n, (\sqrt{0.05})^2)$ for $n = 1, \dots, N$, where $N$ is the observation dimension. Here, the entries of $c_n$ and $d_n$ are sampled i.i.d from $\mathcal{N}(0, 0.1^2)$. We fit SING using a prior SDE in the same class as the generative model and learn the drift parameters. We report latents RMSE, dynamics RMSE, and total runtime (including compilation time) on an A100 NVIDIA GPU, along with their 95% confidence intervals obtained from 5 different initial seeds for Monte Carlo sampling. All results are obtained by using a single sample per expectation evaluation.
>
> | latent dim ($D$) | latents RMSE | dynamics RMSE|runtime (s)|
> |------------|---------------------|----------------------|--------------------|
> | 3 | 0.1416 ± 0.0001| 0.1241 ± 0.0107     | 58.05 ± 0.16     |
> | 5| 0.1867 ± 0.0001| 0.1413 ± 0.0055     | 60.11 ± 0.36   |
> |10| 0.2721 ± 0.0001| 0.3102 ± 0.0059     | 89.20 ± 0.09    |
> | 20| 0.3971 ± 0.0001| 0.2043 ± 0.0048     | 136.28 ± 0.18     |
> | 50| 0.6761 ± 0.0001| 0.2193 ± 0.0059     | 561.44 ± 0.63     |
>
> Gaussian quadrature (with $6$ quadrature nodes per dimension) achieves latents and dynamics RMSE $0.1415$ and $0.1220$ in $D = 3$ latent dimensions and $0.1866$ and $0.1337$ in $D = 5$ latent dimensions.
>
> We observe that SING applied with Monte Carlo recovers the true latents and dynamics with comparable accuracy to Gaussian quadrature in $3$ and $5$ latent dimensions. Although latents RMSE continues to increase with $D$ beyond $5$ latent dimensions, this is expected for two reasons: (1) the error is computed over higher-dimensional vectors, and (2) accurate inference becomes more challenging as latent dimension increases. Additionally, we find that the runtime of joint inference and learning of SING remains tractable in up to 50 latent dimensions.
>
> > Would the baseline methods from related work would have the same scalability issues to higher-dimensional problems.
>
> Several related works on latent SDE inference already use sampling-based approaches to estimate intractable expectations, which allows them to scale to higher-dimensional settings (e.g., [1], [2]). We agree that Monte Carlo estimation is a natural and effective choice in this context, and in our additional results above we show that this can be seamlessly integrated into the SING variational inference algorithm. As a result, we expect SING (with Monte Carlo) to exhibit similar scaling behavior to these related methods. We will revise our manuscript to include this Monte Carlo extension and its implementation details.
>
> ## Questions
>
> > Choice of linear posterior drift and whether it allows for inference in bifurcating systems. How does linear variational family reconcile with the prior SDE, which could admit bifurcations?
>
> The reviewer is correct in noting that the prior SDE need not be linear and indeed could exhibit bifurcations, while the variational posterior is chosen to be a time-dependent linear SDE, which necessarily corresponds to a Gaussian process. This mismatch is reconciled by the fact that the prior SDE is shared across trials, whereas the variational posterior is not.
>
> Specifically, under the generative model the latent trajectories are *i.i.d.* draws $x^{(i)} \sim p(x)$ from the prior SDE. Because the observation likelihood also factorizes across trials, then the joint distribution of $N$ trials can be represented as $ \prod_{i=1}^N p(x^{(i)}) p(y^{(i)} | x^{(i)})$. This implies that the true (intractable) posterior distribution also factorizes across trajectories. Hence, we construct our variational approximation as $q = \prod_{i=1}^N q(x^{(i)})$, where each $q(x^{(i)})$ is represented by a linear, time-varying SDE with parameters $A^{(i)}(t)$ and $b^{(i)}(t)$.  And so, although each $q(x^{(i)})$ is linear, the collection $(q(x^{(i)}))_{i=1}^N$ can represent multiple modes induced by a nonlinear/bistable system.
>
> > Should one expect any connection between (1) the latent linear SDE learned by NGVI and (2) the learned vector field?
>
> The latent linear time-varying SDE from NGVI is a trial-specific linear approximation of the posterior drift; the learned vector field (prior drift) is global and shared. The ELBO term $\sum_{i=1}^N \text{KL}(q(x^{(i)}) || p(x^{(i)} |  \Theta))$ encourages the learned vector field to explain the dynamics of the inferred variational posterior for each trial.
>
> > However, are there some scenarios (such as bifurcations) where more flexible model classes may help?
>
> For the reasons we detailed in our previous response, we find that SING performs comparably to methods with more flexible methods variational families, even on examples where the underlying latent dynamics are bifurcating (e.g., the Lorenz attractor).
>
> We compare SING to SDE Matching [2] on the stochastic Lorenz attractor dataset [3] with 3D neural-SDE prior and Gaussian observation model. SDE Matching performs amortized variational inference by using a normalizing flow to model the posterior marginals. When trained on $1024$ trials, i.e. latent trajectories, with an equal number of gradient steps, SING recovers the underlying latents more accurately than SDE Matching (RMSE $0.440$ versus $0.549$). The neural network prior recovers the Lorenz attractor dynamics slightly worse than with SDE Matching (root mean square relative error $0.297$ versus $0.164$). In the data-limited regime ($< 30$ trials), SING outperforms SDE Matching on both metrics.
>
> > Can anything be said in the case where the diffusion coefficient has some dependency on the state $x$?
>
> We agree with the reviewer that latent-SDE models with state-dependent diffusion coefficients are of practical relevance, and we detail how SING can be applied in this setting. To “whiten” the noise of the state-dependent prior process, one can apply the Lamperti transform $h$, $z(t) = h(x(t))$ (see e.g., [4] Section 3.6). We propose applying inference, with SING, and learning in the transformed latent space $z$. One can then apply Itô’s rule to recover the variational posterior in the original space $x$, which may not be a Gaussian process.
>
> As to the reviewer’s example of the Cox-Ingersoll-Ross equation, $dx(t) = a(b - x(t)) dt + \sigma \sqrt{x(t)}dw(t)$, the Lamperti transform is $h(x) = \frac{2}{\sigma} \sqrt{x}$, which takes nonnegative values. And so the linear, time-varying SDE variational approximation made by SING is not appropriate to approximate the posterior of $z(t) = h(x(t))$.
>
> We validate our proposed approach on the one-dimensional geometric Brownian motion $dx(t) = \mu x(t) dt + \sigma x(t) dw(t)$ on $[0, 2]$, which has Lamperti transform $h(x) = \frac{1}{\sigma}\log(x)$ taking values in $\mathbb{R}$. We sample $1024$ latent trials with $\mu = 1$ and $\sigma = 1$, and observations at every $\Delta t = 0.02$ that are equal to the latents perturbed by Gaussian noise with standard deviation $0.1$. We choose our prior to be a geometric Brownian motion with unknown $\mu$, initialized at $0$. We fix the output parameters $C, d$ as well as $\sigma$ and aim to (i) perform inference with SING and (ii) recover $\mu$. We demonstrate that SING recovers the latents in the true latent space ($x$) with MSE $(0.084)^2$ as well as $\hat{\mu} \approx 1.08$.
>
> > Is extension to posterior inference over $\theta$ straightforward?
>
> Yes, extension of SING to perform posterior inference over parameters $\Theta$ is straightforward.
>
> One could put a prior $p(\Theta)$ on model parameters, and use a variational family that factorizes over parameters and latents, $q = q(\Theta) q(x)$. Then, one could alternate between inferring $q(x)$ using SING and inferring an approximate posterior over parameters using gradient descent (or natural gradient descent, if possible) over $\Theta$.
>
> We also note that, if the goal is to get an approximate posterior over the drift, one could use SING-GP to solve this problem. SING-GP models the drift function using Gaussian processes, and it allows for uncertainty quantification over the dynamics directly in function space as opposed to in model parameter space. This approach could be a better alternative than Bayesian inference over drift parameters $\Theta$ if the true drift of the system is unknown or if the parameter space $\Theta$ is high-dimensional.
>
> **Thank you again for your positive comments and thoughtful response!** If you have any further questions, we would be happy to answer them.
>
> – The SING authors
>
> References:
> [1] Course and Nair. Amortized reparametrization: efficient and scalable variational inference for latent SDEs. NeurIPS, 2023. [2] Bartosh et al. SDE Matching: Scalable and simulation-free training of latent stochastic differential equations. ICML, 2025. [3] Li et al. Scalable gradients for stochastic differential equations. AISTATS, 2020. [4] Pavliotis. Stochastic processes and applications. Texts in Applied Mathematics, 2014.

---

> > ### Comment · Reviewer_bEC3 · 2025-08-03
> >
> > Thank you for the response to my questions, indeed I had misunderstood that the linear SDE parameters are learned trajectorywise. Thank you for the additional results on scaling to higher dimensions with Monte Carlo, I think these would be very helpful to include in the paper. I will keep my high score.

---

### Official Review · Reviewer_ma3P · 2025-07-03

**Clarity:** 4
**Significance:** 3
**Originality:** 3
**Rating:** 5
**Confidence:** 5

**Summary:**

The paper introduces a new method for inferring (latent) stochastic differential equations based on variational inference of natural parameters of probability densities represented in terms of families of natural parameters.


The work builds on the natural-gradient formulation of variational inference and extends existing literature in two directions. First, it provides a derivation for the closed-form natural-gradient updates for a latent SDEs discretised with the Euler–Maruyama scheme, exploiting the Gaussian structure of the discretised state marginals. Second, it introduces a parallelisation scheme for converting natural-to-mean parameters
via associative scans that reduce the computational cost from $\mathcal{𝒪}(D^3 T)$ to $\mathcal{𝒪}(D^3 log T)$ enabling thereby learning of long sequences.

The authors prove that the evidence lower bound (ELBO) computed on a finite grid converges to its continuous-time counterpart linking thereby the discrete-time and continuous-time objectives.

The authors validate their approach empirically on two synthetic datasets: one created with a latent linear dynamical system with Gaussian observations, and one with latent dynamics from a Van der Pol oscillator coupled with Poisson observations. In both settings, the proposed algorithm converges more quickly and recovers the latent dynamics more accurately than existing variational and filtering baselines. The authors further compare the approach against the variational diffusion process of Archambeau et al.. They further demonstrate their method on calcium imaging data for on modeling neural dynamics during aggression.



While natural-gradient variational inference for SDEs is not entirely new, see for example  Verma et al. (2024), the present work differentiates itself through the parallelisation acceleration and the convergence analysis, both detailed in the appendix.

I find the paper well fit for the NeurIPS audience.

**Questions:**

- How would the method scale computationally with increasing latent dimension and/or increasing observation dimension (number of observed neurons)?

- For the SING-GP, how do you select inducing points? How sensitive is the performance of the method to the number and the method of selection of inducing points?

- Is there a reason you do not compare the synthetic datasets with the VDP-GP?

**Ethical Concerns:**

["NO or VERY MINOR ethics concerns only"]

**Final Justification:**

All reviewers were quite positive to accept this paper. The concerns raised during the review were minor technical questions that the authors addressed either by providing additional information or by providing additional numerical results.

**Limitations:**

the authors discuss the limitations of their approach in discussion and in the appendix.

**Quality:**

4

**Strengths And Weaknesses:**

**Strengths:**

- The paper tackles a challenging problem, that of inference of latent stochastic systems, where the posterior inference is intractable for general systems.
- The authors provide rigorous derivations.
- They provide empirical evidence on both conjugate and non-conjugate models, as well as on synthetic and experimental datasets.


**Weaknesses:**

- The authors demonstrate the results relying only on measurements up to 10 neurons. While impressive I wonder how the performance and computational demands of the method would scale if the considered neurons were on the order of hundreds.
- Variational inference with natural gradients has been introduced in previous work.

---

> ### Author Rebuttal · Authors · 2025-07-31
>
> We thank the reviewer for taking the time to read our submission and for their thoughtful response! We are especially glad that they found our work to be rigorous and a good fit for the NeurIPS audience.
>
> ## Weaknesses
> >  I wonder how the performance and computational demands of the method would scale if the considered neurons were on the order of hundreds.
>
> As with most variational inference methods, the computational scaling with respect to observation dimension primarily depends on the structure of the likelihood $p(y | x)$, which is used to compute $\mathbb{E}[\log p(y | x)]$ in the ELBO. For our work, we consider a GLM likelihood (e.g., Gaussian, Poisson) that factorizes over the observation dimension. In this case, the complexity of the expected log-likelihood scales linearly with the observation dimension and can be computed in parallel across it.
>
> We note that SING could also be applied in settings with more complex or non-factorized likelihoods, though such cases may introduce additional computational overhead. This tradeoff, however, is not specific to SING; it arises in virtually all variational inference frameworks from the expected log-likelihood in the ELBO.
>
> We conduct additional experiments that test the scalability of SING with respect to the observation dimension on synthetic data. We first simulate ground-truth latent trajectories from the SDE $dx(t) = f(x(t)) dt + dw(t)$ where $f(x)$ is a Lorenz attractor with parameters $(\rho, \sigma, \beta) = (28, 10, 8/3)$. We then simulate observations $y_n(t) \sim \mathcal{N}(c_n^\top x(t) + d_n, (\sqrt{0.05})^2)$ for $n = 1, \dots, N$, where $N$ is the observation dimension (i.e. number of neurons). Here, the entries of $c_n$ and $d_n$ are sampled *i.i.d.* from $\mathcal{N}(0, 0.1^2)$. We fit SING using a prior SDE in the same class as the generative model and learn the drift parameters. We report the RMSE between the true and inferred latents, as well as the RMSE between the true and learned dynamics at the locations of the true latents. We also report runtime in seconds on an A100 NVIDIA GPU. Each entry reports the mean and 95% confidence interval obtained from $5$ random initializations of hyperparameters.
>
> | obs dim ($N$) | latents RMSE         | dynamics RMSE        | runtime (s)  |
> |---------|----------------------|-----------------------|--------------------|
> | 10      | 0.2637 ± 0.0004      | 0.3245 ± 0.0674      | 57.02 ± 0.15     |
> | 25      | 0.2058 ± 0.0001      | 0.1523 ± 0.0025      | 57.08 ± 0.20     |
> | 50      | 0.1643 ± 0.0001      | 0.1726 ± 0.0012      | 58.05 ± 0.37     |
> | 100    | 0.1416 ± 0.0001     | 0.1315 ± 0.0028     | 58.33 ± 0.40     |
> | 200    | 0.1211 ± 0.0001     | 0.1446 ± 0.0016     | 58.22 ± 0.21     |
>
> We observe that both latents RMSE and dynamics RMSE generally improve with greater observation dimensions, likely because seeing more data improves performance. Furthermore, the runtime remains effectively constant. This is due to computing the expected Gaussian likelihood in parallel over observation dimension. While extremely high-dimensional observations (e.g., thousands of neurons) could eventually strain parallel resources, we stress that standard GLM likelihoods that factorize over observation dimension remain highly scalable in practice.
>
> > Variational inference with natural gradients has been introduced in previous work.
>
> We agree with the reviewer that natural gradient variational inference has been introduced in prior work, and we certainly build upon it in our work here. Our contribution lies in deriving closed-form natural gradient updates specifically for latent SDEs, enabling parallelization of these updates for scalability to long sequences, providing theoretical bounds on the ELBO approximation error, and extending this approach to inference in models with nonparametric (Gaussian process) drift.
>
> ## Questions
> > How would the method scale computationally with increasing latent dimension?
>
> The bottleneck with respect to latent dimension lies in computing expectations of functions of the drift as described in Section 3.3. For the experiments performed in our submission, we used Gaussian quadrature, which scales poorly with latent dimension (i.e. the number of quadrature nodes necessary to achieve fixed approximation error increases exponentially).
>
> Moreover, we have conducted additional experiments using Monte Carlo to approximate these expectations, which we find enables efficient scaling to higher-dimensional latent spaces while maintaining accurate inference.
>
> For our additional experiments, we apply SING with Monte Carlo to systems with latent dimension $D = 3, 5, 10, 20, 50$. For all of these settings we simulate ground-truth latents from an “embedded Lorenz attractor” whose first 3 dimensions evolve according to an SDE $dx(t) = f(x(t)) dt + dw(t)$ where $f(x)$ is a Lorenz attractor with parameters $(\rho, \sigma, \beta) = (28, 10, 8/3)$, and whose other dimensions $d > 3$ evolve according to a random walk with identity diffusion. We simulate observations $y_n(t) \sim \mathcal{N}(c_n^\top x(t) + d_n, (\sqrt{0.05})^2)$ for $n = 1, \dots, N$, where here we fix $N = 100$. As before, the entries of $c_n$ and $d_n$ are sampled i.i.d from $\mathcal{N}(0, 0.01)$. We report latents RMSE, dynamics RMSE, and total runtime (including compilation time) on an A100 NVIDIA GPU, along with their 95% confidence intervals obtained from $5$ different initial seeds for Monte Carlo sampling. All results are obtained by using a single sample per expectation evaluation.
>
> | latent dim ($D$) | latents RMSE        | dynamics RMSE       | runtime (s)        |
> |------------------|---------------------|----------------------|--------------------|
> | 3                | 0.1416 ± 0.0001     | 0.1241 ± 0.0107      | 58.05 ± 0.16       |
> | 5                | 0.1867 ± 0.0001     | 0.1413 ± 0.0055      | 60.11 ± 0.36       |
> | 10               | 0.2721 ± 0.0001     | 0.3102 ± 0.0059      | 89.20 ± 0.09       |
> | 20               | 0.3971 ± 0.0001     | 0.2043 ± 0.0048      | 136.28 ± 0.18      |
> | 50               | 0.6761 ± 0.0001     | 0.2193 ± 0.0059      | 561.44 ± 0.63      |
>
> Gaussian quadrature (with $6$ quadrature nodes per dimension) achieves latents and dynamics RMSE $0.1415$ and $0.1220$ in $D = 3$ latent dimensions and $0.1866$ and $0.1337$ in $D = 5$ latent dimensions.
>
> We observe that SING applied with Monte Carlo recovers the true latents and dynamics with comparable accuracy to Gaussian quadrature in $3$ and $5$ latent dimensions. Although latents RMSE continues to increase with $D$ beyond $5$ latent dimensions, this is expected for two reasons: (1) the error is computed over higher-dimensional vectors, and (2) accurate inference becomes more challenging as latent dimension increases. Additionally, we find that the runtime of joint inference and learning of SING remains tractable in up to 50 latent dimensions. We will include a description of this Monte Carlo extension along with implementation details in our revised paper.
>
> > For the SING-GP, how do you select inducing points? How sensitive is the performance of the method to the number and the method of selection of inducing points?
>
> In practice, we typically select inducing points by (1) running PCA (for Gaussian observations) or a discrete-time state-space model (for non-Gaussian observations) to estimate the scale of the latents, (2) choose a grid of inducing points that covers this region of latent space, and (3) initialize model output parameters to those obtained by the dimensionality reduction technique from step 1.
>
> In general, SING-GP tends to improve in dynamics accuracy with more inducing points, as this increases the flexibility of the variational posterior over the dynamics. However, this comes at the cost of increased computational complexity, since each update of the GP posterior costs $\mathcal{O}(TM^2)$ where $M$ is the number of inducing points and $T$ is the length of the time grid on which inference is performed. We have not yet systematically explored sensitivity to the number or selection method of inducing points for SING-GP, but we agree that this is an important direction.
>
> Several works in the GPLVM literature (e.g., [1], [2]) adopt a similar initialization strategy but subsequently treat the inducing point locations as learnable parameters, often optimized jointly with other variational parameters. These approaches could potentially be incorporated into SING-GP; for instance, it would be possible to learn inducing locations by direct optimization of the ELBO. It may also be possible to incorporate adaptive inducing points selection to SING-GP, as in [3]. We will add a discussion of these future directions to the paper.
>
> > Is there a reason you do not compare the synthetic datasets with the VDP-GP?
>
> We thank the reviewer for their comment and apologize for the confusion our plot may have caused. In Figure 2, we do indeed compare against VDP-GP (purple dashed line). SING-GP outperforms VDP-GP in terms of recovering the ground truth latents (‘latents RMSE’) as well as the ground truth Duffing Oscillator dynamics (‘dynamics RMSE’).
>
> **Thank you again for your positive comments and thoughtful response!** If you have any further questions, we would be happy to answer them.
>
> – The SING authors
>
> References:
> [1] Lalchand et al. Generalised GPLVM with stochastic variational inference. AISTATS,  2022.
> [2] Jensen et al. Manifold GPLVMs for discovering non-Euclidean latent structure in neural data. NeurIPS, 2020.
> [3] Uhrenholt et al. Probabilistic selection of inducing points in sparse Gaussian processes. Uncertainty in Artificial Intelligence, 2021.

---

> > ### Comment · Reviewer_ma3P · 2025-08-04
> >
> > I would like to thank the authors for thoroughly addressing my comments and the points raised by the other reviewers. I support acceptance of this work.
> >
> > I have technical question. You mentioned in your response regarding the selection of inducing points
> > > In practice, we typically select inducing points by (1) running PCA (for Gaussian observations) or a discrete-time state-space model (for non-Gaussian observations) to estimate the scale of the latents...
> >
> > Can you expand a bit on the case of non-Gaussian observations? Do you have any reference for this?

---

> > > ### Author Response · Authors · 2025-08-05
> > >
> > > We thank the reviewer for their positive response and support of our work!
> > >
> > > For non-Gaussian observations, we often find it beneficial to first fit a simpler state-space model to initialize the output mapping parameters and to extract smoothed latent trajectories. For example, a common observation model used in neuroscience is Poisson with exponential or softplus inverse link; for this we could fit a Poisson linear dynamical system (LDS) or switching LDS as an initialization strategy. The resulting smoothed latent trajectories provide an initial low-dimensional representation of the data (akin to PCA for Gaussian observations with identity link), which we can then use to guide our selection of inducing point locations.
> > >
> > > This strategy of using simpler model classes for initialization of non-Gaussian models is common in practice, though it is often not explicitly mentioned in related papers. One relevant line of work is Gaussian process factor analysis (GPFA) and its variants for Poisson data [1, 2], which use factor analysis to initialize the latent trajectories. Similarly, complex state-space models such as the recurrent SLDS [3, 4] by default use a simpler model, the autoregressive hidden Markov model, to initialize model parameters. We note that the choice of initialization strategy is ultimately a heuristic design choice and might depend on the data modality. Other strategies could be used for SING-GP as well; for instance, one could follow the GPFA approach and use factor analysis to initialize the model.
> > >
> > > [1] Yu et al. Gaussian-process factor analysis for low-dimensional single-trial analysis of neural population activity. NeurIPS, 2008.
> > > [2] Zhao and Park. Variational latent gaussian process for recovering single-trial dynamics from population spike trains. Neural Computation, 2017.
> > > [3] Linderman et al. Bayesian learning and inference in recurrent switching linear dynamical systems. AISTATS, 2017.
> > > [4] Zoltowski et al. A general recurrent state space framework for modeling neural dynamics during decision-making. ICML, 2020.

---

> > > > ### Comment · Reviewer_ma3P · 2025-08-05
> > > >
> > > > Great, thank you very much for answering exhaustively all my questions!

---

### Note · Authors · 2025-08-14

We again thank the reviewers for their helpful feedback on our manuscript and for supporting the acceptance of SING. We believe that we have thoroughly addressed all reviewer comments in our rebuttals, the main points of which we summarize below.

**Scalability**. We conducted additional experiments demonstrating that by replacing Gaussian quadrature with Monte Carlo, SING is capable of accurate approximative inference in high-dimensional latent SDE models (up to 50 latent dimensions).

**Comparisons**. We provided rigorous comparisons of SING to SDE Matching [1], a state-of-the-art amortized VI algorithm that can produce non-Gaussian posterior approximations. When learning both the posterior approximation and prior dynamics on the stochastic Lorenz attractor benchmark dataset, SING demonstrates competitive performance to SDE Matching and outperforms it in recovering the ground truth latents.

**Generality**. We detailed how SING can be applied to models with fixed, non-identity noise $\Sigma$ as well as to certain models with state-dependent noise e.g., the geometric Brownian motion.

These experiments, together with those in the manuscript, suggest SING as a general-purpose tool for the unsupervised discovery of latent dynamical systems from high-dimensional, noisy observations. We will release a modular and efficient codebase together with the camera-ready version so that SING can be applied out-of-the-box by practitioners. We hope that the AC and reviewers find both the practical and theoretical contributions of SING worthy of acceptance.

Kindly,

– The SING authors

[1] Bartosh et al. SDE Matching: Scalable and simulation-free training of latent stochastic differential equations. ICML, 2025.

---

### Decision · Program_Chairs · 2025-09-17

**Decision:**

Accept (poster)

**Comment:**

Authors propose a novel approach for variational inference in latent SDEs that effectively addresses the common issues of slow convergence and numerical instability. In recent years, there have been many approaches to this problem. While natural-gradient variational inference for SDEs is not a new concept, the authors successfully differentiate their work through this parallelization acceleration and a detailed convergence analysis (in the appendix). The reviewers were positive about the paper and indicated that it is a good fit for the conference. The authors also honestly acknowledged one limitation: that SING does not effectively refine its prior estimate of Lorenz dynamics with more trials. This work represents a signficant and orignal contribution relevant for both the machine learning and statistical neuroscience communities.